

# Thermal decay without information loss
# in horizonless microstate geometries

**Iosif Bena** [1*], **Pierre Heidmann** [1†], **Ruben Monten** [1‡] **and Nicholas P. Warner** [1,2◦]

**1** Institut de Physique Théorique, Université Paris Saclay, CEA, CNRS,
Orme des Merisiers, Gif sur Yvette, 91191 CEDEX, France
**2** Department of Physics and Astronomy and Department of Mathematics,
University of Southern California, Los Angeles, CA 90089, USA

⋆ iosif.bena@ipht.fr, † pierre.heidmann@ipht.fr, ‡ ruben.monten@ipht.fr, ◦ warner@usc.edu

## Abstract

We develop a new hybrid WKB technique to compute boundary-to-boundary scalar Green functions in asymptotically-AdS backgrounds in which the scalar wave equation is separable and is explicitly solvable in the asymptotic region. We apply this technique to a family of six-dimensional $\frac{1}{8}$-BPS asymptotically $AdS_3 \times S^3$ horizonless geometries that have the same charges and angular momenta as a D1-D5-P black hole with a large horizon area. At large and intermediate distances, these geometries very closely approximate the extremal-$BTZ \times S^3$ geometry of the black hole, but instead of having an event horizon, these geometries have a smooth highly-redshifted global-$AdS_3 \times S^3$ cap in the IR. We show that the response function of a scalar probe, in momentum space, is essentially given by the pole structure of the highly-redshifted global-$AdS_3$ modulated by the BTZ response function. In position space, this translates into a sharp exponential black-hole-like decay for times shorter than $N_1 N_5$, followed by the emergence of evenly spaced "echoes from the cap," with period $\sim N_1 N_5$. Our result shows that horizonless microstate geometries can have the same thermal decay as black holes without the associated information loss.

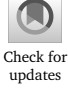
# 1 Introduction

At a time in which classical general relativity goes from triumph to triumph, the problem of information loss has grown deeper and sharper. The information problem comes from applying local field theory to a smooth vacuum spanning the horizon region [1,2] and we now know that this problem cannot be solved by small corrections to the horizon-scale physics [3–6]. The issue is then to determine what new structure must be present at the horizon, and how to describe and support it in a manner that is consistent with the incredible, large-scale successes of general relativity as evidenced at LIGO/Virgo and the EHT [7,8].

One of the most promising ideas is that the pure states that give rise to the black-hole entropy do not correspond, in the bulk, to a classical solution with a horizon, but rather to configurations of string theory that have the same large-scale structure as a classical black hole but differ from it at the scale of the horizon. Certain of these configurations can be described in supergravity, as "microstate geometries:" smooth, horizonless solutions to the supergravity equations of motion that cap off above the location of the would-be horizon of the corresponding black hole. The smooth capping-off means that such geometries give rise to unitary scattering and thus avoid the apparent loss of information associated with the presence of event horizons. Furthermore, microstate geometries for the D1-D5-P black hole can always be given $AdS_3 \times S^3$ UV asymptotics, and then they correspond holographically to certain pure states of the dual CFT.

There are many unresolved issues in the microstate geometry programme, most particularly whether solutions constructed within supergravity can capture all the microstates of a black hole. However, independent of such issues, there are now large families of explicitly-known microstate geometries that look like black holes until the observer is very close to the horizon scale. Indeed, microstate geometries provide the only known gravitational mechanism for classically supporting structure at the black-hole horizon scale, without actually forming a horizon [9–11]. As a result, even though one might not subscribe to all the goals of the microstate geometry programme, the geometries themselves provide extremely valuable theoretical laboratories for proposing, testing and probing horizon-scale microstructure. In this paper we will examine a family of such geometries in detail and study a particular scalar response function. Our goal is to show how the long throat of the microstate geometries creates black-hole-like behavior at short and intermediate times and how the cap returns information at very long time scales.

There have been several recent investigations of Green functions in three-charge microstate geometries [12–17] but many of these papers focus on the limit where the momentum charge is small. This has the advantage of allowing the application of the highly-developed AdS-CFT dictionary for the two-charge D1-D5 system, but in the small-momentum limit, the geometry is very close to that of $AdS_3 \times S^3$ and does not resemble a black hole. Here we focus on the other extreme: geometries that have large momentum and small angular momentum and hence have long BTZ throats. These are the most important geometries from the perspective of microstate geometry and fuzzball programmes, both because they resemble a black hole arbitrarily close to the horizon and also because their mass gap is exactly the same as the mass gap of the CFT states that give rise to the black hole entropy [15, 18–20].

## 1.1 The microstate geometries and their holographic duals

For largely technical reasons, the majority of explicitly-known microstate geometries are supersymmetric. One of the most interesting classes of such geometries are those of the D1-D5-P system compactified on $T^4$ or $K^3$ to six-dimensions, which is the BPS system originally used by Strominger and Vafa [21] to count black-hole microstates at vanishing string coupling. There has also been a lot of recent progress in finding the holographic duals of this system and some

of its microstates [22–27]. In particular, in [25, 27] it was shown how to obtain three-charge microstate geometries with vanishingly small angular momenta. These microstate geometries have a large region that looks exactly like an extremal BTZ black hole (times an $S^3$): They can be arranged to be asymptotic to $\text{AdS}_3 \times S^3$, but they transition to a long $\text{AdS}_2 \times S^1 \times S^3$ throat. However, unlike the BTZ black hole, this $\text{AdS}_2 \times S^1$ throat caps off smoothly at a finite, but large, depth: as one approaches the bottom, the $S^1$ shrinks with the radial coordinate, $r$, so as to create a smooth patch that looks like the origin of polar coordinates in $\mathbb{R}^{2,1}$.

These new geometries correspond to adding special classes of momentum excitations to the D1-D5 system [25, 27]. The geometries we will concentrate on here are dual to (coherent superpositions of) the CFT states at the orbifold point:

$$(\,|++\rangle_1)^{N_{++}} \left( \frac{1}{n!} (L_{-1} - J_{-1}^3)^n \,|00\rangle_1 \right)^{N_{00}}, \tag{1.1}$$

with

$$N_{++} + N_{00} = N \equiv N_1 N_5, \tag{1.2}$$

where $N_1, N_5$ are the numbers of D1 and, respectively, D5 branes and the subscripts on $|\ldots\rangle_p$ indicate the strand length of the twisted CFT state. The angular momenta of the state are determined by the number of $|++\rangle_1$ strands, while the momentum charge of the state is determined by $n$ and the number of strands involved in the second factor:

$$J_L = J_R = \frac{1}{2} N_{++}, \qquad N_P = n N_{00}. \tag{1.3}$$

When $N_{00}$ is very large and $N_{++}$ is very small, this horizonless geometry has essentially the same charges as a large extremal BTZ black hole (up to $1/N$ corrections).

In the supergravity picture, $N_{++}$ and $N_{00}$ translate into two parameters $a, b$, with

$$a^2 \propto N_{++}, \quad b^2 \propto N_{00}. \tag{1.4}$$

The constraint (1.2) on the $N$'s translates into a supergravity smoothness constraint. For more details about this notation and the class of states see [25, 27]. What is important here is that the states are characterized by three parameters, $a$, $b$ and $n$. One of the remarkable features of these microstate geometries is that one can choose $a$ and $b$ to have any values, modulo the constraint (4.7) (or, equivalently, (1.2)). The depth of the $\text{AdS}_2 \times S^1$ throat, from the $\text{AdS}_3$ region to the cap, is determined by $b/a$ and if one takes the naïve limit $a \to 0$ while keeping the UV fixed one obtains the extremal BTZ geometry.[1]

Thus, by considering very small values of $a$, one can obtain deep, scaling microstate geometries that approximate the BTZ geometry arbitrarily closely. Note that $a^2$ corresponds to the $J_R$ angular momentum of the solution, which is also proportional to the number of $|++\rangle_1$ strands of the dual CFT, so it cannot be continuously taken to zero. Hence, the throats have a maximal length.[2]

## 1.2 Capped BTZ physics and a new "hybrid WKB" method for correlators

While the geometry on the $S^3$ is quite non-trivial, it is only mildly deformed between the $\text{AdS}_3$ region and the cap. Moreover, as was noted in [34], the deformations of the $S^3$ appear to conform to a Kaluza-Klein Ansatz for reduction to three dimensions. Thus the core of the

---

[1]If one rather takes this limit by keeping fixed the IR structure of the microstate geometry, one obtains asymptotically-$\text{AdS}_2$ IR-capped microstate geometries, corresponding holographically to the pure states of the CFT dual to $\text{AdS}_2$ [20].

[2]This phenomenon was also found in microstate geometries constructed using scaling bubbling geometries [18, 28–31], but there this restriction came because of quantum effects [28, 32, 33].

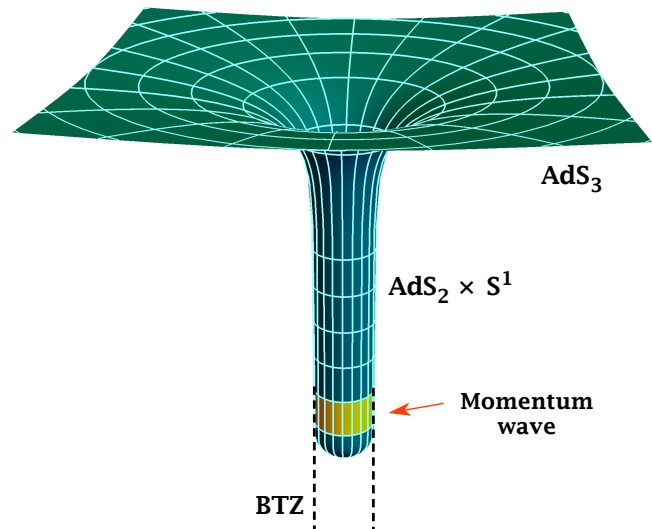

Figure 1: *Schematic description of the three-dimensional geometry of Superstrata.*

physics of these microstate geometries lies in the dynamics of the fields and the metric in $(2 + 1)$ dimensions, and so, in this paper, we will focus on the $(2 + 1)$-dimensional physics. This approach and analysis is greatly facilitated by the fact that the six-dimensional, massless scalar wave equation is separable in the full geometry [34].[3]

While separability hugely simplifies computations, there is still one very complicated ordinary differential equation that needs to be solved. This "radial" equation encodes all the interesting physics between the cap ($r = 0$) and the asymptotic AdS region ($r \to \infty$). Indeed, the capped BTZ geometry naturally comes with two intermediate physical scales. First, like the BTZ geometry, there is the scale, $r \sim \sqrt{n}\, b$, at which the momentum charge (1.3) precipitates the changeover between the AdS$_3$ region and the AdS$_2 \times S^1$ throat (see Fig. 1) and the $y$-circle stops shrinking and stabilizes at a fixed size for $r \lesssim \sqrt{n}\, b$. This is identical to what happens in the geometry of an extremal non-rotating BTZ black hole, with temperatures

$$T_L = \frac{1}{2\pi} \sqrt{\frac{N_p}{N_1 N_5}}\,, \qquad T_R = 0\,, \tag{1.5}$$

in units of the AdS length, $\ell = (Q_1 Q_5)^{1/4}$.

The second, and new, scale in the microstate geometry is $r \sim \sqrt{n}\, a$, which is the locus of the momentum wave excitation (see Fig. 1). For $r \lesssim \sqrt{n}\, a$, the $y$-circle resumes shrinking as $r$ decreases and thereby creates the smooth cap. This is the scale of the black-hole microstructure. As a result, the radial differential equation cannot be solved exactly and needs to be studied in special limits or solved by some careful approximation.

To solve the radial wave equation and to compute the holographic correlator, we introduce and develop a new "hybrid WKB" method. Traditional WKB methods are usually excellent approximations for studying bound states and quasi-normal modes but are mostly ill-adapted to the computation of holographic Green functions. The problem is that it is very hard to separate out the asymptotically leading and sub-leading modes from the asymptotic WKB wave-function: the WKB wave-function that grows at infinity generically contains some non-trivial "decaying-mode" component that is extremely hard to estimate. In the computation of boundary-to-boundary Green functions it is essential to be able to identify and separate the "decaying-modes" from the purely "growing modes." [35–41] To address this problem we

---

[3]This would not be the case for the corresponding superstrata in asymptotically flat space.

use the WKB approximation in the core of the geometry, where it is reliable, and then match this WKB solution to an exactly solvable and known class of wave-functions in the asymptotic region for intermediate and large $r$ (roughly, $\sqrt{n}\,a \ll r \ll \sqrt{n}\,b$ ). We will discuss this "hybrid WKB" in some generality but the primary application will be to match WKB wave-functions in the core of superstrata to the exactly-known BTZ wave-functions in the BTZ throat.

The presence of a cap introduces a different class of boundary conditions on the scalar wave equation. In the BTZ geometry, the boundary conditions at the horizon ($r = 0$ having taken the limit $a \to 0$) are purely absorbing. For the microstate geometry, one simply imposes regularity in the center of the cap ($r = 0$). The fact that an incoming particle, or wave, can pass through the cap and come out the other side means that scattered waves will "echo" off the cap at very long time scales. This difference in boundary conditions leads to very different physics: Bound states in microstate geometries cannot decay and are thus represented *only* by normal modes whereas "bound states" in BTZ black holes necessarily decay, with energy leaking across the horizon, and thus they are quasi-normal modes. Indeed the BTZ wave-functions are purely decaying with no oscillatory parts. Thus the momentum-space Green functions of microstate geometries have poles *only* on the real axis, while the poles for the BTZ response function lie at purely imaginary momenta.

Part of our purpose in computing the Green function is to see, in more detail, how the scattering from microstate geometries differs from the scattering off the BTZ geometry. Based on the clear separation of scales (when $b/a \gg 1$), one would expect the scattering from the microstate geometries to replicate that of BTZ scattering for time scales less than the return time of the echoes from the cap. We will indeed recover this result and see for short and intermediate times a microstate "thermal" behavior, that is identical to that of a BTZ black hole with left-moving temperature, $T_L$.

At very long time-scales $\Delta t \sim N_1 N_5$, or low frequencies $\Omega \sim \frac{1}{N_1 N_5}$, we also see strong resonant, periodic echoes off the cap at the bottom of the throat of the solution. The echoes are sharp and resonant because the microstate geometries we consider have a separable equation of motion, and hence a spherically-symmetric wave sent into these geometries will not dissipate its energy into higher spherical harmonics and will return out of the throat relatively unchanged. We expect more generic microstate geometries to give echoes that are much weaker and far more distorted.

We also find a somewhat surprising result for waves that penetrate to near the bottom of the geometry, namely a small, but significant deviation from the BTZ behavior at intermediate scales:

$$\frac{1}{N_1 N_5} \lesssim \Omega \lesssim \frac{1}{\sqrt{N_1 N_5}}, \qquad a \lesssim r \lesssim \sqrt{ab}. \tag{1.6}$$

This may be related to the observation [19] that for time-like geodesic probes dropped from the AdS$_3$ region, the tidal forces become large (exceed the compactification/Planck scale) when the probe reaches the region $a \lesssim r \lesssim \sqrt{ab}$. This phenomenon occurs in the lower section of the BTZ throat because the large blue-shifts, generated by infall from the top of the throat, amplify the small multipole corrections to the BTZ metric coming from the cap of the microstate geometry. It is possible that something similar is happening here but the effects are not as large because we are largely focussed on massless S-waves.

Geodesic probes also give us significant insight into the results we obtain for the Green functions. Even though the massless wave equation is separable, the second order, ordinary differential equations for the radial eigenfunctions are extremely complicated. We will show that WKB methods are extremely effective approximations to the wave-function. In making this approximation, one should also remember that in the WKB expansion:

$$\Psi(x^\mu) = A(x^\mu)\, e^{iS(x^\mu)}, \tag{1.7}$$

one assumes that $A(x^\mu)$ is a slowly varying amplitude and that $S(x^\mu)$ is a rapidly varying phase. Geometric optics then tells us that the function, $S(x^\mu)$, is the Hamilton-Jacobi function for null geodesics. For our Green function, there will be a region $r_0 < r < r_1$ in which the phase function, $S(x^\mu)$, is real and the solutions do indeed describe classical trapped (null) particles. For $r < r_0$ and $r > r_1$, the function $S(x^\mu)$ will be purely imaginary and corresponds to semi-classical barrier penetration. We will then have to perform the standard WKB matching at both "classical turning points," $r_0$ and $r_1$. As a by-product of these computations we will also obtain detailed information about the bound-state spectrum of the microstate geometries. As was already noted in [20], there is something of a surprise in that the discrete energies, $\omega_k$, of the bound states are remarkably linear in the integer label, $k$.

### 1.3 The structure of the paper

In Section 2, we will review the class of boundary-to-boundary Green functions, or response functions, that are of interest and how they are related to two-point functions of the holographic theory. We reduce the problem to the usual simple holographic recipe that involves solving the scalar wave equation in the background geometry. We then briefly review the elements of the WKB method and introduce our hybrid WKB strategy. This strategy requires the exact wave-functions for BTZ and $\text{AdS}_3$ and so we also briefly review the wave modes and momentum-space Green functions in these spaces.

In Section 3, we implement the hybrid WKB strategy in detail for several types of possible problems and we then focus on asymptotically-BTZ geometries for which we write the momentum-space Green function in terms of the corresponding BTZ Green function and the WKB integral that is associated with the bound-state structure of the background geometry. In Section 4 we review the geometry of $(1, 0, n)$ superstrata and set up the application of our hybrid WKB method. In Section 5 we compute the momentum-space Green function for the $(1, 0, n)$ superstrata and examine the diverse limits and features of the geometry and how they emerge from the Green function. In Section 6 we examine the position-space Green functions for the global $\text{AdS}_3$ and extremal BTZ geometries, and for the $(1, 0, n)$ superstrata. The goal is to show how to adapt the Fourier transforms that relate position-space and momentum-space Green functions for $\text{AdS}_3$ and extremal BTZ to the corresponding Green functions for superstrata. Then, in Section 6.3 we discuss the properties of the position-space Green function for the superstrata. Section 7 contains our final comments.

There are also some more technical appendices that contain details about two-point functions and propagators as well as some numerical comparisons of examples in which one knows the exact response function and for which we have used the hybrid WKB method to obtain an approximate response function.

## 2 Holographic response functions

### 2.1 Brief review of holographic correlators

A standard way to compute two-point correlation functions of an operator $\mathcal{O}$ in QFT is to couple the QFT to an external source $\int J\mathcal{O}$ and compute the response function

$$\langle \mathcal{O}\,\mathcal{O}\rangle \;=\; \frac{\delta}{\delta J}\,\langle \mathcal{O}\rangle_J \,\bigg|_{J=0}\,, \tag{2.1}$$

where $\langle \mathcal{O}\rangle_J$ is the one-point function of the operator $\mathcal{O}$, in the state of interest, in presence of the source $J$.

The holographic calculation of the two-point function of an operator $\mathcal{O}$ can be performed in an identical manner, where one uses the holographic dictionary to read off the expectation value of the operator in presence of the source. Concretely, if one considers the bulk solution for a (free) scalar field with mass $m$ dual to the boundary operator $\mathcal{O}$, near the AdS boundary ($r \to \infty$) this solution takes the form

$$\Phi(\vec{x}, t; r) = \beta(\vec{x}, t)\, r^{\Delta-d}\, (1 + \mathcal{O}(r^{-2})) + \alpha(\vec{x}, t)\, r^{-\Delta}\, (1 + \mathcal{O}(r^{-2})), \qquad (2.2)$$

where $\Delta(\Delta - 2) = m^2 l^2$. The coefficient $\alpha$ is identified with the expectation value of $\mathcal{O}$, while $\beta$ corresponds to the source. While $\alpha, \beta$ are independent as far as the asymptotic equations of motion are concerned, they become related through boundary conditions imposed by the correct physics in the interior of the spacetime. (For example, black-hole horizons require incoming boundary and smooth geometries require the absence of singularities.) It is this relation that encodes information about the state of the CFT. The holographic two-point function, (2.1), is then $\frac{\delta \alpha}{\delta \beta}$. To compute two-point functions in a given state at large $N$, it suffices to use linearized analysis of the bulk scalar and then the correlator is simply $\frac{\alpha}{\beta}$.

One should note that the are subtleties when $2\Delta \in \mathbb{Z}$. The solutions to the differential equations no longer involve distinct power series but involve log terms. Disentangling the physical correlators is more complicated but is well understood [42]. Here we will generally take $2\Delta \notin \mathbb{Z}$ and, where possible, analytically continue to integer values of $2\Delta$.

The foregoing calculation can be performed in either Euclidean or Lorentzian signature. In Euclidean signature, operators commute, and one can define a unique two-point function. The dual boundary condition in the bulk usually corresponds to smoothness in the interior of the geometry. In Lorentzian signature, several prescriptions are possible, corresponding to the different time orderings of the operators: retarded, advanced, Feynman, Wightman. In order to compute these various correlators one can work in a doubled formalism, both in the field theory and on the gravity side. If the supergravity background is a black hole, one uses the Schwinger-Keldysh formalism on the CFT side and on the gravity side one uses either the eternal black hole [39] or a doubled time contour [40, 41]. In this doubled formalism, it has been shown that the choice of sources on both contours precisely selects infalling boundary conditions at the black hole horizon [43]. The resulting correlator is the retarded propagator.

When there is no black hole in the bulk, bound states exist and it was emphasized in [40] that the choice of initial and final conditions of the early and late time slices are important. One may nevertheless wonder about the result of a non-doubled formalism for the response function, ignoring said initial and final boundary conditions. For geometries with a smooth interior, the response function we will obtain is the Feynman propagator, which is just the Wick rotation of the Euclidean correlator.

## 2.2 Brief review of WKB

For sufficiently complicated geometries, like the superstratum, one cannot solve the scalar wave equation exactly and must resort to an approximate technique to obtain the boundary-to-boundary Green functions or response functions. Here we will describe our broader strategy in adapting WKB methods to this purpose. As we will discuss, the naive application of WKB methods is, in fact, poorly suited to the computation of response functions using (2.2)[4]. However, in this section, we will describe a hybrid strategy in which we match WKB approximate wave functions calculated in the bulk to exact wave functions calculated in the asymptotic region. The result is a far more reliable, controlled and accurate technique for computing good approximations to response functions. We will apply these ideas to several examples in

---

[4]Note that WKB methods have been used before to compute correlation functions using only normalizable modes [44, 45].

Sections 3 and 4. Here we simply wish to describe the technique and explain how suitably adapted WKB methods can be used to compute correlation functions.

### 2.2.1 The basics of the WKB method

We are interested in computing the response function in backgrounds which have a separable scalar wave equation

$$\Box \Phi = m^2 \Phi, \tag{2.3}$$

so that the non-trivial part of the wave equation is a second order differential equation for the "radial function," $K(r)$, of the wave $\Phi(t, r, y_1, y_2, \ldots) \equiv Y(y_1, y_2, \ldots) K(r) e^{-i\omega t}$. It is this function that contains all the details of the geometry from interior structure, $r \sim 0$, to the asymptotic region, $r \to \infty$. The next step is to rewrite this as a Schrödinger problem. We rescale $K(r)$ to a new wave function $\Psi(r) \equiv f(r)K(r)$, for some carefully chosen function $f(r)$. We will furthermore allow for a change of variables $r \to x = x(r)$ (for which we specify requirements below) so that the radial equation is reduced to the standard Schrödinger problem:

$$\frac{d^2\Psi}{dx^2}(x) - V(x)\Psi(x) = 0, \tag{2.4}$$

for some potential, $V(x)$. This potential encodes all the details of the wave, including its energy, mass, charge and angular momenta. In practice, there are many ways to reduce a given radial equation to Schrödinger form, but these choices do not affect the essential physics of the WKB approximation[5].

First, for the WKB wave functions, $\Psi_{WKB}(x)$, to be good approximations to the exact solutions one must require that the potential itself does not fluctuate wildly. That is, it should obey the following condition:

$$\left| V(x)^{-3/2} \frac{dV}{dx} \right| \ll 1, \quad \text{when } V(x) \neq 0. \tag{2.5}$$

The "boundaries" of the Schrödinger problem depend upon the choice of $x(r)$, but here we assume that the coordinate $x$ has been chosen so that $x \to +\infty$ corresponds to the asymptotic region, $r \to \infty$ and that the other boundary is at $x \to -\infty$. In all of the problems we study, the potential, $V(x)$, will go to a constant, positive value as $x \to +\infty$:

$$\lim_{x \to +\infty} V(x) \equiv \mu^2, \qquad \mu > 0. \tag{2.6}$$

This limit defines the parameter $\mu$. For an asymptotically AdS background, we can choose the rescaling function, $f(r)$, such that this parameter is related to the mass of the particle and to the conformal dimension of the dual CFT operator as $\mu = \sqrt{1 + m^2 \ell^2} = \Delta - 1$. The shift by 1 comes from the rescaling of the wave function. As described above, we will take $2\mu \notin \mathbb{Z}$.

We have now reduced the computation of the response function to the well-understood Schrödinger problem for which WKB was originally developed. In particular, the solution will have oscillatory parts in the classically allowed region where $V(x) < 0$, and exponentially growing or decaying parts in the classically forbidden region $V(x) > 0$. The junctions between these regions, where $V(x) = 0$, are referred to as "(classical) turning points" because classical particles do not "penetrate barriers" but simply reverse course at these turning points. The requirement, (2.6), means that classical particles cannot escape to infinity, which is in accord with the behavior of massive particles in asymptotically AdS spaces.

---

[5]However, choosing the reduction to Schrödinger form carefully can make the WKB approximation algebraically simpler and, in some circumstances, more accurate.

Significantly far from the classical turning points, when (2.5) is valid, the solutions, to first order in WKB, are of the form

$$\Psi(x) = |V(x)|^{-\frac{1}{4}} \exp\left(\pm i \int |V(x)|^{\frac{1}{2}} dx\right), \quad \text{or} \quad \Psi(x) = |V(x)|^{-\frac{1}{4}} \exp\left(\pm \int |V(x)|^{\frac{1}{2}} dx\right).$$
(2.7)

The solutions on the left apply when $V(x) < 0$ and oscillate with two possible phases determined by the $\pm i$. The solutions on the right apply when $V(x) > 0$ and decay or grow depending on the $\pm$. Since the potential limits to a finite positive value at infinity, the solutions can be decomposed in a basis of growing or decaying solutions in the asymptotic region. We will denote these solutions as $\Psi_{WKB}^{\text{grow/dec}}$.

The non-trivial aspect of the WKB approximation is how to connect the classical, oscillatory solutions to the decaying and growing solutions at each turning point. This is done by expanding the potential at the turning points, $x_*$:

$$V(x) = V(x_*) + V'(x_*)(x - x_*) = V'(x_*)(x - x_*).$$
(2.8)

One then uses the fact that the Schrödinger problem, with a linear potential, has an exactly-known solution in terms of Airy functions

$$Ai(x) = \frac{1}{\pi} \int_0^\infty dt \cos\left(xt + \frac{t^3}{3}\right),$$
(2.9)

$$Bi(x) = \frac{1}{\pi} \int_0^\infty dt \left[\sin\left(xt + \frac{t^3}{3}\right) + \exp\left(xt - \frac{t^3}{3}\right)\right].$$

The oscillatory and decaying properties of Airy functions are matched to the behavior of the corresponding WKB functions in (2.7). This process of connecting solutions in distinct regions ($V(x) > 0$ and $V(x) < 0$) enables one to construct the expected two dimensional basis of WKB solutions for $-\infty < x < \infty$, and, in particular, extend $\Psi_{WKB}^{\text{grow/dec}}$ to all values of $x$.

The physical WKB wave function, $\Psi_{WKB}^{\text{phys}}(x)$, is then obtained by imposing some boundary condition or property deep in the interior region, typically as $x \to -\infty$. One then finds the unique solution (up to an overall normalization) of the form

$$\Psi_{WKB}^{\text{phys}}(x) = \Psi_{WKB}^{\text{grow}} + \mathcal{A}\,\Psi_{WKB}^{\text{dec}},$$
(2.10)

for some parameter $\mathcal{A}$ determined through the interior boundary condition.

This is extremely effective at computing bound-state and other "interior" structure but is typically quite problematic when it comes to computing the asymptotic structure of the wave function, that is essential to the evaluation of a response function. Indeed, for large $x$ where $V(x) > 0$, the physical WKB solution will have the form

$$\Psi_{WKB}^{\text{phys}}(x) = \Psi_{WKB}^{\text{grow}} + \mathcal{A}\,\Psi_{WKB}^{\text{dec}} = e^{\mu x}(1 + \ldots + e^{-2\mu x} + \ldots) + \mathcal{A}\,e^{-\mu x}(1 + \ldots).$$
(2.11)

Comparing with (2.2), one can unambiguously identify the coefficient of the growing modes at infinity, $\beta = 1$, however, the identification of the full coefficient of the decaying mode may be hard in practice because $\Psi_{WKB}^{\text{grow}}$ will generically contain a "decaying piece" that is extremely hard to determine. Thus $\alpha$ is very difficult, if not impossible, to extract purely from the WKB wave-function.

## 2.3 The WKB hybrid technique

There is a very effective way to adapt WKB techniques to the computation of response functions and, for superstrata, this approach works extremely well for much of the values of the

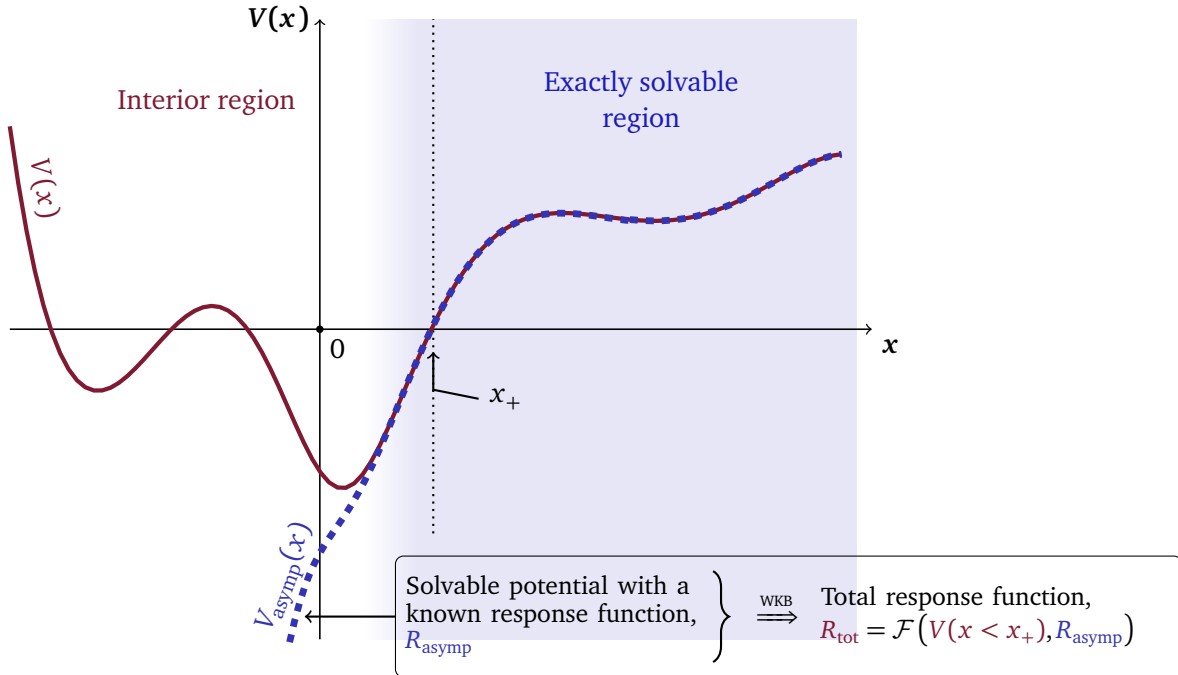

Figure 2: Schematic description of the WKB hybrid technique to derive the response function of a Schrödinger problem given by a potential $V(x)$ in red. The response function will be a deformation of the "asymptotic" response function given by the solvable asymptotic potential $V_{\text{asymp}}(x)$ with a term which only depends on the potential in the interior.

momentum and the energy. For simplicity, we assume that there is at least one classical turning point where the potential vanishes and that (2.6) is satisfied. Let $x_+$ be the outermost classical turning point, that is, one has $V(x_+) = 0$ and $V(x) > 0$ for $x > x_+$. The position of $x_+$ depends on the background and on the energy and momenta of the wave, and, in particular, changes with the energy of the wave[6].

The hybrid WKB technique supposes that there is an asymptotic potential, $V_{\text{asymp}}(x)$, that very closely approximates $V(x)$ in the region $x > x_+$ and that the Schrödinger problem for $V_{\text{asymp}}(x)$ is *exactly* and analytically solvable (see Fig.2).

$$\Psi''_{ex}(x) - V_{\text{asymp}}(x)\Psi_{ex}(x) = 0. \tag{2.12}$$

These solutions closely approximate those of the original Schrödinger problem and they can be separated into distinct and *non-overlapping* growing and decaying modes, $\Psi^{\text{grow}}_{ex}$ and $\Psi^{\text{dec}}_{ex}$, normalized according to

$$\Psi^{\text{grow}}_{ex} = e^{+\mu x}(1 + \ldots), \qquad \Psi^{\text{dec}}_{ex} = e^{-\mu x}(1 + \ldots), \qquad x \to \infty. \tag{2.13}$$

Importantly, $\Psi^{\text{grow}}_{ex}$ contains only the "purely growing" mode and has no sub-leading term involving $\Psi^{\text{dec}}_{ex}$ since $2\mu \notin \mathbb{Z}$. In the region $x > x_+$, an approximate solution to the original Schrödinger problem can now be written as in (2.2):

$$\Psi(x) \approx \beta\,\Psi^{\text{grow}}_{ex} + \alpha\,\Psi^{\text{dec}}_{ex} \sim \left[\beta\,e^{\mu x} + \alpha\,e^{-\mu x}\right], \tag{2.14}$$

---

[6]In many applications of the WKB method the energy is included explicitly and the integrals involve $|E - V(x)|$. In our formalism, $E$ is absorbed into $V(x)$.

and, by construction and to the level of approximation of $V(x)$ by $V_{\text{asymp}}(x)$, the response function is indeed given by $\alpha/\beta$. The problem is that the relationship between $\alpha$ and $\beta$ is determined by some boundary conditions deep in interior ($x \to -\infty$), where (2.14) is no longer a valid solution. The goal is thus to hybridize this exact asymptotic solution with the WKB solution and match them in a domain in which they are both reliable; more specifically, we will match them at the outermost classical turning point, $x_+$.

Putting this idea slightly differently, we will still work with the complete, physical WKB solution of the form (2.11) but use the exact solution to determine $\alpha$ and $\beta$ in terms of $\mathcal{A}$. When $x$ is close to and slightly larger than $x_+$, we have

$$
\begin{aligned}
\Psi_{WKB}^{\text{dec}}(x) &\approx a_1 \Psi_{ex}^{\text{dec}}(x), \\
\Psi_{WKB}^{\text{grow}}(x) &\approx b_1 \Psi_{ex}^{\text{grow}}(x) + b_2 \Psi_{ex}^{\text{dec}}(x), \quad x \geq x_+,
\end{aligned}
\tag{2.15}
$$

for some coefficients $a_1, b_1$ and $b_2$. Obviously $\Psi_{WKB}^{\text{dec}}(x)$ cannot have a term proportional to $\Psi_{ex}^{\text{grow}}(x)$. However, the parameter $b_2$ represents the problem ambiguity of $\Psi_{WKB}^{\text{grow}}(x)$.

The response function is then given by

$$
R_{WKB} = \frac{\alpha}{\beta} = \frac{a_1}{b_1} \mathcal{A} + \frac{b_2}{b_1}.
\tag{2.16}
$$

The ratio $a_1/b_1$ only involves the relative normalizations of $\Psi_{WKB}$ and $\Psi_{ex}$ in the asymptotic region, so it only depends on the asymptotic potential. The quantity $\mathcal{A}$ is determined by the boundary condition in the deep interior, and will be calculated by WKB approximation. Finally, it is the ratio $b_2/b_1$ that is difficult to determine purely from WKB methods but, as we will see below, this ratio can be extracted by matching the WKB and exact wavefunctions in the region around $x_+$.

In other words, one can use WKB methods to determine the solution in the region $x \leq x_+$ and then use the Airy function procedure around $x = x_+$ to match this WKB solution to the exactly solvable solution (2.14) in the region $x > x_+$. This will enable us to use the WKB method to capture all the interesting interior structure in the region $x < x_+$ and accurately extract the Green function by coupling this "interior data" to the exactly-solvable asymptotic problem.

In the next section, we will show that the WKB response function, $R_{WKB}$ in (2.16), is given by

$$
R_{WKB} = \left( \mathcal{A} + \frac{\sqrt{3}}{2} \right) e^{-2I_+} - \frac{\Psi_{ex}^{\text{grow}}(x_+)}{\Psi_{ex}^{\text{dec}}(x_+)},
\tag{2.17}
$$

where $I_+$ is defined by:

$$
I_+ \equiv -\mu x_+ + \int_{x_+}^{\infty} \left( |V(z)|^{\frac{1}{2}} - \mu \right) dz,
\tag{2.18}
$$

where we have "added and subtracted" $\mu$ to the integrand so as to handle the leading divergence in the integral, since we want to calculate $\alpha/\beta$ which remains finite. The quantity $\mathcal{A}$ encodes the information about the potential $V(x)$ for $x < x_+$ as well as the physical boundary condition imposed as $x \to -\infty$:

- If $V(x)$ has only one turning point, $x_+$, then one necessarily has $V(x) < 0$ for $x < x_+$. The interesting physical boundary conditions are those of a black hole in which the modes are required to be purely infalling as $x \to -\infty$. We will show that[7]

$$
\mathcal{A} = \text{sign}(\omega) \frac{i}{2}.
\tag{2.19}
$$

---

[7]Remember that $\omega$ is the momentum conjugate to time, $\Psi(x, t) = \Psi(x)e^{-i\omega t}$.

- If $V(x)$ has only two turning points , $x_-$ and $x_+$, then one necessarily has $V(x) \geq 0$ in the "interior region," $x < x_-$. The physical boundary condition as $x \to -\infty$ is that $\Psi$ is smooth in this limit. These are the conditions relevant to global $AdS_3$ and the (1,0,n)-superstratum background. We will show that $\mathcal{A}$ can be expressed in terms of the standard WKB "bound-state" integral:

$$\mathcal{A} = \frac{1}{2} \tan \Theta, \qquad \Theta \equiv \int_{x_-}^{x_+} |V(z)|^{\frac{1}{2}} \, dz. \tag{2.20}$$

If the potential has more than two turning points in the inner region, $\mathcal{A}$ gets more complicated but can still be computed as a function of the integrals between successive turning points of the square root of the potential. The general expression of $\mathcal{A}$ for a potential with arbitrarily many turning points is given in Appendix C.

The great strength of the WKB formula (2.17) is that it decouples the contribution of the geometry before the turning point (given by $\mathcal{A}$) from the contribution of the geometry outside the outermost turning point (given by $I_+$, $\Psi_{ex}^{\text{grow}}$ and $\Psi_{ex}^{\text{dec}}$). Since, the wave equation in the geometry beyond the last turning point has been assumed to be solvable, the exact response function in this background, $R_{ex}$, is known. One can then relate $I_+$, $\Psi_{ex}^{\text{grow}}$ and $\Psi_{ex}^{\text{dec}}$ to this exact response function. Specifically, the response function in the full background, $R_{\text{WKB}}$, can considered to be a function of the coefficient $\mathcal{A}$ and the response function of the exactly solvable problem determined by $V_{\text{asymp}}(x)$, $R_{ex}$,

$$R_{WKB} = \mathcal{F}(R_{ex}, \mathcal{A}). \tag{2.21}$$

In this way we will show that for any asymptotically BTZ metric, like the superstratum, the response function is well approximated, when $x_+$ lies in the BTZ region, by an expression of the form:

$$R_{WKB} \approx \text{Re}\left[R^{\text{BTZ}}\right] + 2\,\text{sign}(\omega)\,\mathcal{A}\,\text{Im}\left[R^{\text{BTZ}}\right], \tag{2.22}$$

where $R^{\text{BTZ}}$ is the exactly-known BTZ response function and $\mathcal{A}$ is determined by the WKB calculation in the "inner region," $x < x_+$. There are some important subtleties in applying this expression, especially relating to the frequency ranges, pole structure and the validity of the WKB approximation. We will return to these later. For the moment we simply observe that this formula makes good intuitive sense: the structure of the interior, represented by $\mathcal{A}$, is carried up the BTZ throat by the BTZ response function.

We will now give the full derivation of the expressions for $R_{WKB}$, (2.17), for potentials with one turning point and for the potentials with two turning points. We will then discuss asymptotically BTZ problems and how the result (2.17) can be re-written as (2.22). The application of this technique to the superstratum background is detailed in Section 4.

## 3 Details of the WKB analysis

In this section, we explicitly compute the formula of the response function from our hybrid WKB method, (2.17), for potentials with one or two turning points. Then, we focus on backgrounds which have an extremal-BTZ or a global-$AdS_3$ regions as the superstratum geometries. We will show the adaptability of our method to compute the response function accurately. This will require to briefly review the exact computation of the response functions in these two well-known backgrounds [40, 41].

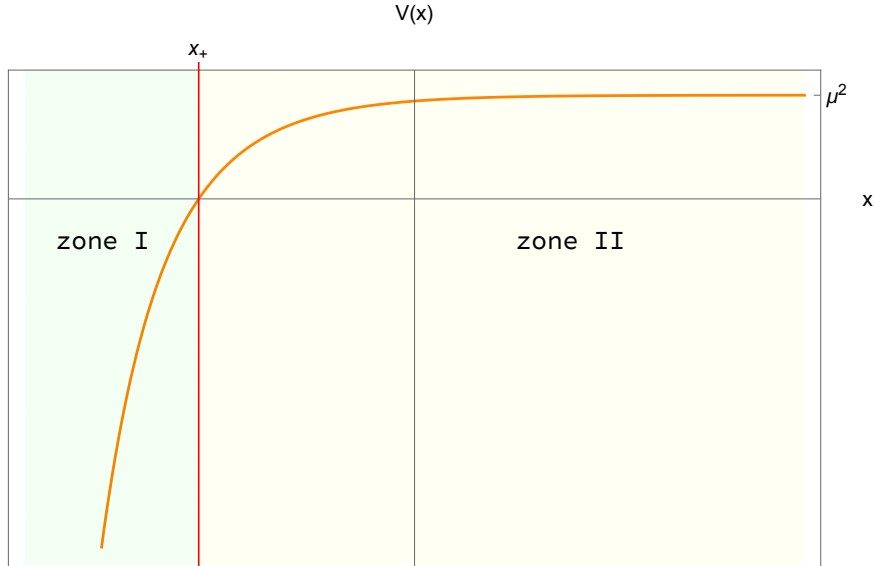

Figure 3: Schematic description of a potential $V(x)$ with one turning point, $x_+$ and the two zones depending on the sign of $V(x)$. For the illustration, we have considered that $V(x)$ tends to $-\infty$ at $-\infty$ but it is not necessary for the discussion. Any kind of behavior is possible as long as $V(-\infty) < 0$.

### 3.1 Derivation of the response function using hybrid WKB

#### 3.1.1 Potentials with one turning point

We consider a Schrödinger equation of the type (2.4) satisfying the assumptions detailed in the previous section with one turning point, $x_+$ (see Fig.3). At first order of the WKB approximation, the generic solution is

$$
\Psi(x) =
\begin{cases}
\dfrac{1}{|V(x)|^{\frac{1}{4}}} \left[ D_+^{\mathrm{I}} \exp\left( +i \int_x^{x_+} |V(z)|^{\frac{1}{2}} dz \right) + D_-^{\mathrm{I}} \exp\left( -i \int_x^{x_+} |V(z)|^{\frac{1}{2}} dz \right) \right], & x < x_+, \\[2ex]
d_+^{\mathrm{I}} \, \mathrm{Bi}\left( V'(x_+)^{\frac{1}{3}} (x - x_+) \right) + d_-^{\mathrm{I}} \, \mathrm{Ai}\left( V'(x_+)^{\frac{1}{3}} (x - x_+) \right), & x \sim x_+, \\[2ex]
\dfrac{1}{|V(x)|^{\frac{1}{4}}} \left[ D_+^{\mathrm{II}} \exp\left( + \int_{x_+}^{x} |V(z)|^{\frac{1}{2}} dz \right) + D_-^{\mathrm{II}} \exp\left( - \int_{x_+}^{x} |V(z)|^{\frac{1}{2}} dz \right) \right], & x > x_+,
\end{cases}
$$

(3.1)

where $D_\pm^{\mathrm{I/II}}$ and $d_\pm^{\mathrm{I}}$ are constants and Bi and Ai are the usual Airy functions. The usual WKB connection with Airy functions relates the constants $D_\pm^{\mathrm{I/II}}$ and $d_\pm^{\mathrm{I}}$ in (3.1) via

$$
\begin{pmatrix} d_+^{\mathrm{I}} \\ d_-^{\mathrm{I}} \end{pmatrix} \equiv e^{-i\frac{\pi}{4}} \sqrt{\pi}\, V'(x_+)^{-\frac{1}{6}} \begin{pmatrix} 1 & i \\ i & 1 \end{pmatrix} \begin{pmatrix} D_+^{\mathrm{I}} \\ D_-^{\mathrm{I}} \end{pmatrix}, \quad
\begin{pmatrix} d_+^{\mathrm{I}} \\ d_-^{\mathrm{I}} \end{pmatrix} \equiv \sqrt{\pi}\, V'(x_+)^{-\frac{1}{6}} \begin{pmatrix} 1 & 0 \\ 0 & 2 \end{pmatrix} \begin{pmatrix} D_+^{\mathrm{II}} \\ D_-^{\mathrm{II}} \end{pmatrix},
$$

(3.2)

and hence one obtains

$$
\begin{pmatrix} D_+^{\mathrm{I}} \\ D_-^{\mathrm{I}} \end{pmatrix} \equiv \begin{pmatrix} \frac{1}{2} e^{i\frac{\pi}{4}} & e^{-i\frac{\pi}{4}} \\ \frac{1}{2} e^{-i\frac{\pi}{4}} & e^{i\frac{\pi}{4}} \end{pmatrix} \begin{pmatrix} D_+^{\mathrm{II}} \\ D_-^{\mathrm{II}} \end{pmatrix}.
$$

(3.3)

We define $\Psi_{WKB}^{\mathrm{grow}}(x)$ and $\Psi_{WKB}^{\mathrm{dec}}(x)$ to be the WKB solutions that involve $\exp\left( \int_{x_+}^{x} |V|^{\frac{1}{2}} \right)$ and

$\exp\left(-\int_{x_+}^{x} |V|^{\frac{1}{2}}\right)$ at large $x$. These are obtained by settting, respectively, $D_-^{II} = 0$ and $D_+^{II} = 1$ or $D_-^{II} = 1$ and $D_+^{II} = 0$, in (3.1) and, using (3.2), we obtain:

$$
\Psi_{WKB}^{grow}(x) = \begin{cases}
\dfrac{1}{|V(x)|^{\frac{1}{4}}} \cos\left(\int_x^{x_+} |V(z)|^{\frac{1}{2}} dz + \dfrac{\pi}{4}\right), & x < x_+, \\[3mm]
\sqrt{\pi}\, V'(x_+)^{\frac{1}{6}}\, \mathrm{Bi}\left(V'(x_+)^{\frac{1}{3}}(x - x_+)\right), & x \sim x_+, \\[3mm]
\dfrac{1}{V(x)^{\frac{1}{4}}} \exp\left(+\int_{x_+}^{x} |V(z)|^{\frac{1}{2}} dz\right), & x > x_+,
\end{cases}
\tag{3.4}
$$

and

$$
\Psi_{WKB}^{dec}(x) = \begin{cases}
\dfrac{2}{|V(x)|^{\frac{1}{4}}} \cos\left(\int_x^{x_+} |V(z)|^{\frac{1}{2}} dz - \dfrac{\pi}{4}\right), & x < x_+, \\[3mm]
2\sqrt{\pi}\, V'(x_+)^{\frac{1}{6}}\, \mathrm{Ai}\left(V'(x_+)^{\frac{1}{3}}(x - x_+)\right), & x \sim x_+, \\[3mm]
\dfrac{1}{V(x)^{\frac{1}{4}}} \exp\left(-\int_{x_+}^{x} |V(z)|^{\frac{1}{2}} dz\right), & x > x_+.
\end{cases}
\tag{3.5}
$$

The response function is given by (2.16), where the coefficients $a_1$ and $b_1$, defined in (2.15), are determined using the leading terms in (3.4) and (3.5) and the normalizations in (2.13). Indeed, one finds:

$$
a_1 = \mu^{-1/2} e^{-I_+}, \qquad b_1 = \mu^{-1/2} e^{I_+},
\tag{3.6}
$$

where $I_+$ is given in (2.18). The coefficient $b_2$ can be obtained by evaluating both sides of (2.15) at any finite value of $x \geq x_+$. It is convenient to evaluate $b_2$ at $x = x_+$. One then obtains:

$$
b_2 = a_1 \frac{\Psi_{WKB}^{grow}(x_+)}{\Psi_{WKB}^{dec}(x_+)} - b_1 \frac{\Psi_{ex}^{grow}(x_+)}{\Psi_{ex}^{dec}(x_+)}.
\tag{3.7}
$$

From (3.4) and (3.5) one finds

$$
\frac{\Psi_{WKB}^{grow}(x_+)}{\Psi_{WKB}^{dec}(x_+)} = \frac{\mathrm{Bi}(0)}{2\,\mathrm{Ai}(0)} = \frac{\sqrt{3}}{2}.
\tag{3.8}
$$

The constant $\mathcal{A}$ is determined by the physical boundary condition that the wave must satisfy at $x \to -\infty$. For one turning point, we can ensure ingoing boundary conditions by choosing the physical WKB solution to be

$$
\Psi_{WKB}^{phys}(x) \equiv \Psi_{WKB}^{grow}(x) + \mathrm{sign}(\omega) \frac{i}{2} \Psi_{WKB}^{dec}(x),
\tag{3.9}
$$

where $\omega$ is the momentum conjugate to time at the horizon. This result corresponds to $\mathcal{A} = \mathrm{sign}(\omega)\frac{i}{2}$ and observe that, by construction, one has, for $x < x_+$,

$$
\Psi_{WKB}^{phys}(x) = |V(x)|^{-\frac{1}{4}} \exp\left[\mathrm{sign}(\omega)\, i \left(\int_x^{x_+} |V(z)|^{\frac{1}{2}} dz + \frac{\pi}{4}\right)\right].
\tag{3.10}
$$

The important point is that because $V(x) < 0$ for $x < x_+$, the integral in (3.10) is monotonically decreasing as $x$ increases. Therefore, when this wave-function is multiplied by a

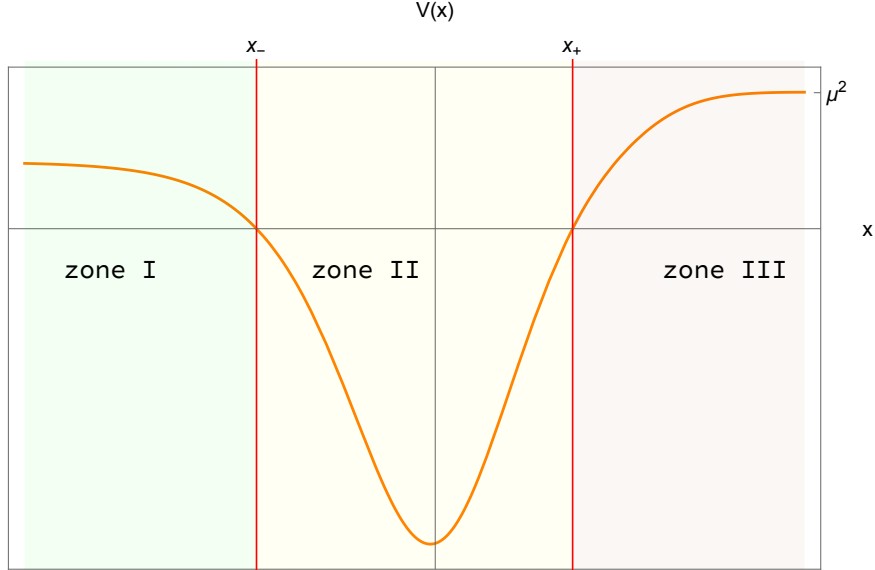

Figure 4: Schematic description of a potential $V(x)$ with the two turning points, $x_-$ and $x_+$ and the three zones depending on the sign of $V(x)$. For the illustration, we have considered that $V(x)$ tends to a constant at $-\infty$ but it is not necessary for the discussion. Any kind of behavior is possible as long as $V(-\infty) \geq 0$.

time-dependent phase, $e^{-i\omega t}$, $\Psi_{WKB}^{\text{phys}}$ is a purely "infalling" wave function at $x \to -\infty$. The resulting response function can be read off from (2.16)

$$R_{\text{WKB}} = \frac{1}{2} \left( \sqrt{3} + \text{sign}(\omega) i \right) e^{-2I_+} - \frac{\Psi_{ex}^{\text{grow}}(x_+)}{\Psi_{ex}^{\text{dec}}(x_+)}. \tag{3.11}$$

To illustrate this result, we will consider asymptotically extremal BTZ backgrounds as well as the exactly extremal BTZ background in Section 3.2 and Appendix D.

### 3.1.2 Potentials with two turning points

We perform a similar computation by deriving the response function of a Schrödinger equation, (2.4), with now two classical turning points, $x_-$ and $x_+$ (see Fig.4). In the WKB approximation, the generic solution consists of the usual growing or decaying exponentials in the regions where $V(x) > 0$, oscillatory solutions where $V(x) < 0$ which are then connected by Airy functions:

$$\Psi(x) = \begin{cases} \dfrac{1}{|V(x)|^{\frac{1}{4}}} \left[ D_+^{\text{I}} \exp\left( \int_x^{x_-} |V(z)|^{\frac{1}{2}} dz \right) + D_-^{\text{I}} \exp\left( -\int_x^{x_-} |V(z)|^{\frac{1}{2}} dz \right) \right], & x < x_-, \\[4mm] d_+^{\text{I}} \, \text{Bi}\left( -|V'(x_-)|^{1/3} (x - x_-) \right) + d_-^{\text{I}} \, \text{Ai}\left( -|V'(x_-)|^{1/3} (x - x_-) \right), & x \sim x_-, \\[4mm] \dfrac{1}{|V(x)|^{\frac{1}{4}}} \left[ D_+^{\text{II}} \exp\left( i \int_{x_-}^x |V(z)|^{\frac{1}{2}} dz \right) + D_-^{\text{II}} \exp\left( -i \int_{x_-}^x |V(z)|^{\frac{1}{2}} dz \right) \right], & x_- < x < x_+, \\[4mm] d_+^{\text{II}} \, \text{Bi}\left( |V'(x_-)|^{1/3} (x - x_-) \right) + d_-^{\text{II}} \, \text{Ai}\left( |V'(x_-)|^{1/3} (x - x_-) \right), & x \sim x_+, \\[4mm] \dfrac{1}{|V(x)|^{\frac{1}{4}}} \left[ D_+^{\text{III}} \exp\left( \int_{x_+}^x |V(z)|^{\frac{1}{2}} dz \right) + D_-^{\text{III}} \exp\left( -\int_{x_+}^x |V(z)|^{\frac{1}{2}} dz \right) \right], & x > x_+. \end{cases} \tag{3.12}$$

The connection formulae at each junction determines all the unknown coefficients in terms of $D_+^{\mathrm{I}}$ and $D_-^{\mathrm{I}}$:

$$
\begin{pmatrix} d_+^{\mathrm{I}} \\ d_-^{\mathrm{I}} \end{pmatrix} \equiv \frac{\sqrt{\pi}}{V'(x_-)^{\frac{1}{6}}} \begin{pmatrix} 1 & 0 \\ 0 & 2 \end{pmatrix} \begin{pmatrix} D_+^{\mathrm{I}} \\ D_-^{\mathrm{I}} \end{pmatrix}, \qquad \begin{pmatrix} d_+^{\mathrm{II}} \\ d_-^{\mathrm{II}} \end{pmatrix} \equiv \frac{\sqrt{\pi}}{V'(x_-)^{\frac{1}{6}}} \begin{pmatrix} -\sin\Theta & 2\cos\Theta \\ \cos\Theta & 2\sin\Theta \end{pmatrix} \begin{pmatrix} D_+^{\mathrm{I}} \\ D_-^{\mathrm{I}} \end{pmatrix},
$$

$$
\begin{pmatrix} D_+^{\mathrm{II}} \\ D_-^{\mathrm{II}} \end{pmatrix} \equiv \begin{pmatrix} \frac{1}{2} e^{i\frac{\pi}{4}} & e^{-i\frac{\pi}{4}} \\ \frac{1}{2} e^{-i\frac{\pi}{4}} & e^{i\frac{\pi}{4}} \end{pmatrix} \begin{pmatrix} D_+^{\mathrm{I}} \\ D_-^{\mathrm{I}} \end{pmatrix}, \qquad \begin{pmatrix} D_+^{\mathrm{III}} \\ D_-^{\mathrm{III}} \end{pmatrix} \equiv \begin{pmatrix} -\sin\Theta & 2\cos\Theta \\ \frac{1}{2}\cos\Theta & \sin\Theta \end{pmatrix} \begin{pmatrix} D_+^{\mathrm{I}} \\ D_-^{\mathrm{I}} \end{pmatrix}, \tag{3.13}
$$

where

$$
\Theta \equiv \int_{x_-}^{x_+} |V(z)|^{\frac{1}{2}}\, dz\,. \tag{3.14}
$$

There are obvious parallels between these connection formula and those in (3.2). The only new element is $\Theta$, which appears in the formulas because of the trivial identity:

$$
\exp\left( \pm i \int_{x_-}^{x} |V(z)|^{\frac{1}{2}}\, dz \right) \;=\; \exp(\pm i\Theta) \exp\left( \mp i \int_{x}^{x_+} |V(z)|^{\frac{1}{2}}\, dz \right). \tag{3.15}
$$

The connection formulae at $x = x_+$ are simple expressions akin to those in (3.2) when the WKB wave-functions are written in terms of $\int_{x}^{x_+} |V(z)|^{\frac{1}{2}}\, dz$, however the expressions in (3.13) for Region II are written in terms of $\int_{x_-}^{x} |V(z)|^{\frac{1}{2}}\, dz$ and so (3.15) is needed to convert these expressions before using the simple connection formulae at $x_+$.

The decaying and "growing" WKB modes can now be isolated by setting $D_+^{\mathrm{III}} = 0$ or $D_-^{\mathrm{III}} = 0$, respectively. Following our hybrid WKB strategy, we again assume that, for $x > x_+$ there is an exactly-known pair of wave-functions $\Psi_{ex}^{\mathrm{grow}}$ and $\Psi_{ex}^{\mathrm{dec}}$. As in Section 3.1.1, one can match these to the decaying and growing WKB modes at $x_+$, to arrive at essentially the same results as in (3.6) and (3.7).

The physical wave function should be regular in the interior of the geometry and so the physical wave function should not blow up as $x \to -\infty$. This means that $\Psi_{WKB}^{\mathrm{phys}}$ is given by (3.12) with $D_+^{I} = 0$ and hence

$$
\Psi_{WKB}^{\mathrm{phys}}(x) \;=\; \cos\Theta\, \Psi_{WKB}^{\mathrm{grow}}(x) + \frac{1}{2}\sin\Theta\, \Psi_{WKB}^{\mathrm{dec}}(x). \tag{3.16}
$$

This gives the response function in (2.17) with $\mathcal{A} = \frac{1}{2}\tan\Theta$.

To illustrate this result, we will compute the response function in the global-AdS$_3$ geometry via the WKB approximation in Section 3.3, and we will examine the accuracy of the approximation in Appendix D, where we will compare the WKB result to the exact result.

## 3.2 Response function in asymptotically extremal BTZ geometries

Our main goal is to apply our WKB hybrid technique on asymptotically extremal BTZ geometries, such as superstrata. We will be able to relate their response functions to the response function of an extremal BTZ black hole. This naturally requires a good understanding of the later which we will quickly review here.

The standard form of the metric outside an extremal BTZ black hole in "Schwarzschild" coordinates is given by:

$$
ds^2 \;=\; -\ell^2 f(\rho)\, dt^2 + \frac{d\rho^2}{f(\rho)} + \rho^2 \left( dy - \frac{r_H^2}{\rho^2}\, dt \right)^2, \qquad f(\rho) = \frac{(\rho^2 - r_H^2)^2}{\ell^2 \rho^2}, \tag{3.17}
$$

where the coordinate $y$ is identified as $y \sim y + 2\pi R_y$ and the AdS length is given by $\ell$. This solution has mass and angular momentum $J = M\ell = \frac{r_H^2}{4G\ell}$. It is more convenient to work in terms of a new radial coordinate $r^2 \equiv (\rho^2 - r_H^2)$, so that $r = 0$ corresponds to the horizon, and $r \to \infty$ is the conformal boundary and with the null coordinates $u \equiv y + t$ and $v \equiv y - t$. The metric in this coordinate system is given by:

$$ds^2 = r^2 \, du \, dv + r_H^2 \, dv^2 + \ell^2 \, \frac{dr^2}{r^2}. \tag{3.18}$$

The wave equation for a scalar field $\Phi$ of mass $m^2\ell^2 = \Delta(\Delta - 2)$, with $\Delta > 2$ can be solved with the Ansatz

$$\Phi = e^{-i(\omega u + p v)} K(r) \tag{3.19}$$

and is given by the Klein–Gordon equation:

$$\frac{1}{\ell^2 r} \partial_r \left( r^3 \, \partial_r K \right) \; - \; \left( \frac{\Delta(\Delta - 2)}{\ell^2} + \frac{4\omega p}{r^2} - \frac{4\omega^2 r_H^2}{r^4} \right) K = 0. \tag{3.20}$$

For convenience, we set the radius $\ell$ to 1 which can be also reabsorbed by scaling $(\ell\omega, \ell p) \to (\omega, p)$.

### 3.2.1 Exact treatment for extremal BTZ black holes

It is easy to see that in terms of a new variable $\tilde{r} = 1/r^2$, this is a confluent hypergeometric equation whose solutions are Whittaker functions[8]:

$$K(r) = c_1 \, \text{M}\left( \frac{i p}{2 r_H}, \frac{1}{2}(\Delta - 1), \frac{2i\omega r_H}{r^2} \right) + c_2 \, \text{M}\left( \frac{i p}{2 r_H}, \frac{1}{2}(1 - \Delta), \frac{2i\omega r_H}{r^2} \right). \tag{3.21}$$

The Whittaker $M$ functions have the virtue of having a simple expansion around the conformal boundary ($r \to \infty$):

$$K = c_1 \left( \frac{2i\omega r_H}{r^2} \right)^{\Delta/2} [1 + \mathcal{O}(r^{-2})] + c_2 \left( \frac{2i\omega r_H}{r^2} \right)^{1 - \Delta/2} [1 + \mathcal{O}(r^{-2})].$$

The response function is given by:

$$R^{\text{BTZ}} = \frac{c_1}{c_2} \left( 2i\omega r_H \right)^{\Delta - 1}. \tag{3.22}$$

The ratio $c_1/c_2$ is determined by the boundary conditions in the interior. For the BTZ black hole, the solution must be purely ingoing at the horizon ($r = 0$).

The expansion of the solution (3.21) at $r = 0$ is more complicated:

$$c_1 \Gamma(\Delta) \left[ \frac{\left( \frac{2i\omega r_H}{r^2} \right)^{-\frac{ip}{2r_H}} e^{\frac{ir_H\omega}{r^2}}}{\Gamma\left( \frac{1}{2}(\Delta - \frac{ip}{r_H}) \right)} + e^{i\frac{\pi}{2}\Delta \, \text{sign}(\omega)} \frac{\left( \frac{-2i\omega r_H}{r^2} \right)^{\frac{ip}{2r_H}} e^{-\frac{ir_H\omega}{r^2}}}{\Gamma\left( \frac{1}{2}(\Delta + \frac{ip}{r_H}) \right)} \right] + c_2 \left( \Delta \leftrightarrow (2 - \Delta) \right). \tag{3.23}$$

---

[8]For $2\Delta \in \mathbb{Z}$, these two solutions are linearly dependent and it is necessary to use the Whittaker-$W$ function instead.

To ensure ingoing boundary conditions at the horizon, the coefficient of $e^{-\frac{ir_H\omega}{r^2}}$ must vanish[9] (remember that the horizon is at $r = 0$). This leads to the condition

$$c_1 \frac{\Gamma(\Delta)\,e^{i\frac{\pi}{2}\Delta\,\mathrm{sign}(\omega)}}{\Gamma\big(\frac{1}{2}(\Delta+\frac{ip}{r_H})\big)} \;+\; c_2 \frac{\Gamma(2-\Delta)\,e^{i\frac{\pi}{2}(2-\Delta)\,\mathrm{sign}(\omega)}}{\Gamma\big(1-\frac{1}{2}(\Delta-\frac{ip}{r_H})\big)} \;=\; 0\,. \tag{3.24}$$

Combining this with (3.22), we find the holographic "response" in momentum space [39]:

$$R^{\mathrm{BTZ}}(\omega,p) \;=\; -(-2i\,\omega\,r_H)^{\Delta-1}\,\frac{\Gamma(2-\Delta)\,\Gamma\big(\frac{1}{2}(\Delta+\frac{ip}{r_H})\big)}{\Gamma(\Delta)\,\Gamma\big(1-\frac{1}{2}(\Delta-\frac{ip}{r_H})\big)}\,. \tag{3.25}$$

In Appendix B we will compare this result to the thermal CFT two-point function (in the appropriate extremal limit). Ignoring a $\Delta$-dependent factor, which can be absorbed in the normalization of the CFT operator $\mathcal{O}$, we will see that this is the retarded propagator (B.11) with inverse left-moving temperature $\beta_L = R_y\pi/r_H$.

As we will see in Section 6, the Fourier transformation towards the Green function in position space is sensitive to the pole structure of (3.25). Because of the ingoing boundary conditions, these poles correspond to the quasi-normal mode frequencies of the BTZ black hole (as opposed to normal modes in horizonless geometries). They are located on the imaginary axis, spaced by the temperature of the black hole. The fact that they all have a positive imaginary part will single out the retarded propagator in position space.

### 3.2.2 Hybrid WKB for an asymptotically BTZ solution

We now consider the response function for a general, asymptotically extremal BTZ geometry. We assume that the scalar wave equation closely matches the BTZ wave equation (3.20) outside a certain radius. There are several ways to convert this equation to the standard Schrödinger form (2.4). We choose a version that leads to the simplest mode expansions:

$$K(r) \equiv \frac{\Psi(r)}{r}\,, \qquad x \equiv \ln r\,, \quad x \in \mathbb{R}\,. \tag{3.26}$$

The Schrödinger potential in the asymptotic region is

$$V(x) \sim V_{\mathrm{BTZ}}(x) \equiv (\Delta-1)^2 + 4\,\omega p\,e^{-2x} - 4\,\omega^2 r_H^2\,e^{-4x}\,. \tag{3.27}$$

This potential approaches $(\Delta-1)^2$ at the boundary $x \sim \infty$ and so $\mu = \Delta - 1 > 0$. The BTZ horizon would be at $x \to -\infty$. The BTZ potential has a unique classical turning point, $x_+$, given by:

$$x_+ \equiv \frac{1}{2}\ln\left[\frac{2}{(\Delta-1)^2}\left(|\omega|\sqrt{p^2 + r_H^2(\Delta-1)^2} - \omega p\right)\right]\,. \tag{3.28}$$

For general asymptotically BTZ geometries, we consider a regime of $\omega$ and $p$ where the outermost classical turning point of $V(x)$, $x_+$, is inside the BTZ region: $V(x) \sim V_{\mathrm{BTZ}}(x)$, $x \gtrsim x_+$. This requires that the energy of the mode is higher than a certain value.

In the inner region, $x < x_+$, the potential can take a very complicated form. As explained in Section 2.3, the relevant quantity to describe the inner region is the quantity $\mathcal{A}$ which has different expressions according to the form of the potential. Whenever $x_+$ is in the BTZ region, the other quantities that enter in the WKB hybrid formula for the response function (2.17) can

---

[9]Indeed, close to the horizon of the extremal BTZ geometry, a set of orthogonal Killing vectors is given by $\partial_t$ and $\partial_v$. The conjugate momenta can be read off from (3.19): $\Phi = e^{-i(2\omega t+(p+\omega)v)}K(r)$. Therefore, $2\omega$ is really the momentum conjugate to $t$ near the black hole horizon.

be derived from $V_{\text{BTZ}}(x)$. The expression for $I_+$, defined in (2.18), is an elementary integral leading to:

$$e^{-2I_+} = (\Delta-1)^{2-2\Delta}\,|\omega|^{\Delta-1}\,\left(p^2+r_H^2(\Delta-1)^2\right)^{\frac{\Delta-1}{2}}$$
$$\times \exp\left(\Delta-1-\frac{2p}{r_H}\arctan\left[\frac{r_H(\Delta-1)}{\sqrt{p^2+r_H^2(\Delta-1)^2}-p}\right]\right). \tag{3.29}$$

For $\Psi_{ex}^{\text{grow}}(x)$ and $\Psi_{ex}^{\text{dec}}(x)$, we simply use the growing and decaying scalar modes in an extremal-BTZ black hole (3.21). In our coordinate system, this gives

$$\Psi_{ex}^{\text{grow}}(x) = (-2i\,r_H\,\omega)^{\frac{\Delta}{2}-1}\,e^{(2-\Delta)x}\,\text{M}\left(-\frac{ip}{2r_H},-\frac{\Delta-1}{2},-2i\,r_H\,\omega\,e^{-2x}\right),$$
$$\Psi_{ex}^{\text{dec}}(x) = (-2i\,r_H\,\omega)^{-\frac{\Delta}{2}}\,e^{\Delta x}\,\text{M}\left(-\frac{ip}{2r_H},\frac{\Delta-1}{2},-2i\,r_H\,\omega\,e^{-2x}\right). \tag{3.30}$$

It should be noted that these functions are actually real for real values of $\omega, p, r_H, \Delta$ and $x$. We now have all the ingredients of (2.17), except for $\mathcal{A}$, which depends entirely on the details of the "inner region".

As a special case, we can apply our WKB method to the BTZ geometry itself. Indeed, the BTZ potential only has one classical turning point and so the WKB response function is given by (3.11), and, in particular, we have $\mathcal{A} = \frac{1}{2}\text{sign}(\omega)\,i$ in (2.17). We therefore conclude that within the validity of the WKB approximation we must have

$$R^{\text{BTZ}}(\omega,p) \approx \frac{1}{2}\left(\sqrt{3}+\text{sign}(\omega)\,i\right)e^{-2I_+} - \frac{\Psi_{ex}^{\text{grow}}(x_+)}{\Psi_{ex}^{\text{dec}}(x_+)}, \tag{3.31}$$

where $R^{\text{BTZ}}(\omega,p)$ is given by (3.25). Taking the real and imaginary parts of this expression, we arrive at the approximate identities:

$$\frac{\sqrt{3}}{2}\,e^{-2I_+} - \frac{\Psi_{ex}^{\text{grow}}(x_+)}{\Psi_{ex}^{\text{dec}}(x_+)} \approx \text{Re}\left[R^{\text{BTZ}}\right], \qquad \frac{\text{sign}(\omega)}{2}\,e^{-2I_+} \approx \text{Im}\left[R^{\text{BTZ}}\right], \tag{3.32}$$

where we have used the fact that $e^{-2I_+}$, $\Psi_{ex}^{\text{grow}}(x_+)$ and $\Psi_{ex}^{\text{dec}}(x_+)$ are all real. Using these expressions in (2.17) we arrive at the result advertised in (2.22) for a generic, asymptotically BTZ response function, $R_{WKB}$,

$$R_{WKB}(\omega,p) \approx \text{Re}\left[R^{\text{BTZ}}(\omega,p)\right] + 2\,\text{sign}(\omega)\,\mathcal{A}\,\text{Im}\left[R^{\text{BTZ}}(\omega,p)\right]. \tag{3.33}$$

This is our primary result for the general WKB analysis of asymptotically BTZ metrics. This formula has been computed for $2\Delta$ non-integer but the analytic continuation to integer values of $2\Delta$ is well-defined since we know the expression of $R^{\text{BTZ}}$ for $\Delta$ integer from the literature. The formula (3.33) is only valid when the largest turning point is inside the BTZ region, in other words, in a regime of momenta where the waves stop oscillating at a distance which lies inside the BTZ region.

It is also important to note that, in deriving (2.17), and especially in making the identifications (3.32) that led to (2.22), we crucially required both $\omega$ and $p$ to be real. This will be very important for understanding the pole structure of the response function.

In addition to obtaining the formula (3.33), the application of the WKB technique to the BTZ metric also illustrates the method very simply and affords us the opportunity to assess the accuracy of the WKB approximation. Indeed, one can numerically evaluate and compare both sides of the approximate identities (3.32) using (3.28), (3.29), (3.30) and (3.25). The details of this can be found in Appendix D.1, where we show that, even for relatively small values of $\Delta$, the accuracy is within a few percent and that the accuracy greatly improves as $\Delta$ increases.

### 3.3 Model response function in the interior: global AdS$_3$

The superstratum geometries we will explore look like BTZ geometries that end in the IR with a smooth global AdS$_3$ cap. In this section we briefly review the computation of the response function in global AdS$_3$ and we apply our WKB hybrid method to check the suitability of this method to such backgrounds. The global Lorentzian AdS$_3$ metric of radius $\ell$ is:

$$
\begin{aligned}
ds^2 &= \frac{\ell^2 \, dr^2}{r^2 + \ell^2} - (r^2 + \ell^2) \frac{dt^2}{R_y^2} + r^2 \frac{dy^2}{R_y^2} \\
&= \frac{\ell^2 \, dr^2}{r^2 + \ell^2} - \frac{\ell^2}{4R_y^2}(du - dv)^2 + \frac{r^2}{R_y^2} \, du \, dv,
\end{aligned}
\tag{3.34}
$$

where $0 \leq y < 2\pi R_y$ and $u = y + t$ and $v = y - t$. The wave equation for a scalar field $\Phi = e^{-i(\omega u + p v)/R_y} K(r)$ of mass $m^2 \ell^2 = \Delta(\Delta - 2)$ is given by the Klein–Gordon equation:

$$
\frac{1}{\ell^2 r} \partial_r \left( r \left( r^2 + \ell^2 \right) \partial_r K \right) - \left( \frac{\Delta(\Delta - 2)}{\ell^2} - \frac{(\omega - p)^2}{r^2 + \ell^2} + \frac{(\omega + p)^2}{r^2} \right) K = 0.
\tag{3.35}
$$

Moreover, we assume without loss of generality that $\Delta > 1$. For convenience, we set the radius $\ell$ to 1 which can be also reabsorbed by scaling $r/\ell \to r$.

#### 3.3.1 Exact analysis for global AdS$_3$

The Klein–Gordon equation can be reduced to a standard hypergeometric equation whose solutions can be written as a linear combination of:

$$
K_1(\omega, p; r) \equiv r^{(\omega + p)} (r^2 + 1)^{\frac{1}{2}(\omega - p)} \, _2F_1\left( 1 + \omega - \tfrac{1}{2}\Delta, \omega + \tfrac{1}{2}\Delta, \omega + p + 1; -r^2 \right),
\tag{3.36}
$$

$$
K_2(\omega, p; r) \equiv r^{-(\omega + p)} (r^2 + 1)^{\frac{1}{2}(\omega - p)} \, _2F_1\left( 1 - \tfrac{1}{2}\Delta - p, \tfrac{1}{2}\Delta - p, 1 - \omega - p; -r^2 \right).
\tag{3.37}
$$

These functions are defined by their power expansion about $r = 0$ in which the generic term is $r^{2n \pm (\omega + p)}$, $n \in \mathbb{Z}, n \geq 0$. It is worth noting that in the neighbourhood of $r = 0$, the wave equation, (3.35), becomes, at leading order, the Laplace equation of flat $\mathbb{R}^{2,1}$ written in polar coordinates:

$$
\frac{1}{r} \partial_r \left( r \, \partial_r K \right) - \frac{(\omega + p)^2}{r^2} K = 0.
\tag{3.38}
$$

It is this equation that fixes the leading powers of $r$ in (3.37). The solution that is regular at $r = 0$ is thus

$$
K_{reg}(\omega, p; r) = \theta(\omega + p) K_1(\omega, p; r) + \theta(-(\omega + p)) K_2(\omega, p; r),
\tag{3.39}
$$

where $\theta(\omega)$ is the Heaviside step function.

One can also expand about infinity by using the inversion, $r \to \frac{1}{r}$, and one finds that an equivalent basis of solutions is given by:

$$
\widetilde{K}_1(\omega, p; r) \equiv r^{(\omega - p + \Delta - 2)} (r^2 + 1)^{\frac{1}{2}(p - \omega)} \, _2F_1\left( 1 + p - \tfrac{1}{2}\Delta, 1 - \omega - \tfrac{1}{2}\Delta, 2 - \Delta; -r^{-2} \right),
\tag{3.40}
$$

$$
\widetilde{K}_2(\omega, p; r) \equiv r^{(\omega - p - \Delta)} (r^2 + 1)^{\frac{1}{2}(p - \omega)} \, _2F_1\left( p + \tfrac{1}{2}\Delta, \tfrac{1}{2}\Delta - \omega, \Delta; -r^{-2} \right).
\tag{3.41}
$$

These functions are defined by their expansions as $r \to \infty$:

$$\widetilde{K}_1(\omega, p; r) = r^{(\Delta-2)} \sum_{n=0}^{\infty} a_n r^{-2n}, \qquad \widetilde{K}_2(\omega, p; r) = r^{-\Delta} \sum_{n=0}^{\infty} b_n r^{-2n}. \tag{3.42}$$

Since $\Delta > 2$, $\widetilde{K}_1$ and $\widetilde{K}_2$ purely contain non-normalizable and normalizable modes. Note that if $\Delta \notin \mathbb{N}_{>2}$ then there can be no mixing of these series and so $\widetilde{K}_1$ unambiguously represents the purely non-normalizable mode.

Finally, note that the wave equation (3.35) is invariant under $(\omega, p) \to -(\omega, p)$ and one can use the Euler transformation of the hypergeometric functions to verify that under this transformation $K_1 \leftrightarrow K_2$ while the $\widetilde{K}_j$ are individually invariant.

**Response functions**

To get the boundary-to-boundary Green function, or response function, for AdS$_3$, one should expand $K_{reg}$ in (3.39) around infinity. In particular, one finds for $K_1$ and $K_2$:

$$K_1(\omega, p; r) \sim r^{-\Delta} \Gamma(1 + \omega + p)$$
$$\left[ \frac{\Gamma(1-\Delta)}{\Gamma(1+p-\frac{1}{2}\Delta)\Gamma(1+\omega-\frac{1}{2}\Delta)} + r^{2(\Delta-1)} \frac{\Gamma(\Delta-1)}{\Gamma(p+\frac{1}{2}\Delta)\Gamma(\omega+\frac{1}{2}\Delta)} \right], \tag{3.43}$$

$$K_2(\omega, p; r) \sim r^{-\Delta} \Gamma(1 - \omega - p)$$
$$\left[ \frac{\Gamma(1-\Delta)}{\Gamma(1-\omega-\frac{1}{2}\Delta)\Gamma(1-p-\frac{1}{2}\Delta)} + r^{2(\Delta-1)} \frac{\Gamma(\Delta-1)}{\Gamma(\frac{1}{2}\Delta-\omega)\Gamma(\frac{1}{2}\Delta-p)} \right]. \tag{3.44}$$

Taking the ratio of normalizable and non-normalizable parts yields to a response function for each of the solutions

$$R_1(\omega, p) = \frac{\Gamma(1-\Delta)\Gamma(\omega+\frac{1}{2}\Delta)\Gamma(p+\frac{1}{2}\Delta)}{\Gamma(\Delta-1)\Gamma(1+\omega-\frac{1}{2}\Delta)\Gamma(1+p-\frac{1}{2}\Delta)},$$
$$R_2(\omega, p) = \frac{\Gamma(1-\Delta)\Gamma(\frac{1}{2}\Delta-\omega)\Gamma(\frac{1}{2}\Delta-p)}{\Gamma(\Delta-1)\Gamma(1-\frac{1}{2}\Delta-\omega)\Gamma(1-\frac{1}{2}\Delta-p)}. \tag{3.45}$$

The response function for smooth solutions in global AdS$_3$ is therefore:

$$R^{\text{AdS}_3}(\omega, p) = \theta(\omega+p)R_1(\omega, p) + \theta(-(\omega+p))R_2(\omega, p). \tag{3.46}$$

These Green functions are "formal" in that they have poles that require careful interpretation. First, these functions are infinite when $\Delta \in \mathbb{Z}, \Delta \geq 2$. This happens because of the standard degeneration inherent in Frobenius' method: the non-normalizable solution is no longer a power series but contains a logarithmic term that multiplies the normalizable solution. We avoid this issue by taking $\Delta \notin \mathbb{Z}$, and then analytically continuing in $\Delta$ when possible. Second, the response function has also a tower of evenly spaced poles on the real axis corresponding to the frequencies and momenta where the regular solution (3.39) is normalizable. They correspond to the values where the arguments of the Gamma function in the numerators of (3.45) cross a negative integer value. To make the pole structure more manifest in the formulation of the response function, it is worth to rewrite $R^{\text{AdS}_3}$ in a more compact but non-analytic expression

$$R^{\text{AdS}_3}(\omega, p) = \frac{\Gamma(1-\Delta)}{\Gamma(\Delta-1)} \frac{\Gamma(\bar{\omega}+\frac{1}{2}\Delta)}{\Gamma(1+\bar{\omega}-\frac{1}{2}\Delta)} \frac{\Gamma(\bar{p}+\frac{1}{2}\Delta)}{\Gamma(1+\bar{p}-\frac{1}{2}\Delta)}, \tag{3.47}$$

where we have defined

$$\bar{\omega} \equiv \frac{|\omega+p| - |\omega-p|}{2}, \qquad \bar{p} \equiv \frac{|\omega+p| + |\omega-p|}{2}. \tag{3.48}$$

Since $\bar{p}$ is always positive, normalizable modes exist only in the range of frequency and momentum where $\bar{\omega} + \frac{1}{2}\Delta < 0$. These poles have the evenly spaced spectrum:

$$\left|\omega_j - p_j\right| - \left|\omega_j + p_j\right| - \Delta = 2j, \qquad j \in \mathbb{N}. \tag{3.49}$$

Moreover, by anticipating the comparison with the WKB answer, it is also useful to write down $R^{\text{AdS}_3}$ in the range where poles exist, $\bar{\omega} + \frac{1}{2}\Delta < 0$,

$$\begin{aligned}
R^{\text{AdS}_3}(\omega, p) = & \frac{\Gamma(1-\Delta)}{\Gamma(\Delta-1)} \frac{\Gamma(\frac{1}{2}\Delta - \bar{\omega})}{\Gamma(1-\bar{\omega}-\frac{1}{2}\Delta)} \frac{\Gamma(\bar{p}+\frac{1}{2}\Delta)}{\Gamma(1+\bar{p}-\frac{1}{2}\Delta)} \\
& \times \left[ -\sin(\pi\Delta)\tan\left(\frac{\pi}{2}(2\bar{\omega}+\Delta+1)\right) - \cos(\pi\Delta) \right],
\end{aligned} \tag{3.50}$$

where the coefficient in front of the bracket is smooth in this range and where the divergencies in the term containing the tangent give the spectrum.

The more subtle issue is how to integrate around all the poles in the response functions. Fortunately one can find a very thorough treatment of this issue in [40, 41, 43]. This involves careful combinations of analytic continuation, matching conditions and contour selection. Selecting different contours around poles either includes or excludes normalizable modes in the solution. We will discuss this further in Section 6, where we will use a particular contour prescription to relate the formal momentum-space Green functions derived here to a position-space Green function of interest.

### 3.3.2  WKB analysis for global AdS₃

Once again we use the metric (3.34) and the wave equation (3.35). We rescale the wavefunction and change the variable:

$$K(r) \equiv \frac{\Psi(r)}{\sqrt{r^2+1}}, \qquad x \equiv \ln(r), \quad x \in \mathbb{R}, \tag{3.51}$$

and the wave equation takes the Schrödinger form (2.4) with

$$V_{\text{AdS}}(x) \equiv \frac{e^{2x}}{e^{2x}+1}\left( (\Delta-1)^2 - \frac{(\omega-p)^2-1}{e^{2x}+1} + \frac{(\omega+p)^2}{e^{2x}} \right). \tag{3.52}$$

The boundaries were at $r = 0$ and at $r = \infty$ and these are now at $x = +\infty$ and $x = -\infty$, where the potential limits to $\mu^2 \equiv (\Delta-1)^2$ and $(\omega+p)^2$, respectively.

The potential is thus bounded by the value at infinity and the centrifugal barrier at $x = -\infty$ ($r = 0$). The two turning points, $x_-$ and $x_+$, and "classical region" $x_- < x < x_+$ only exists if the middle "energy term" in (3.52) is sufficiently negative. More precisely, to have classical turning points one must have

$$(\Delta-1+|\omega+p|)^2 < (\omega-p)^2-1 \quad \Leftrightarrow \quad (\Delta-1)^2+4\omega p+1 < -2(\Delta-1)|\omega+p|. \tag{3.53}$$

If this is satisfied, then the two turning points are at real values of $x$ and are given by:

$$e^{2x_\pm} = -\frac{1}{2(\Delta-1)^2}\left( ((\Delta-1)^2+4p\omega+1) \mp \sqrt{((\Delta-1)^2+4p\omega+1)^2-4(\Delta-1)^2(p+\omega)^2} \right). \tag{3.54}$$

The WKB integrals are elementary and we find:

$$\Theta = \frac{\pi}{2}\left[-\Delta + 1 - |\omega + p| + \sqrt{(\omega - p)^2 - 1}\right],\tag{3.55}$$

and

$$e^{-2I_+} = \left(\frac{e^{2x_+} - e^{2x_-}}{4}\right)^{\Delta - 1}\left(\frac{e^{x_+} - e^{x_-}}{e^{x_+} + e^{x_-}}\right)^{(\Delta - 1)\,e^{x_- + x_+}}$$
$$\times\left(\frac{\sqrt{e^{x_+} + 1} + \sqrt{e^{x_-} + 1}}{\sqrt{e^{x_+} + 1} - \sqrt{e^{x_-} + 1}}\right)^{(\Delta - 1)\,\sqrt{(e^{x_+} + 1)(e^{x_-} + 1)}}.\tag{3.56}$$

As one would expect, (3.53) implies that $\Theta > 0$.

The exact asymptotic problem is just the wave equation in the global-AdS$_3$ background and so $\Psi_{ex}^{\text{grow}}(x)$ and $\Psi_{ex}^{\text{dec}}(x)$ are given by (3.41) and (3.51):

$$\Psi_{ex}^{\text{grow}}(x) = e^{(\Delta - 2 + \omega - p)x}\left(1 + e^{2x}\right)^{(1 - \omega + p)/2}{}_2F_1\left(1 + p - \frac{\Delta}{2}, 1 - \omega - \frac{\Delta}{2}, 2 - \Delta, -e^{-2x}\right),$$
$$\Psi_{ex}^{\text{dec}}(x) = e^{(-\Delta + \omega - p)x}\left(1 + e^{2x}\right)^{(1 - \omega + p)/2}{}_2F_1\left(p + \frac{\Delta}{2}, -\omega + \frac{\Delta}{2}, \Delta, -e^{-2x}\right).\tag{3.57}$$

The WKB response function, $R_{WKB}^{\text{AdS}}$, can be then computed easily from the formula (2.17) with $\mathcal{A} = \tan\Theta$. This $\tan\Theta$ term is the only unbounded term and its poles, at $\Theta = \frac{\pi}{2}(2k + 1), k \in \mathbb{Z}$, correspond to the normalizable modes. This reproduces very accurately the spectrum dependence of the exact response function, $R^{\text{AdS}_3}$ (3.49). Indeed, in the range of parameters satisfying (3.53), we have that $\bar{\omega} < 0$ and (3.50) takes the form

$$R^{\text{AdS}_3} = g_1(\omega, p) + g_2(\omega, p)\,\tan\Theta_{ex},\tag{3.58}$$

where $g_1(\omega, p)$ and $g_2(\omega, p)$ are two non-diverging functions given in (3.50) and the exact spectrum function $\Theta_{ex}$ is

$$\Theta_{ex} = \frac{\pi}{2}\left[-\Delta - 1 - |\omega + p| + |\omega - p|\right].\tag{3.59}$$

We see that $\Theta \sim \Theta_{ex}$ as long as $|\omega - p| \gg 1$ which is guaranteed from (3.53) if we assume that $\Delta$ is large. Moreover, in Appendix D.2 we perform a numerical exploration, using the expressions of $I_+$, $\Psi_{ex}^{\text{grow}}$ and $\Psi_{ex}^{\text{dec}}$ above, that shows that

$$\frac{\sqrt{3}}{2}\,e^{-2I_+} - \frac{\Psi_{ex}^{\text{grow}}(x_+)}{\Psi_{ex}^{\text{dec}}(x_+)} \approx g_1(\omega, p), \qquad \frac{1}{2}\,e^{-2I_+} \approx g_2(\omega, p),\tag{3.60}$$

which implies

$$R_{WKB}^{\text{AdS}} \sim R^{\text{AdS}_3},\tag{3.61}$$

when $\Delta$ is large and when $|\omega - p|$ is also larger than $\Delta - 1 + |\omega + p|$ (for $\Delta \sim 5$ and $|\omega - p| - \Delta + 1 - |\omega + p| \sim 5$ the error of the WKB formula is already below 5%). Thus, our WKB technique provides an accurate approximation to describe response functions in global AdS$_3$.

# 4 The (1,0,n) superstrata

In this section, we briefly review all the features of the superstratum metric that are essential to our computation. More details can be found in the original papers [23–25][10]

---

[10]The existence of superstrata was conjectured in [46] and the equations underlying their solution were found in [47], inspired by the string emission calculations of [48].

## 4.1 The metric

Asymptotically AdS$_3$, $(1,0,n)$ superstrata are solutions of $\mathcal{N} = (0,1)$ six-dimensional supergravity coupled to two tensor multiplets. In the D1-D5-P frame of Type IIB string theory on T$^4$, they are bound states of $N_1$ D1-branes wrapped on a S$^1$ of the six-dimensional space, $N_5$ D5-branes wrapped on a S$^1 \times$T$^4$ and $N_P$ quanta of the momentum, $P$, along S$^1$. We denote $y$ as the coordinate around the S$^1$ with the identification

$$y = y + 2\pi R_y. \tag{4.1}$$

The six-dimensional metric with the null coordinates $u = y - t$ and $v = y + t$ is [23–25, 34]

$$
\begin{aligned}
ds_6^2 = \sqrt{Q_1 Q_5}\,\Lambda \Bigg[ &\frac{dr^2}{r^2 + a^2} - \frac{F_1(r)}{a^2(2a^2 + b^2)^2 F_2(r) R_y^2}\left(dv - \frac{a^2(a^4 + (2a^2 + b^2)r^2)}{F_1(r)}du\right)^2 \\
&+ \frac{a^2 r^2(r^2 + a^2)}{F_1(r) R_y^2}du^2 + d\theta^2 + \frac{1}{\Lambda^2}\sin^2\theta\left(d\varphi_1 + \frac{a^2}{(2a^2 + b^2)R_y}(du - dv)\right)^2 \\
&+ \frac{F_2(r)}{\Lambda^2}\cos^2\theta\left(d\varphi_2 - \frac{1}{(2a^2 + b^2)F_2(r)R_y}\left[a^2(du + dv) + b^2 F_0(r)dv\right]\right)^2 \Bigg],
\end{aligned}
\tag{4.2}
$$

where

$$
\begin{aligned}
F_0(r) &\equiv 1 - \frac{r^{2n}}{(r^2 + a^2)^n}, \qquad F_1(r) \equiv a^6 - b^2(2a^2 + b^2)r^2 F_0(r), \\
F_2(r) &\equiv 1 - \frac{a^2 b^2}{2a^2 + b^2}\frac{r^{2n}}{(r^2 + a^2)^{n+1}}, \\
\Lambda &\equiv \sqrt{1 - \frac{a^2 b^2}{2a^2 + b^2}\frac{r^{2n}}{(r^2 + a^2)^{n+1}}\sin^2\theta}.
\end{aligned}
\tag{4.3}
$$

This has the form of an S$^3$ fibration, parametrized by $(\theta, \varphi_1, \varphi_2)$, over a 2+1-dimensional base space, parameterized by $(t, u, v)$. The supergravity charges, $Q_1, Q_5, Q_P$, are related to the quantized charges, $N_1, N_5$ and $N_P$, via [23]:

$$Q_1 = \frac{(2\pi)^4 N_1 g_s \alpha'^3}{V_4}, \qquad Q_5 = N_5 g_s \alpha', \qquad Q_P = \mathcal{N}^{-1} N_P, \tag{4.4}$$

where $V_4$ is the volume of $T^4$ in the IIB compactification to six dimensions and $\mathcal{N}$ is defined via:

$$\mathcal{N} \equiv \frac{N_1 N_5 R_y^2}{Q_1 Q_5} = \frac{V_4 R_y^2}{(2\pi)^4 g_s^2 \alpha'^4} = \frac{V_4 R_y^2}{(2\pi)^4 \ell_{10}^8} = \frac{\text{Vol}(T^4) R_y^2}{\ell_{10}^8}. \tag{4.5}$$

Here $\ell_{10}$ is the ten-dimensional Planck length and one has $(2\pi)^7 g_s^2 \alpha'^4 = 16\pi G_{10} \equiv (2\pi)^7 \ell_{10}^8$. The quantity, $\text{Vol}(T^4) \equiv (2\pi)^{-4} V_4$, is sometimes used [49] as a "normalized volume" that is equal to 1 when the radii of the circles in the $T^4$ are equal to 1.

The angular momenta and the momentum of the solution are

$$J_L = J_R = \frac{1}{2}\mathcal{N} a^2, \qquad N_P = \frac{1}{2}\mathcal{N} n b^2. \tag{4.6}$$

In the CFT state dual to this geometry (1.1), the angular momenta are determined by the number of $|++\rangle_1$ strands, while the momentum is determined by $n$ and the number of strands

involved in the second factor. The partitioning of strands in the CFT, (1.2), has a supergravity counterpart as a regularity condition on the solution:

$$\frac{Q_1 Q_5}{R_y^2} = a^2 + \tfrac{1}{2} b^2. \tag{4.7}$$

The regularity condition also relates $a$ and $b$ to the quantized charges $N_1$, $N_5$ and $J_R$:

$$\frac{J_R}{N_1 N_5} = \frac{a^2}{2a^2 + b^2}. \tag{4.8}$$

We consider the solutions which have a long BTZ-like throat. This requires

$$a^2 \ll \{Q_1, Q_5, Q_P\} \qquad \Longleftrightarrow \qquad a \ll b. \tag{4.9}$$

The longest throat geometry is obtained by taking the minimum value of angular momentum $J_R$. In the dual CFT, this state has only one $|++\rangle_1$ strand, with quantized momentum $J_R = \tfrac{1}{2}$. Henceforth, we consider the solutions with $J_R = \tfrac{1}{2}$, and for such throats one has $N_1 N_5 = 1 + \frac{b^2}{2a^2} \sim \frac{b^2}{2a^2}$.

Thus, one can read off from the metric the two regions of the solutions which are depicted in Fig.1:

- The smooth cap geometry: for $r \lesssim \sqrt{n}\, a$, the geometry is an $S^3$ fibration over a global AdS$_3$ space with a highly red-shifted time and a non-zero angular momentum along $y$,

$$\begin{aligned}
ds_6^2 = \sqrt{Q_1 Q_5} \Bigg[ &\frac{dr^2}{r^2 + a^2} - (r^2 + a^2)\frac{1}{a^2 R_y^2} d\tau^2 + \frac{r^2}{a^2 R_y^2}\left(dy + \frac{b^2}{2a^2} d\tau\right)^2 \\
&+ d\theta^2 + \sin^2\theta \left(d\varphi_1 - \frac{d\tau}{R_y}\right)^2 + \cos^2\theta \left(d\varphi_2 - \frac{dy}{R_y} - \frac{b^2}{2a^2}\frac{d\tau}{R_y}\right)^2 \Bigg],
\end{aligned} \tag{4.10}$$

where $\tau = (1 + \frac{b^2}{2a^2})^{-1}\, t = (N_1 N_5)^{-1}\, t$. One can check that the local geometry at $r \sim 0$ is a smooth $S^3$ fibration over $\mathbb{R}^{1,2}$.

- The extremal-BTZ geometry: for $r \gtrsim \sqrt{n}\, a$, the geometry is $S^3$ fibration over extremal BTZ

$$ds_6^2 = \sqrt{Q_1 Q_5} \left[ \frac{d\rho^2}{\rho^2} - \rho^2 dt^2 + \rho^2 dy^2 + \frac{n}{R_y^2}(dy + dt)^2 + d\theta^2 + \sin^2\theta\, d\varphi_1^2 + \cos^2\theta\, d\varphi_2 \right], \tag{4.11}$$

where $\rho = \frac{r}{\sqrt{Q_1 Q_5}}$. The left and right temperatures of the BTZ region are

$$T_L = \frac{\sqrt{n}}{2\pi R_y}, \qquad T_R = 0. \tag{4.12}$$

The overall superstratum geometry is then the combination of a $S^3$ fibration over a BTZ geometry which ends with a highly red-shifted global-AdS$_3 \times S^3$ cap. It is natural to expect that wave perturbations will combine the features of both these geometries and we will show in the following sections that this is precisely what happens.

## 4.2 The massless scalar wave perturbation

The minimally coupled massless scalar wave equation

$$\frac{1}{\sqrt{-\det g}} \partial_M \left( \sqrt{-\det g}\, g^{MN} \partial_N \Phi \right) = 0 \,, \tag{4.13}$$

is separable in the $(1,0,n)$ superstratum and so we can expand the eigenfunctions as:

$$\Phi = K(r) S(\theta)\, e^{-i\left( \frac{\Omega}{R_y} u + \frac{P}{R_y} v + q_1 \varphi_1 + q_2 \varphi_2 \right)}. \tag{4.14}$$

Note that we are now labelling the modes by $(\Omega, P)$ as opposed to $(\omega, p)$. The reason for this will become apparent shortly. The wave equation reduces to one radial and one angular wave equation:

$$\frac{1}{r} \partial_r \left( r \left( r^2 + a^2 \right) \partial_r K(r) \right) + \left( \frac{a^2 (P - \Omega + q_1)^2}{r^2 + a^2} - \frac{a^2 (P + \Omega + q_2)^2}{r^2} \right) K(r) \tag{4.15}$$

$$+ \frac{b^2 \Omega \left( -2a^2 P + F_0(r)(2a^2 (\Omega - q_1) + b^2 \Omega) \right)}{a^2 (r^2 + a^2)} K(r) \;=\; m\, K(r),$$

$$\frac{1}{\sin\theta \cos\theta} \partial_\theta \left( \sin\theta \cos\theta\, \partial_\theta S(\theta) \right) - \left( \frac{q_1^2}{\sin^2\theta} + \frac{q_2^2}{\cos^2\theta} \right) S(\theta) \;=\; -m\, S(\theta), \tag{4.16}$$

where $m$ corresponds to the constant eigenvalue of the Laplacian operator along the $S^3$ which results in an effective mass in the three dimensional space. Thus, we define the conformal weight of the wave, $\Delta$, as $m \equiv \Delta(\Delta - 2)$. Without loss of generality, we consider that $\Delta > 1$. The angular equation (4.16) is solvable and there is only one branch of well-defined solutions [20]:

$$S(\theta) \propto (\sin\theta)^{|q_1|} (\cos\theta)^{|q_2|}\, {}_2F_1 \left( -\mathfrak{s}, 1 + \mathfrak{s} + |q_1| + |q_2|, |q_2| + 1, \cos^2\theta \right), \tag{4.17}$$

where $\mathfrak{s}$ is given by

$$\mathfrak{s} \equiv \frac{1}{2} \left( \Delta - 2 - |q_1| - |q_2| \right). \tag{4.18}$$

The solution is regular at $\cos^2\theta = 1$ if and only if $\mathfrak{s}$ is a non-negative integer. Consequently, the angular wave function is regular when

$$\Delta - 1 \geq 1 + |q_1| + |q_2|, \qquad q_1, q_2, \Delta, \mathfrak{s} \in \mathbb{Z}, \quad s \geq 0. \tag{4.19}$$

Moreover, the regularity of the modes at $r = 0$ requires [19]:

$$k \equiv \Omega + P \in \mathbb{Z}. \tag{4.20}$$

The radial wave equation is not solvable in general. In [20], the equation has been analytically solved for the $\sqrt{n}$ first normalizable modes when $n$ is taken to be large. These modes are essentially supported and determined by the AdS$_3$ cap geometry. Their discrete spectrum is given by[11]

$$\left| \left( \frac{b^2}{a^2} + 1 \right) \Omega_j - P_j - q_1 \right| - |\Omega_j + P_j + q_2| - \Delta \;=\; 2j, \qquad j \in \mathbb{N}, \tag{4.21}$$

---

[11]Note that this spectrum appears slightly different to that of [20] because our coordinates, $\{u, v\} = \{y - t, y + t\}$, differs from those of [20], $\{t, t + y\}$.

where the index $j$ is the mode number. This corresponds to the spectrum of an AdS$_3$ geometry computed in (3.49) with the additional quantum numbers $q_1$ and $q_2$ coming from vector-field reduction of the S$^3$ and the red-shifted frequency and momentum:

$$\omega = \left(1 + \frac{b^2}{2a^2}\right)\Omega, \qquad p = P - \frac{b^2}{2a^2}\Omega. \tag{4.22}$$

These expressions provide a valuable insight into the physics of the deep superstrata: their bound-state excitations (at least at large $n$) are simply those of a global AdS$_3$ geometry but with a red-shifted time (4.10). Since the superstratum also asymptotes to a (non-red-shifted) global AdS$_3$ geometry at infinity one needs to interpolate between these two limits in order to compute the response function. This requires non-normalizable modes and the high-energy normalizable modes. For that purpose, we will apply the WKB strategy detailed in Section 2.3 and 3 and we will find that the interpolation along the throat is provided by the BTZ response function.

## 4.3 WKB analysis

The first step is to reduce the wave equation to Schrödinger form, just as we did in previous sections. We first rescale the wave function and change variables:

$$K(r) = \frac{\Psi(r)}{\sqrt{r^2 + a^2}}, \qquad x = \ln\left(\frac{r}{a}\right), \quad x \in \mathbb{R}. \tag{4.23}$$

The radial wave equation gives

$$\frac{d^2}{dx^2}\Psi(x) - V(x)\Psi(x) = 0, \tag{4.24}$$

where $V(x)$ is given by:

$$V(x) \equiv \frac{e^{2x}}{e^{2x} + 1}\left[(\Delta - 1)^2 - \frac{1}{e^{2x} + 1}\left(B^2 - 1\right) + e^{-2x}A^2 + \frac{e^{-2x}}{(e^{-2x} + 1)^{n+1}}C\right], \tag{4.25}$$

and

$$A \equiv |\Omega + P + q_2|, \quad B \equiv \left|\Omega\left(1 + \frac{b^2}{a^2}\right) - P - q_1\right|, \quad C \equiv \frac{b^2}{a^2}\Omega\left(2(\Omega - q_1) + \frac{b^2}{a^2}\Omega\right). \tag{4.26}$$

The general form of such potential is depicted in Fig.4. In this case we have $\mu = (\Delta - 1)$ and $V(-\infty) = A^2$.

We define $x_\pm$ as the two turning points of the potential[12], $V(x_\pm) = 0$. Zone II is the classically allowed region where $V(x) < 0$ and zone I and III are the classically forbidden regions with positive potential.

We compute the physical wave function $\Psi(x)$ at leading order in each zone by applying the WKB approximation as we did for a scalar wave in global AdS$_3$ in section 3.1.2. We impose the regularity of the solution at $x = -\infty$ ($r = 0$) and we apply the junction rules with Airy functions to connect the three parts of the solution at the turning points. Therefore, the WKB

---

[12]If $A = 0$, then $x_- = -\infty$. This does not compromise our discussion in any way since $|V(x)|^{1/2}$ remains integrable at this location.

approximation gives

$$
\Psi_{\text{phys}}(x) = \begin{cases}
\dfrac{1}{|V(x)|^{\frac{1}{4}}} \, \exp\left(-\int_x^{x_-} |V(z)|^{\frac{1}{2}} dz\right), & x < x_-, \\[3mm]
\dfrac{1}{|V(x)|^{\frac{1}{4}}} \, \cos\left(\int_{x_-}^x |V(z)|^{\frac{1}{2}} dz + \frac{\pi}{4}\right), & x_- < x < x_+, \\[3mm]
\dfrac{2\cos\Theta}{|V(x)|^{\frac{1}{4}}} \left[ \exp\left(\int_{x_+}^x |V(z)|^{\frac{1}{2}} dz\right) + \frac{\tan\Theta}{2} \exp\left(-\int_{x_+}^x |V(z)|^{\frac{1}{2}} dz\right) \right], & x > x_+,
\end{cases}
\tag{4.27}
$$

where the WKB integral $\Theta$ is defined in (2.20). The validity of the WKB approximation is guaranteed when the condition given in (2.5) is satisfied. This imposes

$$
|\Omega| \gtrsim \frac{2a^2}{b^2} \quad \text{and} \quad \Delta \gtrsim 1. \tag{4.28}
$$

From the discrete spectrum of the modes (4.21), we expect that the WKB approximation will not be accurate for the first few modes. We will see how to deal this issue in the next sections. Moreover, the integral of $|V|^{\frac{1}{2}}$ cannot be performed analytically and one needs to divide our computation into different ranges of frequencies $\Omega$ and $k = \Omega + P$ to approximate its value.

To apply the WKB technique developed in Section 3 one needs to have a good understanding of the behavior of the superstratum potential, in particular to identify the interior and asymptotic potentials depending of the range of values of $\Omega$ and $P$. From now on, we will assume for simplicity that the wave perturbations are independent of $\varphi_1$ and $\varphi_2$ by setting $q_1 = q_2 = 0$. The inclusion of non-zero values of $q_1$ and $q_2$ is fairly straightforward. Moreover, we are interested in superstratum backgrounds with $1 \ll \sqrt{n} \ll \frac{b}{a} \sim \sqrt{N_1 N_5 / J_R}$. This assumption is not necessary for the computation as it was in [20]. It simply allows the geometry to have a large cap region $(0 < r < \sqrt{n}a)$ which can support the first few modes.

First, we observe that the term proportional to $C$ in (4.25) is the core difference between the superstratum potential and that of global-AdS$_3$, (3.52). This term is irrelevant as long as $e^{2x} \lesssim n$, or $r \lesssim \sqrt{n}\,a$, which exactly corresponds to the validity of the cap geometry (4.10). Above this transition, there are various possibilities that depend on the values of $\Omega$ and $P$. Before detailing those possibilities, we define three limits of potential

$$
V^{\text{cap}}(x) \equiv \frac{e^{2x}}{e^{2x}+1} \left[ (\Delta-1)^2 - \frac{1}{e^{2x}+1} \left( \left( \Omega\left(1+\frac{b^2}{a^2}\right) - P \right)^2 - 1 \right) + e^{-2x}(\Omega+P)^2 \right],
$$

$$
V^{\text{BTZ}}(x) \equiv (\Delta-1)^2 + \frac{2b^2 P\Omega}{a^2} e^{-2x} - \frac{b^4 n\Omega^2}{a^4} e^{-4x},
$$

$$
V^{\text{I-B}}(x) \equiv (\Delta-1)^2 + \frac{2b^2 P\Omega - a^2\,\Delta(\Delta-2)}{a^2} e^{-2x} - \frac{b^4 n\Omega^2}{a^4} e^{-4x}.
\tag{4.29}
$$

The potential, $V^{\text{cap}}(x)$, is obtained by dropping $C$, and so:

$$
V(x) \sim V^{\text{cap}}(x), \qquad e^{2x} \lesssim n. \tag{4.30}
$$

In the range $|\Omega| \lesssim \frac{2\sqrt{n}a^2}{b^2}$, we will show explicitly in Section 5.2 that the superstratum potential will be well approximated by $V^{\text{cap}}(x)$ also when $e^{2x} \gtrsim n$. Intuitively, this is comes from the fact that the bump induced by the term proportional to $C$ in (4.25) is subleading compared to the other terms in this range of frequency.

In the range $|\Omega| \gtrsim \frac{2\sqrt{n}\,a^2}{b^2}$, we can perform an expansion of the superstratum potential for $e^{2x} \gtrsim n$ which gives a BTZ-type of potential (3.27)

$$V(x) \sim (\Delta-1)^2 + e^{-2x}[A^2 - B^2 + 1 + C - (\Delta-1)^2] - e^{-4x}[-B^2 + 1 - (n+1)C - (\Delta-1)^2]\,. \quad (4.31)$$

By carefully analyzing which terms in the coefficients in front of $e^{-2x}$ and $e^{-4x}$ are leading or subleading at large $b/a$ and $n$, we can show that

$$
\begin{aligned}
V(x) &\sim V^{\text{BTZ}}(x), &&\quad n \lesssim e^{2x} \text{ and } k = \Omega + P \nsim 0\,, \\
V(x) &\sim V^{\text{I-B}}(x), &&\quad n \lesssim e^{2x} \text{ and } k \sim 0\,.
\end{aligned}
\qquad (4.32)
$$

The two first potentials in (4.29) can be directly derived by computing the wave equations in the smooth cap region (4.10) and in the extremal-BTZ region (4.11). Thus, $V^{\text{cap}}(x)$ is the scalar potential in a global $\text{AdS}_3$ background given in (3.52) with red-shifted momentum and frequency, $\omega = \left(1 + \frac{b^2}{2a^2}\right)\Omega$ and $p = P - \frac{b^2}{2a^2}\Omega$. Similarly, $V^{\text{BTZ}}(x)$ matches the scalar potential in extremal BTZ (3.27) with the same red-shifted frequency. However, $V^{\text{I-B}}(x)$ (where "I-B" stands for "intermediate BTZ" and not for the initials of one of the authors) does not correspond to a potential of a specific region in the superstratum background. Thus, according to (4.32), the wave perturbation with $k \sim 0$ will feel a potential which differs from the expectation of the BTZ region of the superstratum background.

## 5 Response function for (1,0,n) superstrata

We are interested in computing the response function in (1,0,n)-superstratum solutions with $1 \ll \sqrt{n} \ll \frac{b}{a} \sim \sqrt{N_1 N_5 / J_R}$ and with $q_1 = q_2 = 0$. We therefore apply the hybrid WKB computation of Section 2.3 to a scalar field in the superstratum background detailed in Section 4.

### 5.1 Summary of results

We will show that the response function in momentum space has four distinct regimes depending on $\Omega$ and $k = \Omega + P$ depicted in Fig.5:

- **The cap regime.** For small $\Omega$, $|\Omega| \lesssim \frac{2\sqrt{n}\,a^2}{b^2}$, and at any $k$, the turning points $x_\pm$ are both located inside the cap (4.30), so the wave only sees the cap geometry. The response function is given by the response function in Global $\text{AdS}_3$, $R^{\text{AdS}_3}$ (3.46), with highly red-shifted frequency and momentum.

- **The BTZ regime.** For large $\Omega$, $|\Omega| \gtrsim \frac{2\sqrt{n}\,a^2}{b^2}$, and for $k \nsim 0$, the outermost turning point is no longer in the cap region and the wave starts to explore the extremal-BTZ region of the geometry. The response function in momentum space is a deformation by $\tan\Theta$ of the response function of extremal-BTZ, $R^{\text{BTZ}}$ (3.25), as detailed in section 3.2.

- **The intermediate BTZ regime.** For large $\Omega$, $|\Omega| \gtrsim \frac{2\sqrt{n}\,a^2}{b^2}$, but for $k \sim 0$, the wave differs from the extremal-BTZ expectation. The response function in momentum space is similar to the one in the BTZ regime but with a "rescaled" momentum $\bar{P}$ which differs from $P$ only when $|\Omega| \lesssim \frac{a}{b}$:

$$\bar{P} \equiv P - \frac{a^2}{2b^2\,\Omega}\Delta(\Delta-2)\,. \qquad (5.1)$$

- **The centrifugal-barrier regime.** When $k\Omega > 0$ and $|k| > \frac{b^2}{a^2}|\Omega|$, the centrifugal barrier at the origin of the space is very high, nothing can penetrate the throat and the potential of

the scalar perturbation is always positive. Correspondingly, there are no bound states. When $|\Omega| \lesssim \frac{2\sqrt{n}a^2}{b^2}$, this effect is well captured by the response function in global AdS$_3$. However, when $|\Omega| \gtrsim \frac{2\sqrt{n}a^2}{b^2}$, the wave is strongly repulsed outside the BTZ throat which is not captured by the BTZ response function. In this specific regime, the response function cannot be computed using the WKB hybrid method. We denote as $R^{bar}(\Omega, P)$ the response function in this regime.

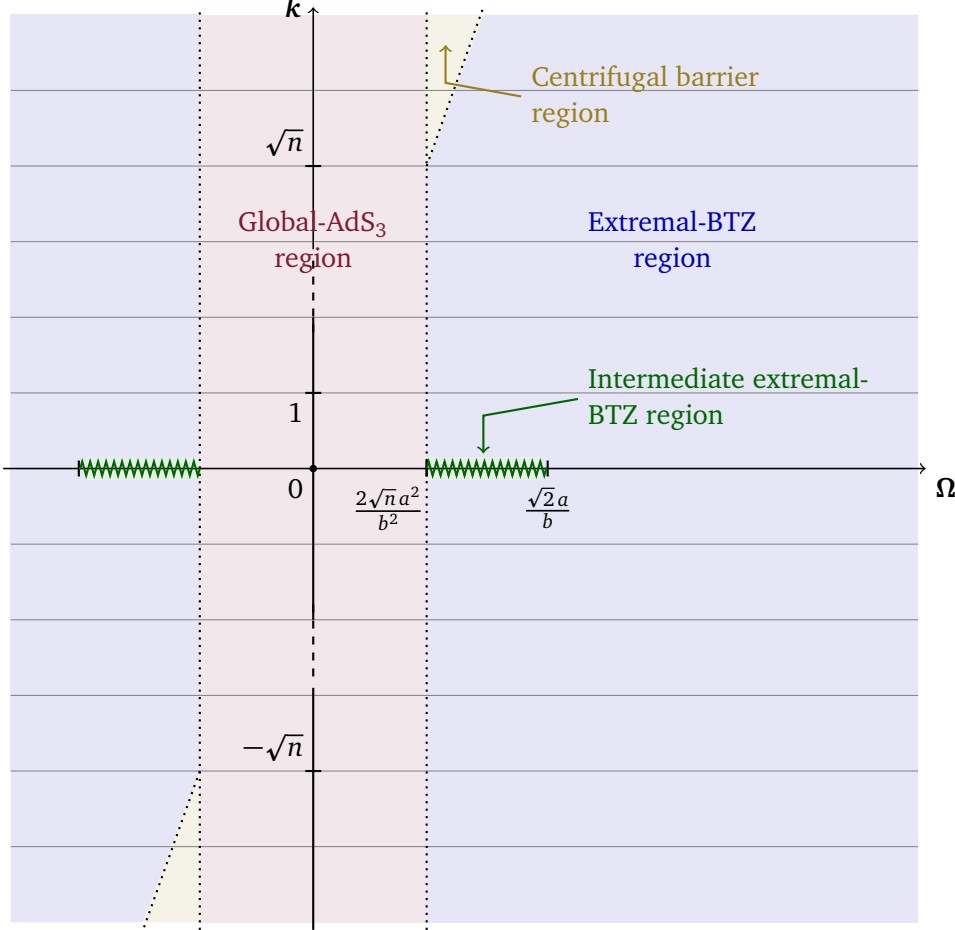

Figure 5: The four regimes for the (1,0,n)-superstratum response function as a function of $\Omega$ and $k = \Omega + P$. For very small $\Omega$, the wave goes far inside the throat and is essentially determined by the global-AdS$_3$ geometry at the cap. For larger frequencies, the wave starts to explore the extremal-BTZ region. The value of $k$ changes only the centrifugal barrier felt very close to the origin. However, around $k \sim 0$ there is a third domain called "intermediate" extremal-BTZ (discussed in detail in subsection 5.4).

We will show that in these four regimes the superstratum response function is determined by the following combinations of the global-AdS$_3$ response function, $R^{\text{AdS}_3}(\omega, p)$ (3.46), and

the extremal-BTZ response function, $R^{\text{BTZ}}(\omega, p)$ (3.25):

$$R^{(1,0,n)}(\Omega, P) \tag{5.2}$$

$$\sim \begin{cases} R^{\text{AdS}_3}\left(\frac{b^2\Omega}{2a^2}, P + (1 - \frac{b^2}{2a^2})\Omega\right) & , \quad |\Omega| \lesssim \frac{2\sqrt{n}\,a^2}{b^2}, \\ \text{Re}\left[R^{\text{BTZ}}\left(\frac{b^2\Omega}{2a^2}, \bar{P}\right)\right] + \text{sign}(\Omega)\tan\Theta\,\text{Im}\left[R^{\text{BTZ}}\left(\frac{b^2\Omega}{2a^2}, \bar{P}\right)\right] & , \quad \frac{2\sqrt{n}\,a^2}{b^2} \lesssim |\Omega|,\ P \sim \Omega, \\ \text{Re}\left[R^{\text{BTZ}}\left(\frac{b^2\Omega}{2a^2}, P\right)\right] + \text{sign}(\Omega)\tan\Theta\,\text{Im}\left[R^{\text{BTZ}}\left(\frac{b^2\Omega}{2a^2}, P\right)\right] & , \quad \frac{2\sqrt{n}\,a^2}{b^2} \lesssim |\Omega|,\ P \not\sim \Omega, \\ R^{\text{bar}}(\Omega, P) & , \frac{2\sqrt{n}\,a^2}{b^2} \lesssim |\Omega|,\ \Omega k > 0, \quad |k| > \frac{b^2|\Omega|}{a^2}. \end{cases}$$

The $\tan\Theta$ term which captures the microstate structure of the geometry when $\frac{2\sqrt{n}\,a^2}{b^2} \lesssim |\Omega|$ is given by the IR cap geometry. We will show that

$$\Theta \sim \frac{\pi}{2}\left[|\gamma\Omega - P| - \delta|\Omega + P| - \eta\right], \qquad \frac{2\sqrt{n}\,a^2}{b^2} \lesssim |\Omega|, \tag{5.3}$$

with $\gamma \sim b^2/a^2$, $\delta \sim 1$ and $\eta \sim |\Delta - 1|$. This is very close to the same spectrum function which is included inside $R^{\text{AdS}_3}\left(\frac{2a^2\Omega}{b^2}, P + (1 - \frac{b^2}{2a^2})\Omega\right)$ when $|\Omega| \lesssim \frac{2\sqrt{n}\,a^2}{b^2}$ and which can be derived from (3.59)

$$\Theta \sim \frac{\pi}{2}\left[\left|\left(\frac{b^2}{a^2} + 1\right)\Omega - P\right| - |\Omega + P| - |\Delta - 1|\right], \qquad |\Omega| \lesssim \frac{2\sqrt{n}\,a^2}{b^2}. \tag{5.4}$$

In the next subsections, we will show how to obtain the first three lines of (5.2) using the hybrid WKB technique detailed in Section 2.3, and from (2.17) in particular. The only quantity which will not be computable with WKB is $R^{\text{bar}}(\Omega, P)$.

The expression for the response function of the superstratum, (5.2), strongly reflects the intuitive physical picture of the superstratum. There is a global AdS$_3$ cap at a very high red-shift relative to infinity. Thus, the modes that explore the bottom of the throat produce a response function that looks like that of global AdS$_3$ but with highly blue-shifted modes relative to the frequencies at infinity. The AdS$_3$ cap is connected to the asymptotic region at infinity by a long BTZ throat, and modes that explore this throat have a response function that is modulated by the BTZ response function.

In this way one will see what appears to be the "absorptive behavior" of the BTZ throat over short and intermediate time scales, while over long time-scales, of order $N_1 N_5$, one will see strong echoes from the cap, in agreement with unitarity requirements. Because of the explicit appearance of the BTZ response function, the superstratum will also contain information about the left-moving temperature of the extremal BTZ metric. This temperature governs the decay of the response function over time-scales smaller than the echo return-time, $N_1 N_5$. We will discuss the position-space response functions in more detail in Section 6.

The remaining part, $R^{\text{bar}}(\Omega, P)$, of the response function is perhaps rather less interesting because the probe has so much angular momentum, $k$, that it simply cannot penetrate the throat of the superstratum.

Finally we note that the response function has poles in the real $(\Omega, P)$-plane. They appear through $R^{\text{AdS}_3}$ and through $\tan\Theta$ in (5.2) and simply represent the bound states of the cap. Indeed, because of (5.3), these bound states are almost identical to those of a global AdS$_3$. As explained in the Introduction, the superstratum cannot have quasi-normal modes. On the other hand, the BTZ response function, (3.25), has poles along the imaginary $P$-axis and these do indeed correspond to quasi-normal modes. The important point is that even though (5.2) involves the BTZ response function, the poles are specifically excluded because the approximation we made is only valid in the *real* $(\Omega, P)$-plane. Thus the BTZ response function merely modulates the amplitude of the response function and does not (and cannot) introduce imaginary poles.

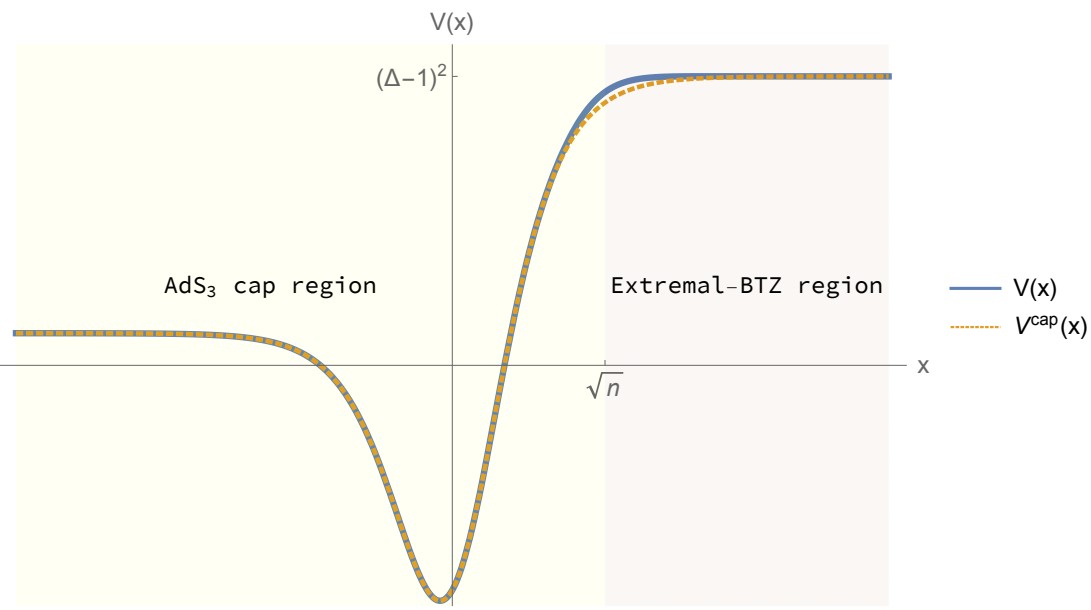

Figure 6: *The cap regime:* The superstratum potential $V(x)$ and the approximated potential $V^{\text{cap}}(x)$ when $|\Omega| \lesssim \frac{2\sqrt{n}\,a^2}{b^2}$.

## 5.2 The cap regime

We consider the range of frequencies $|\Omega| \lesssim \frac{2\sqrt{n}\,a^2}{b^2}$. The graph in Fig.6 shows the superstratum potential $V(x)$ and the cap potential $V^{\text{cap}}(x)$ as a function of $x$ for $\Omega = \frac{\sqrt{n}\,a^2}{2b^2}$, $\Delta = 5$ and $k = 1$. From the figure, the potentials look very close to each other. More rigorously, we have

$$\delta V(x) \equiv \left| \frac{V(x) - V_{\text{cap}}(x)}{V_{\text{cap}}(x)} \right| \lesssim \left( \frac{b^2\,\Omega}{a^2(\Delta - 1)} \right)^2 \left( 1 + e^{-2x} \right)^{-n-2} e^{-2x}, \quad x \nsim x_{\pm}. \tag{5.5}$$

Consequently, for $|\Omega| \lesssim \frac{2\sqrt{n}\,a^2}{b^2}$ and $\Delta$ large, we have $V(x) \sim V_{\text{cap}}(x)$ for any $x$. Moreover, $V_{\text{cap}}(x)$ is simply the potential of a scalar wave in a red-shifted global AdS$_3$ geometry as explained in the previous section. Thus, one can reproduce the results of the Sections 3.3.1 and 3.3, where we have computed the WKB response function in a global AdS$_3$ background and where we have compared it to the exact function. The WKB computation can be applied only if the potential has classical turning points which is guaranteed for global AdS$_3$ when (3.53) is satisfied. For our red-shifted AdS$_3$ cap, this translates into the condition that $k\,\Omega > 0$ and $|k| > \frac{b^2\,|\Omega|}{a^2}$. Moreover, the WKB approximation has been shown to be accurate for values of $\Delta$ of order at least slightly higher than one and for $|\omega - p| = |(\frac{2b^2}{a^2} - 1)\Omega - P| \gg 1$. Thus, under all those assumptions and for $\frac{a^2}{b^2} \ll |\Omega| \lesssim \frac{2\sqrt{n}\,a^2}{b^2}$, the (1,0,n)-superstratum response function is given by

$$R^{(1,0,n)}(\Omega, P) \sim R^{\text{AdS}_3}\left( \frac{2a^2\,\Omega}{b^2}, P + \left( 1 - \frac{b^2}{2a^2} \right)\Omega \right), \tag{5.6}$$

where $R^{\text{AdS}_3}(\omega, p)$ is given by (3.46).

One can actually relax the assumptions of validity of this expression. According to (5.5), one should have $|\Omega| \lesssim \frac{2\sqrt{n}\,a^2}{b^2}$ and $\Delta$ large so as to have $\delta V(x)$ small. The additional requirement that $|(\frac{2b^2}{a^2} - 1)\Omega - P| \gg 1$ is only necessary for the WKB approximation. Indeed, for $|(\frac{2b^2}{a^2} - 1)\Omega - P| \lesssim 1$, the (1,0,n)-superstratum potential is even more closely approximated by the cap potential according to (5.5), and the identification (5.6) is thus even more accurate.

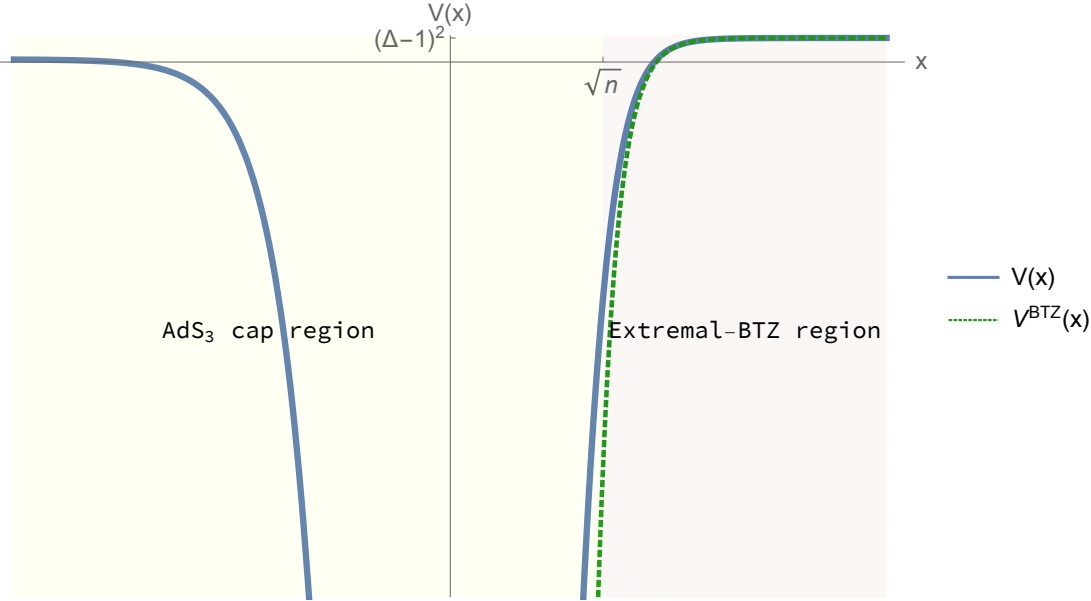

Figure 7: *The extremal-BTZ regime:* the superstratum potential $V(x)$ and the approximated potential $V^{\mathrm{BTZ}}(x)$ when $\frac{2\sqrt{n}a^2}{b^2} \lesssim |\Omega|$ and $P \nsim \Omega$.

Thus we have established the first line of (5.2).

For small frequencies, the response function is determined by the red-shifted global $\mathrm{AdS}_3$ geometry at the cap. The spectrum of the normalizable modes is given by the real poles in (5.6) which corresponds to the spectrum of a highly red-shifted global $\mathrm{AdS}_3$. This spectrum gives exactly the evenly spaced spectrum initially computed in [20] given in (4.21). This is valid as long as $|\Omega| \lesssim \frac{2\sqrt{n}a^2}{b^2}$, which corresponds to the $\sqrt{n}$ first normalizable modes. For higher frequencies, the scalar wave starts to explore the BTZ region of the geometry and then the response function will differ from the global-$\mathrm{AdS}_3$ expectation as we will discuss in the next sections.

## 5.3 The extremal-BTZ regime

We assume that $\frac{2\sqrt{n}a^2}{b^2} \lesssim |\Omega|$ and that $P \nsim \Omega$. As depicted in Fig.7, one can show that $x_+$ is therefore in the extremal-BTZ region, $x_+ \gtrsim \sqrt{n}$. One can then use all the machinery developed in Section 3 by considering $V_{\mathrm{BTZ}}(x)$ as the asymptotic potential "$V_{\mathrm{asymp}}(x)$". Moreover, $V_{\mathrm{BTZ}}(x)$ corresponds to the potential one can compute in an extremal-BTZ black hole (3.18) at the left-moving temperature $T_L = \frac{\sqrt{n}}{2\pi}$ and with a highly red-shifted coordinate $u$. One can apply the computation in Section 3.2.2 with $\omega = \frac{b^2}{2a^2}\Omega \sim N_1 N_5 \,\Omega/J_R$, $p = P$, $r_H = \sqrt{n}$ and $\mathcal{A} = \tan\Theta$ where $\Theta$ is defined in (2.20). The final result for the response function (2.22) gives

$$R^{(1,0,n)}(\Omega, P) \sim \mathrm{Re}\left[R^{\mathrm{BTZ}}\left(\frac{b^2\,\Omega}{2a^2}, P\right)\right] + \mathrm{sign}(\Omega)\tan\Theta \, \mathrm{Im}\left[R^{\mathrm{BTZ}}\left(\frac{b^2\,\Omega}{2a^2}, P\right)\right], \qquad (5.7)$$

where $R^{\mathrm{BTZ}}(\omega, p)$ is the response function in momentum space of a scalar field in an extremal-BTZ black hole (3.25).

This expression shows how the superstratum response function matches the overall shape of the BTZ response function but with a deformation term, $\tan\Theta$. In the cap regime, $|\Omega| \lesssim \frac{2\sqrt{n}a^2}{b^2}$, $\Theta$ is given by (5.4). For $\frac{2\sqrt{n}a^2}{b^2} \lesssim |\Omega|$, finding an analytic expression for $\Theta$ is a harder task since

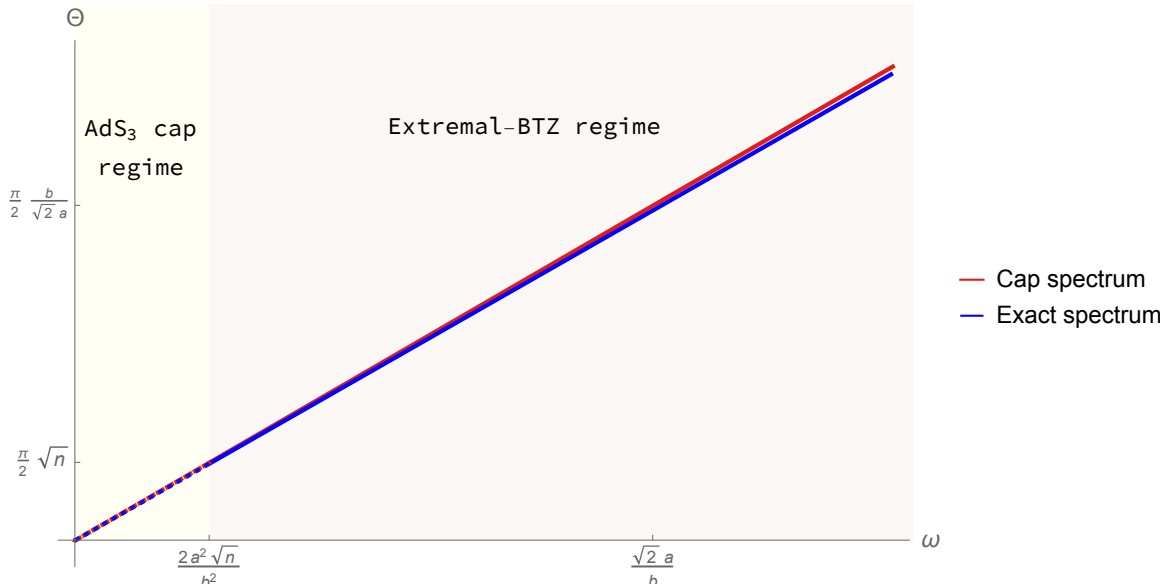

Figure 8: The spectrum function determined by the WKB function $\Theta$ (2.20) as a function of $\Omega$. The graph in blue corresponds to the numerical value of $\Theta$ with the full (1,0,n)-superstratum potential (4.25). The graph in red corresponds to the cap spectrum given by (5.4). The graphs have been computed for $\Delta = 3$, $k = 1$, $a = 1$, $b = 2 \cdot 10^3$ and $n = 750$ but different parameters give the same behaviors.

the superstratum potential is no longer well approximated by $V_{\text{cap}}(x)$ or by any other explicitly integrable potential between the two turning point. We therefore performed a numerical computation of $\Theta$. Surprisingly, $\Theta$ is almost linear as a function of $\Omega$ and $P$ as in the global-AdS$_3$ regime (see Fig.8 as an illustration). Moreover, this behavior is similar to (5.4) with slightly different coefficients

$$\Theta \sim \frac{\pi}{2}\left[\,|\gamma\,\Omega - P| - \delta\,|\Omega + P| - \eta\,\right], \tag{5.8}$$

with $\gamma \sim \frac{b^2}{a^2}$, $\delta \sim 1$ and $\eta \sim |\Delta - 1|$. The spectrum of the (1,0,n) superstratum states is then very close to a linear function of $\Omega$ and $P$, even when the modes start to explore the BTZ throat. We can expect from those evenly spaced poles in the spectrum that the response function in position space will be periodic and not sporadic. This difference comes from the highly coherent nature of the (1,0,n) superstrata and from the separability of the wave equation in these geometries, as we will discuss in more detail in Section 7.

## 5.4 The intermediate extremal-BTZ regime

When $P \sim \Omega$ and $\frac{2\sqrt{n}\,a^2}{b^2} \lesssim \Omega$, the superstratum potential is not well approximated anymore by the BTZ potential since the term $a^2\Delta(\Delta - 2)$ is not subleading compared to $2b^2 P\Omega$ as long as $|\Omega| \lesssim \frac{a}{b}$. We must then consider the "intermediate" extremal-BTZ potential $V^{\text{I-B}}(x)$ defined in (4.29). We use "intermediate" since a rescaling $\bar{P} \equiv P + \frac{a^2}{2b^2\Omega}\Delta(\Delta - 2)$ converts $V^{\text{I-B}}(x)$ to the BTZ potential, $V^{\text{BTZ}}(x)$, with $\bar{P}$ instead of $P$. Thus, we can extrapolate easily the WKB computation of the previous section. For $\frac{2\sqrt{n}\,a^2}{b^2} \lesssim |\Omega|$ with $P \sim \Omega$, the response function is

$$R^{(1,0,n)}(\Omega, P) \sim \text{Re}\left[R^{\text{BTZ}}\left(\frac{b^2\,\Omega}{2a^2}, \bar{P}\right)\right] + \text{sign}(\Omega)\tan\Theta\,\text{Im}\left[R^{\text{BTZ}}\left(\frac{b^2\,\Omega}{2a^2}, \bar{P}\right)\right], \tag{5.9}$$

where $\Theta$ is still of the form of (5.8) in this regime of parameters. It is only when $\frac{a}{b} \lesssim \Omega$ that $\bar{P} \sim P$ and that superstratum response function can fully match the BTZ expectation as in the

previous section. Thus, our computation indicates an intermediate scale in momentum space, $\frac{a}{b} \sim \sqrt{N_1 N_5}$, from where the superstratum response function starts to slightly differ from the BTZ one. This scale in position space is $t \sim \frac{b}{a} R_y$. The superstratum response function will slightly differ from the BTZ response function at $t \sim \frac{b}{a} R_y$ which is parametrically smaller than the scale where the first echo from the cap happens $t \sim \frac{b^2}{a^2} R_y \sim N_1 N_5 R_y$.

A similar slight discrepancy at an intermediate scale has been found in [19, 50], who studied the geodesic motion of probe particles in capped BTZ backgrounds[13]: a probe dropped from infinity will experience Planck-scale tidal stresses at a distance $r \lesssim \sqrt{ab}$, which is way above the region where the cap is. However, a similar computation in an extremal-BTZ geometry gives a constant and small tidal stress. The radial scale for a classical particle and our frequency scale for a scalar wave can be related by the usual lore that a classical particle lies where the potential of the wave equals the energy (on-shell condition). This corresponds to the radial distance where the potential vanishes in our convention and to the classical turning points. In the present regime of parameters, the turning point, $x_+$ is given by the intermediate-BTZ potential. From the result (3.28), the turning point in the $x$ coordinate, $e^x = r/a$, is

$$x_+ = \frac{1}{2} \ln\left[ \frac{b^2}{a^2 (\Delta-1)^2} \left( |\Omega| \sqrt{\bar{P}^2 + n(\Delta-1)^2} - \Omega \bar{P} \right) \right]. \tag{5.10}$$

Thus, for $P \sim \Omega \sim \frac{a}{b}$ we have

$$x_+ \sim \frac{1}{2} \ln\left[ \frac{b}{a} \right] \qquad \Rightarrow \qquad r_+ \sim \sqrt{ab}. \tag{5.11}$$

Hence, the frequency scale where the scalar waves start to differ from the extremal-BTZ expectation matches the radial scale where the tidal stresses of classical infalling probes reach the Planck scale.

## 5.5 The centrifugal-barrier regime

The WKB hybrid computation requires the existence of at least one turning point. Our attempts to extend the technique to strictly positive potentials have either failed or been inaccurate. When the potential is always positive, there is no classically allowed region and the scalar waves are either growing or decaying. The physical waves which are smooth at $x \sim -\infty$ are then necessary growing at the boundary. As explained in Section 2.2.1, the WKB approximation is inefficient to extract the vev part from an only growing wave function. However, this does not mean that the response function is zero. As an illustration, the scalar wave equation in global-AdS$_3$ has a range of frequencies and momenta where the potential is strictly positive (3.53). However, from a straightforward computation, the response function, (3.46), is not zero or even close to zero in this regime. Nevertheless, the momentum space response function has no poles in this regime.

The superstratum potential, (4.25), has no classical turning point when the centrifugal barrier at the origin given by $A$ is larger than the penetration parameter given by $B$. A straightforward analysis shows that the potential has a large centrifugal barrier and is strictly positive when

$$k\Omega > 0 \quad \text{and} \quad |k| > \frac{b^2 |\Omega|}{a^2}.$$

For small values of $\Omega$, we have shown that the superstratum potential is well-approximated by the global-AdS$_3$ cap potential. Thus, the centrifugal-barrier regime is taken into account

---

[13]For other interesting work on probing microstate geometries see [51–53].

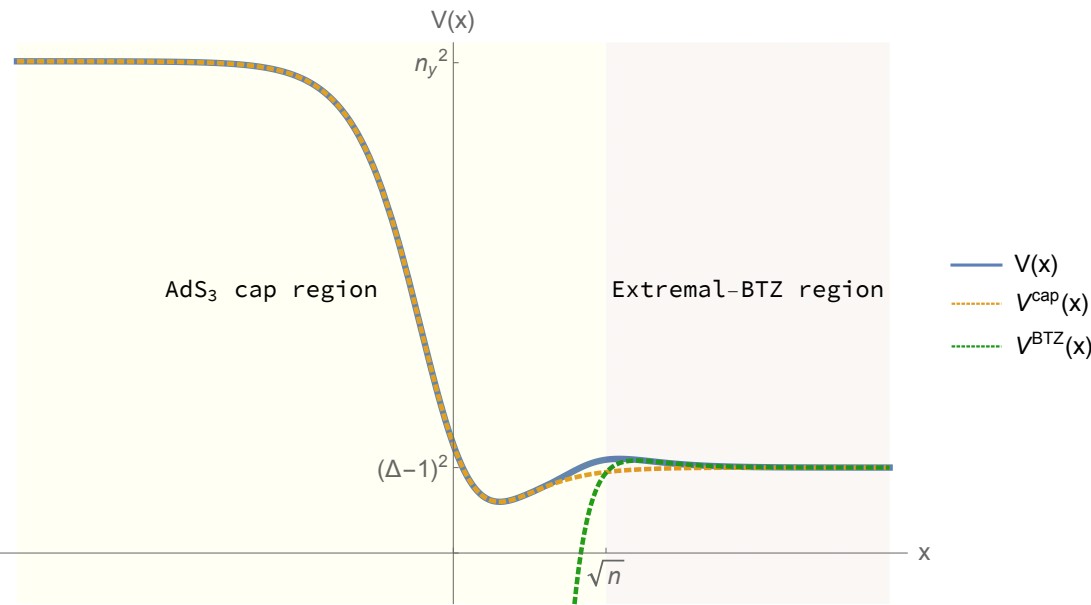

Figure 9: *The centrifugal-barrier regime:* the superstratum potential $V(x)$ and the approximated potential $V^{\text{BTZ}}(x)$ when $\frac{2\sqrt{n}\,a^2}{b^2} \lesssim |\Omega|$, $k\,\Omega > 0$ and $|k| > \frac{b^2\,|\Omega|}{a^2}$.

by the identification of the superstratum response function to the AdS$_3$ cap response function (5.6).

The issue occurs for large $\Omega$, $\frac{2\sqrt{n}\,a^2}{b^2} \lesssim |\Omega|$, in the BTZ regime. The superstratum potential is no longer well-approximated by the cap potential and one cannot apply our WKB hybrid method to extract the response function from the asymptotic BTZ potential. As an illustration, Figure 9 gives the behavior of the potentials in this regime.

However, we have strong intuition that the response function in this regime does not have a significant impact on the physics of wave perturbations in superstratum backgrounds for two reasons: First, in momentum space, the relevant information is contained in the pole structure of the response function, particularly in their locations and in their envelopes. The centrifugal-barrier regime is essentially characterized by the absence of normalizable modes. Thus, it will only reduce the expected zone where normalizable modes exist. Second, this regime corresponds to very high momentum $k$ and has a small size in the two-dimensional momentum space given by $k$ and $\Omega$ (Fig.5). Indeed, it is delimited by $|\Omega| \gtrsim \frac{2\sqrt{n}\,a^2}{b^2} \sim \frac{\sqrt{n}J_R}{N_1 N_5}$ and by the sharp line $|k| > \frac{b^2\,|\Omega|}{a^2} \sim \frac{N_1 N_5}{J_R}|\Omega|$.

Consequently, we neglect the response function in this regime and have good hope that it does not compromise the overall understanding of the response function in (1,0,n) superstratum.

In the next section, we will discuss the Fourier transform of the response function to position-space.

## 6 Position-space Green functions

We now return to our original goal of assessing to what extent the $(1, 0, n)$-superstratum differs from the full black hole ensemble it is part of. While the calculations performed above are most natural in momentum space, an observer probing the microstate geometry from far away will be more interested in position space results.

There are many choices of correlation functions in a Lorentzian field theory, including the Feynman, Wightman, Advanced and Retarded propagators (see Appendix A). These represent distinct physical quantities and are obtained by choosing different time orderings in position space, or by integrating in a particular way around poles of the momentum-space propagator. To clarify the different two-point functions, we will begin with two well understood examples in detail: global AdS and extremal BTZ. In the last part of this section, we study the position-space propagator in the superstratum background.

## 6.1   Position-space Green functions in extremal BTZ

The inverse Fourier transform of the BTZ response function (3.25) splits into a left-moving and right-moving part. Since the BTZ metric has no *a priori* periodic identification of the $y$-circle[14], the result is valid for continuous or quantized conjugate momentum. We will therefore perform the inverse transformation in $p$ and $\omega$ independently, and impose spatial periodicity later.

The left-moving part of the propagator (3.25) has poles at $p = ir_H(\Delta + 2m)$ with corresponding residues $-2ir_H(-1)^m/m!$ for $m \in \mathbb{Z}$, $m > 0$. Since they are in the upper imaginary $p$-plane, these poles are only picked up by the contour when it is closed in the upper half-plane. This happens only when $v < 0$, leading to the retarded propagator:

$$
\int \frac{dp}{2\pi} e^{-ip\frac{v}{R_y}} \frac{\Gamma(2-\Delta)\,\Gamma\big(\frac{1}{2}(\Delta + \frac{ip}{r_H})\big)}{\Gamma(\Delta)\,\Gamma\big(1 - \frac{1}{2}(\Delta - \frac{ip}{r_H})\big)} = \theta(-v)\frac{2(1-\Delta)}{r_H\,\Gamma(\Delta)} e^{\frac{r_H\Delta v}{R_y}} \sum_{m=0}^{\infty} \frac{(-1)^m\,\Gamma(1-\Delta)\, e^{\frac{2r_H m v}{R_y}}}{\Gamma(m+1)\,\Gamma(1-\Delta-m)}
$$

$$
= \theta(-v)\frac{2(1-\Delta)}{r_H\,\Gamma(\Delta)} e^{\frac{r_H\Delta v}{R_y}} \left(1 - e^{\frac{2r_H v}{R_y}}\right)^{-\Delta}
$$

$$
= -\frac{2\theta(-v)}{r_H\,\Gamma(\Delta-1)} \left[-2\sinh\left(\frac{r_H v}{R_y}\right)\right]^{-\Delta}. \tag{6.1}
$$

Note that the series that appears is convergent since $|e^{2r_H v/R_y}| < 1$.

For the right-moving part, $(-2i\omega r_H)^{\Delta-1}$, we can split up the integral in $\omega > 0$ and $\omega < 0$. The first part gives the integral representation of the $\Gamma$ function

$$
\int_0^\infty \frac{d\omega}{2\pi} e^{-i\omega\frac{u}{R_y}} \big(-2ir_H\omega\big)^{\Delta-1} = \frac{1}{2\pi} e^{-i\frac{\pi}{2}(\Delta-1)}(2r_H)^{\Delta-1} \int_0^\infty d\omega\, e^{-i\omega\frac{u}{R_y}}\, \omega^{\Delta-1}
$$

$$
= \frac{1}{2\pi} e^{-i\frac{\pi}{2}(\Delta-1)}(2r_H)^{\Delta-1} \Big(\frac{iu}{R_y}\Big)^{-\Delta} \Gamma(\Delta). \tag{6.2}
$$

The contribution from $\omega < 0$ gives exactly the complex conjugate of this result. Since it is purely imaginary for $u < 0$, the sum is only non-zero for positive $u$. Putting it all together we get

$$
R(u,v) = -\theta(u)\theta(-v)\frac{2}{\pi}(\Delta-1)(2r_H)^\Delta \sin(\pi\Delta)\Big(\frac{u}{R_y}\Big)^{-\Delta} \left[-2\sinh\left(\frac{r_H v}{R_y}\right)\right]^{-\Delta}
$$

$$
= -\theta(t)\frac{2i}{\pi}(\Delta-1)\,r_H{}^\Delta \left(\left[\sinh\left(\frac{r_H v}{R_y} + i\epsilon\right)\left(\frac{u}{R_y} - i\epsilon\right)\right]^{-\Delta}\right. \tag{6.3}
$$

$$
\left. - \left[\sinh\left(-\frac{r_H v}{R_y} + i\epsilon\right)\left(-\frac{u}{R_y} - i\epsilon\right)\right]^{-\Delta}\right).
$$

---

[14]This is to be contrasted with the case of AdS, where the absence of a conical singularity at the origin imposes a periodicity condition of thy $y$-circle.

In the final step we wrote the result as the difference of two terms, which cancel against each other outside of the light-cone (i.e. when $u$ and $v$ have the same sign). This allowed us to replace $\theta(u)\theta(-v)$ by $\theta(t)$. That in turn required us to add appropriate $i\epsilon$-prescriptions, since for certain values of $u$ and $v$, we are taking fractional powers ($\Delta$) of a negative real number. The reason we rewrote the result in this way, is because it can now be identified as the retarded propagator (A.18), which is $\theta(t)$ times the expectation value of the commutator of $\mathcal{O}(u,v)$ with $\mathcal{O}(0,0)$. The two terms of the commutator correspond to the two terms in (6.3).

As noticed earlier, the BTZ metric (3.18) is valid for non-compact $y$ with $-\infty < y < \infty$. One way to get the position-space Green function for a compact $y$-circle, is to sum over images:

$$R_c(u,v) = -\frac{2i(\Delta-1)}{\pi}\theta(t)\,r_H{}^\Delta \sum_{m\in\mathbb{Z}}\left(\left[\sinh\left(\frac{r_H(v-2\pi m R_y)}{R_y}\right)+i\epsilon)(\frac{u}{R_y}+2\pi m-i\epsilon)\right]^{-\Delta}\right. \tag{6.4}$$
$$\left.-\left[\sinh\left(-\frac{r_H(v-2\pi m R_y)}{R_y}\right)+i\epsilon)(-\frac{u}{R_y}+2\pi m-i\epsilon)\right]^{-\Delta}\right).$$

The result (6.4) is depicted in Figures 10 and 11.

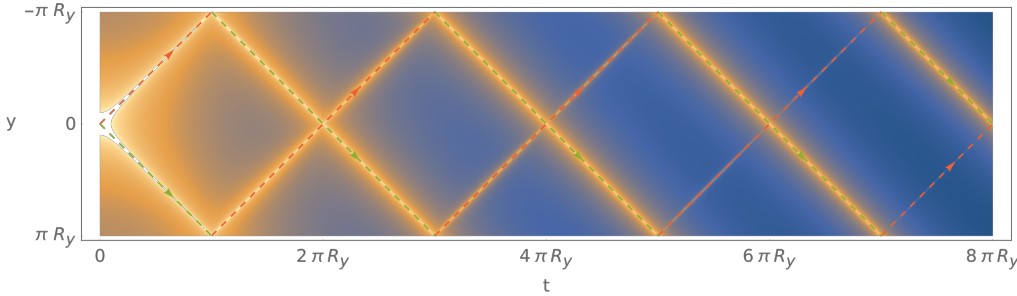

Figure 10: Logarithmic densityplot of the two-point function (6.4) on a cylinder $(t,y) \cong (t,y+2\pi R_y)$ with $r_H/R_y = \pi/20$ (i.e. $\beta_L = 20$ and $\beta_R \to \infty$). The two-point function is sharply peaked on the light-cone (yellow), falling off polynomially in $u$ (green arrows) and exponentially in $v$ (red arrows). When following a light-ray (e.g. at constant $v$), the two-point function has peaks with periodicity $2\pi R_y$ whenever it meets the light-ray going in the other direction. Note that there are no reflecting boundary conditions in this figure. Instead the $y$-circle is periodic.

The periodic images result in a different long-time behavior of the two-point function on the cylinder compared to that on the plane. Their impact is most dramatic in the left-moving sector, i.e. when evolving in $v$ while keeping $u$ fixed. This is depicted in the second panel of Figure 11 and in Figure 12. As long as $\frac{r_H}{R_y}v \lesssim 1$, the two-point function is dominated by the $m=0$ image. Afterwards, the $m=0$ image decays much faster than the total two-point function, and it is the "nearest" image (with $m\approx v/2\pi R_y$) is dominant.

This observation can be made a bit more precise by considering when the $m^{\text{th}}$ image becomes larger than the $0^{\text{th}}$:

$$\left|\sinh(\frac{r_H}{R_y}v)u\right|^{-\Delta} < \left|\sinh[\frac{r_H}{R_y}(v-2\pi R_y m)](u+2\pi R_y m)\right|^{-\Delta}. \tag{6.5}$$

When $\frac{r_H}{R_y}v \ll 1$, the left-hand side can be approximated by $(\frac{r_H}{R_y}vu)^{-\Delta}$, which means that the $m^{\text{th}}$ image only dominates when $|v-2\pi R_y m| \lesssim |v||u/2\pi R_y m|$, for small $u$. This is only true whenever the left-moving light-ray crosses the right-moving one. In the opposite regime $\frac{r_H}{R_y}v \gg 1$, the exponential growth of the sinh-function takes over. The equation (6.5) reduces

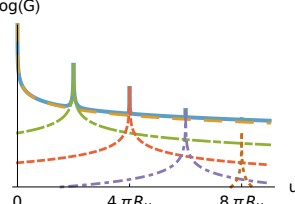 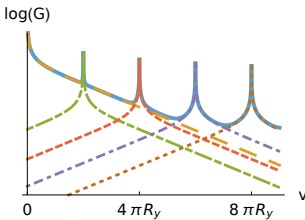 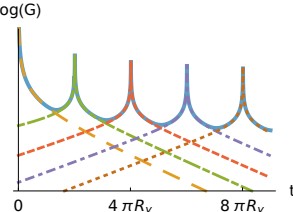

Figure 11: Logarithmic plot of the two-point function (6.4) with $\frac{r_H}{R_y} = \pi/8$ along three different directions ($u$, $v$ and $t$) while keeping the orthogonal direction ($v$, $u$ and $y$, resp.) constant, here: $R_y/20$. The total value of the two-point function is plotted (fat, blue) as well as the individual contributions from the $m = \{0, 1, 2, 3, 4\}$ modes (increasingly finely dashed lines). In the $u$ direction, the behavior remains dominated by the $m = 0$ mode. The other modes appear briefly as poles on the light-cone. The behavior in the $v$ direction is drastically different. Starting at $v$ of order the inverse temperature, the $m = 0$ mode loses dominance to the "nearest" mode. The behavior it $t$ shares properties of both $u$ and $v$. The fall-off is exponential between the poles, but the exchange of dominance between the modes makes the long time behavior polynomial.

to

$$\sinh\left|\frac{r_H}{R_y}(v - 2\pi R_y m)\right| < \left|\frac{u}{u + 2\pi R_y m}\right| e^{\frac{r_H}{R_y} v}. \tag{6.6}$$

This inequality is first satisfied when $v$ is roughly halfway between 0 and $2\pi R_y m$. In this regime, the two-point function is always dominated by the closest image. Thus, at long time scales, the behavior of the two-point function is drastically altered by the compact $y$-circle: the propagator is not suppressed as $e^{-\Delta \frac{r_H}{R_y} v}$ but roughly as $v^{-\Delta}$.

## 6.2 Position-space Green functions in AdS$_3$

The Fourier transform of the AdS propagators $R_1$ and $R_2$ in (3.45) is more complicated because they have poles on the real $\omega$ and $p$ axes. Unlike BTZ, the retarded propagator is no longer singled out by the location of the poles. Instead, there are rather simple and direct routes to link $R_1$ and $R_2$ to various Green functions by altering the contour prescription. We will focus on retarded two-point functions as before. By extracting the $\omega$-dependent part of $R_1$, we define[15]:

$$\tilde{I}_1(\omega) \equiv -i\,\Gamma(1 - \Delta)\frac{\Gamma(\frac{1}{2}\Delta + \omega)}{\Gamma(1 - \frac{1}{2}\Delta + \omega)}, \tag{6.7}$$

and consider the Fourier transform to position space

$$I_1(u) \equiv \int \frac{d\omega}{2\pi} e^{-i\omega \frac{u}{R_y}} \tilde{I}_1(\omega + i\epsilon), \tag{6.8}$$

where we have introduced $\epsilon > 0$ to make $\tilde{I}_1$ analytic in the upper complex $\omega$ plane. This is the appropriate continuation to calculate the retarded propagator. This makes the inverse Fourier transform vanish for $\text{Re}(u) < 0$ where the contour can be deformed to $\text{Im}(\omega) \to \infty$. The $\Gamma$-function in the numerator has poles at $\omega = -i\epsilon - (\frac{1}{2}\Delta + m)$, with residues $(-1)^m/m!$ for

---

[15]The $p$-dependent part of the response function is identical and the analysis can be done in a similar way.

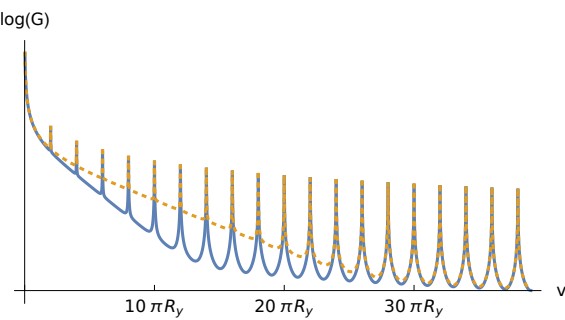

Figure 12: Logarithmic plot of the two-point function (6.4) in the $v$-direction with $u = R_y/20$ held constant. The left-moving temperature is $\frac{r_H}{R_y} = \pi/20$ for the solid blue line and $\frac{r_H}{R_y} = \pi/40$ for the dotted orange line. The exponential fall-off (linear on this plot) is visible at early times, as long as the $m = 0$ mode dominates the propagator. In this regime, the other modes appear only briefly as poles. For $v$ larger than $\pi\beta_L$, the behavior changes drastically. The propagator locally still behaves as an exponential, but the enveloping curve is polynomial at long times.

any non-negative integer $m$. For $\text{Re}(u) > 0$, we can again express as a sum over the residues:

$$
\begin{aligned}
I_1(u) &= -i\,\theta(u)\,\Gamma(1-\Delta)\sum_{m=0}^{\infty}\frac{2i\pi\,e^{iu(\frac{1}{2}\Delta+m+i\epsilon)/R_y}\,(-1)^m}{2\pi\,m!\,\Gamma(1-\Delta-m)} \\
&= \theta(u)\big[-2i\sin\big(\tfrac{u}{2R_y}\big)\big]^{-\Delta}.
\end{aligned}
\tag{6.9}
$$

Note that the series is convergent if $|e^{iu/R_y}| < 1$, which corresponds to $\text{Im}(u) > 0$, or, if one keeps $u$ real, one must deform $u \to u + i\epsilon$:

$$
I_1(u) = \theta(u)\big[-2i\sin\big(\tfrac{(u+i\epsilon)}{2R_y}\big)\big]^{-\Delta}.
\tag{6.10}
$$

Up to an overall normalization, this is the left-moving part of the advanced Green function of the CFT on the cylinder.
Similarly, for the response function, $R_2$, one can make the Fourier transform of

$$
\tilde{I}_2(\omega) \equiv -i\,\Gamma(1-\Delta)\frac{\Gamma(\frac{1}{2}\Delta-\omega)}{\Gamma(1-\frac{1}{2}\Delta-\omega)},
\tag{6.11}
$$

with the same $\omega \to \omega + i\epsilon$. One then finds

$$
I_2(u) = \theta(u)\big[2i\sin\big(\tfrac{(u-i\epsilon)}{2R_y}\big)\big]^{-\Delta}.
\tag{6.12}
$$

The shift $u \to u - i\epsilon$ arises because the sum over poles only converges if $|e^{-iu/R_y}| < 1$, which corresponds to $\text{Im}(u) < 0$. Thus, the two response functions only differ by a phase and the space-time $i\epsilon$ prescription. To get the retarded Green function, one simply gives a small and *negative* imaginary part to the poles $\omega$.

In the foregoing computation, we took the liberty of ignoring the periodicity of the $y$ coordinate. Since this periodicity is fixed by smoothness at the origin of $AdS_3$, we must redo the computation more carefully. The Fourier transform of the response function, $R_1$, is, more correctly, given by:

$$
I_{full}(t,y) \equiv \sum_{k=-\infty}^{\infty}\int\frac{d\varpi}{2\pi}\,e^{-\frac{i\varpi t}{R_y}}\,e^{\frac{iky}{R_y}}\,\frac{\Gamma(1-\Delta)\,\Gamma\big(\frac{1}{2}(\Delta+k+\varpi)\big)\,\Gamma\big(\frac{1}{2}(\Delta+k-\varpi)\big)}{\Gamma(\Delta-1)\,\Gamma\big(1-\frac{1}{2}(\Delta-k-\varpi)\big)\,\Gamma\big(1-\frac{1}{2}(\Delta-k+\varpi)\big)},
\tag{6.13}
$$

where $\varpi = (\omega - p)$ is the continuum momentum along $t$ and $k = (\omega + p) \in \mathbb{Z}$ is the discrete Fourier mode around $y$.

There are now two sets of poles: $\varpi = \Delta + k + 2m$ and $\varpi = -\Delta - k - 2m$, $m \in \mathbb{Z}$, $m \geq 0$. For $2\Delta \notin \mathbb{Z}$, these poles never coincide. Moreover at these poles, the denominator always contains a factor of $\Gamma(1 + k + m)$, which means that there are only non-zero residues for $k + m \geq 0$. Thus, the non-trivial residues split into positive and negative frequencies: $\varpi = -\Delta - k - 2m < 0$ and $\varpi = \Delta + k + 2m > 0$.

A sum over the residues of the poles at $\varpi = \pm(\Delta + k + 2m)$ gives

$$\frac{\Gamma(1-\Delta)}{\Gamma(\Delta-1)} \sum_{k=-\infty}^{\infty} \sum_{m=0}^{\infty} (-1)^m \frac{\Gamma(k+m+\Delta)}{\Gamma(m+1)\Gamma(1-\Delta-m)\Gamma(1+k+m)} e^{\pm i(\Delta+k+2m)\frac{t}{R_y}} e^{\frac{iky}{R_y}}$$

$$= e^{\pm \frac{i\Delta t}{R_y}} \left[ \sum_{s=0}^{\infty} \frac{\Gamma(s+\Delta)}{\Gamma(\Delta-1)\Gamma(1+s)} e^{\pm \frac{is(t\pm y)}{R_y}} \right] \left[ \sum_{m=0}^{\infty} (-1)^m \frac{\Gamma(1-\Delta)}{\Gamma(m+1)\Gamma(1-\Delta-m)} e^{\pm \frac{im(t\mp y)}{R_y}} \right], \tag{6.14}$$

where $s = (k + m)$ and we used the fact that the residues vanish unless $s \geq 0$. Now, we use the identity

$$\Gamma(\Delta + s) = (-1)^s \frac{\Gamma(\Delta)\Gamma(1-\Delta)}{\Gamma(1-\Delta-s)} \tag{6.15}$$

to rewrite the sum over residues as

$$(\Delta-1) e^{\pm \frac{i\Delta t}{R_y}} \left[ \sum_{s=0}^{\infty} (-1)^s \frac{\Gamma(1-\Delta)}{\Gamma(1-\Delta-s)\Gamma(1+s)} e^{\pm \frac{is(t\pm y)}{R_y}} \right]$$

$$\times \left[ \sum_{m=0}^{\infty} (-1)^m \frac{\Gamma(1-\Delta)}{\Gamma(m+1)\Gamma(1-\Delta-m)} e^{\pm \frac{im(t\mp y)}{R_y}} \right]$$

$$= (\Delta-1) \left[ \mp 2i \sin\left(\frac{t\pm(y+i\epsilon)}{2R_y}\right) \right]^{-\Delta} \left[ \pm 2i \sin\left(\frac{t\mp(y+i\epsilon)}{2R_y}\right) \right]^{-\Delta}. \tag{6.16}$$

This is the standard form of the CFT propagator on the cylinder defined by $(t, y)$, with $y \cong y + 2\pi R_y$. Whether one picks up the positive- or negative-frequency poles depends on the sign of $t$ in (6.13) and whether one integrates above or below the poles along the real axis (or, equivalently, whether one shifts the frequencies according to $\varpi \to \varpi \mp i\epsilon$). For example, the Feynman propagator is given by integrating above the positive frequency poles and below the negative frequency poles.

One should note that, in our discussion of the Green functions, we have ignored the Heaviside functions in (3.46) and worked with $R_1$. To get the Feynman propagator from $R^{\text{AdS}_3}$ requires a much more complicated set of contour deformations and analytic continuations. This is discussed in great detail in [40, 41, 43]. The important bottom line here is that the construction of Green functions in global $\text{AdS}_3$ is well-understood and, because the cap of superstratum is a close approximation to the global $\text{AdS}_3$ with all the concomitant bound states and poles, the construction of Green functions will follow the same prescriptions that one uses in global $\text{AdS}_3$ itself.

## 6.3 Position-space Green functions in (1,0,n) superstrata

The momentum-space analysis of the superstratum two-point function leads to a rather unwieldy result (5.2), which makes the Fourier transform back to position space very complicated. In this section, we sketch the overall profile of the Green function in position space.

There are three relevant time scales. We will argue that for times shorter than $\frac{bR_y}{a} \sim \sqrt{N_1 N_5} R_y$, the propagator is dominated by the extremal-BTZ response. Beyond that

time, up to times of order $\frac{b^2 R_y}{\sqrt{n}\, a^2} \sim N_1 N_5 R_y / \sqrt{n}$, the correction from the intermediate regime will change the position space propagator away from the extremal BTZ expectation. Finally, at a time of order $\frac{b^2 R_y}{a^2} \sim N_1 N_5 R_y$ the discrete energy spectrum will become significant and will lead to huge echoes from the cap, similar in shape to the Green function at very small times, but with certain slight deformatons.

First, we will argue that the contribution from $\tan\Theta$ to the extremal-BTZ response does not drastically alter the propagator at short time scales. We will model the full propagator in momentum space as $R^{\text{BTZ}}(\Omega, P) \cdot \tilde{g}(\Omega, P)$[16], where $\tilde{g}$ encodes the modulation from $\tan\Theta$. If the spectrum was perfectly linear, we could choose $\tilde{g}(\Omega, P) = \tan(\frac{b^2 \Omega}{a^2} \pm P)$. The position space propagator is then given by the convolution

$$G(u, v) = \int \frac{\mathrm{d}\lambda \, \mathrm{d}\kappa}{(2\pi)^2} R^{\text{BTZ}}(\lambda, \kappa) \, g(u - \lambda, v - \kappa) \,, \tag{6.17}$$

$$g(u, v) \propto \delta\left(\frac{b^2}{a^2} v \mp u\right) \sum_m \delta\left(u + \frac{2b^2}{a^2} m R_y\right) \,, \tag{6.18}$$

where $g$ is the formal inverse Fourier transform of $\tan(\frac{b^2 \Omega}{a^2} \pm P)$. This implies that the position space propagator is periodic in the direction orthogonal to $u \mp \frac{a^2}{b^2} v$, repeating itself whenever $u$ increases with $\frac{2b^2}{a^2} R_y$, or equivalently after $t$ increases with $\frac{b^2}{a^2} \pm 1$ in that direction. After that time, the response is a perfect echo of the extremal BTZ answer. This was under the assumption that the energy spectrum is perfectly linear. Since the superstratum spectrum is slightly anharmonic, the consecutive echoes will instead be slightly more deformed and attenuated. This is represented by the large peak at $t \sim b^2/a^2$ in Figure 13.

According to the second line of (5.2), the response is not really that of extremal BTZ, but rather with the replacement $P \to \bar{P} = P - \frac{a^2 \Delta(\Delta - 2)}{b^2 \Omega}$. In the Fourier transform to position space, we can change basis to get

$$\int \frac{\mathrm{d}\Omega \, \mathrm{d}\bar{P}}{(2\pi)^2} e^{-i(\Omega u + \bar{P} v)/R_y} e^{\frac{-ia^2 \Delta(\Delta-2)v}{b^2 \Omega}} R^{\text{BTZ}}(\Omega, \bar{P}) \,. \tag{6.19}$$

Using the expression (3.25) for the momentum space BTZ propagator, this integral separates into the integral over $\bar{P}$, which is unaltered, and the integral over $\Omega$

$$\int \frac{\mathrm{d}\Omega}{2\pi} e^{-i\Omega u/R_y} e^{\frac{-ia^2 \Delta(\Delta-2)v}{b^2 R_y \Omega}} (-i\Omega)^{\Delta-1} = \left(\frac{-i\, b^2 u}{a^2 \Delta(\Delta-2) v}\right)^{-\Delta/2} J_{-\Delta}\left(2\sqrt{\frac{ia^2 \Delta(\Delta-2) v u}{b^2 R_y^2}}\right) \,, \tag{6.20}$$

where $J_{-\Delta}$ is the Bessel function of the first kind.[17] When we expand it for small values of $v u$, we find

$$\frac{1}{\Gamma(1+\Delta)} \left(\frac{-u}{R_y}\right)^{-\Delta} \left(1 + \frac{ia^2 \Delta(\Delta - 2) v u}{(1-\Delta) b^2 R_y^2} + \mathcal{O}(\tfrac{a^2 u v}{b^2})^2\right) \,, \tag{6.22}$$

---

[16]One should consider the imaginary part of $R^{\text{BTZ}}$. However, the Fourier transform of the imaginary part can be obtained from the Fourier transform of the function and its conjugate. One can then consider $R^{\text{BTZ}}$ only for the ease of the discussion.

[17]This identity follows from the integral representation of the Bessel function,

$$J_m(x) = \frac{1}{2\pi} \int_{-\pi}^{\pi} \mathrm{d}\tau \, e^{ix \sin\tau - im\tau} \,, \qquad\qquad \Omega = i\sqrt{\frac{a^2 \Delta(\Delta-2) v}{b^2 u}} \, e^{i\tau} \,. \tag{6.21}$$

The $i\epsilon$-prescription used for the purpose of this illustration is therefore $-i\Omega \to -i\Omega + \text{sign}(u)\epsilon$.

which follows the behavior of the extremal BTZ black hole up to time scales where $vu \sim \frac{b^2}{a^2}R_y^2$. This deviation first becomes significant at times of order $t = \frac{b}{a}R_y$, as depicted by the blue and red lines in Figure 13.

Finally, the response (5.2) differs from the extremal-BTZ response whenever $|\Omega| \lesssim \frac{\sqrt{n}a^2}{b^2}$. We can model this in two steps. First the momentum-space BTZ propagator is multiplied by a high-pass filter such as $\tilde{g}(\Omega, P) = \theta(\Omega - \frac{\sqrt{n}a^2}{b^2}) + \theta(-\Omega - \frac{\sqrt{n}a^2}{b^2})$. Second, we add the AdS$_3$ regime of (5.2). The first step is to apply (6.17) again, but with

$$g(u,v) = (2\pi)^2 \delta(u)\delta(v) - \frac{2}{u}\sin\left(\frac{\sqrt{n}a^2 u}{b^2}\right)\delta(v). \tag{6.23}$$

The second function is very spread out, but it is also suppressed by $n^{1/4}a/b$. Its effect is suppressed by $1/\sqrt{N_1 N_5}$. To add the AdS$_3$ regime, we add the position space AdS$_3$ propagator with a low-pass filter $\tilde{g} = \theta(\Omega + \frac{\sqrt{n}a^2}{b^2}) - \theta(\Omega - \frac{\sqrt{n}a^2}{b^2})$, the Fourier transform of which is just the second term of (6.23). All in all, the position space propagator obtained by replacing the BTZ propagator with the AdS$_3$ propagator for energies $|\Omega| \lesssim \frac{\sqrt{n}a^2}{b^2}$ is

$$R^{\text{BTZ}}(u,v) + \frac{1}{\pi u}\int \frac{d\lambda}{2\pi}\left(R^{\text{AdS}}(\lambda, v) - R^{\text{BTZ}}(\lambda, v)\right)\sin\left(\frac{\sqrt{n}a^2(u-\lambda)}{b^2}\right). \tag{6.24}$$

The AdS$_3$ response function has the same poles as the $\tan\Theta$ term, and hence will contribute to the recurrences (6.18).

Notice that $R^{\text{BTZ}}$ decays like $\exp(-\frac{\Delta r_H}{R_y}v)$ as long as $v \lesssim R_y/2$,[18] whereas the second term in (6.24) is of order $n^{1/4}a/b\,(v/R_y)^{-\Delta}$. This is significant, because it means that the second term is comparable to the first at high temperatures

$$r_H = \sqrt{n} \sim \frac{1}{\Delta}\log\left(\frac{b^2}{\sqrt{n}a^2}\right). \tag{6.25}$$

For the superstratum geometry to accurately imitate a black hole at least at times of the order $R_y$, the temperature must be small enough so that the second contribution in (6.24) is negligible. This requires $\Delta\sqrt{n} \ll \log(b^2/(\sqrt{n}a^2))$.

# 7 Final comments

This paper contains the first computation of a correlator of two light operators in a geometry that has the same asymptotic region and the same throat as an extremal three-charge D1-D5-P black hole, but differs from it at the scale of the horizon. This two-point function is holographically dual to a CFT four-point function of two light and two heavy states (HHLL).[19]

Most HHLL correlators that have been so far computed in the bulk [12–17] involve heavy states that are quite far away from the heavy states that one expects to contribute to the entropy of the BTZ black hole. In "microstate geometry" language, the geometries dual to these states have shallow throats that do not contain the $AdS_2$ very-near-horizon geometry characteristic of extremal BTZ black holes.[20] In contrast, the geometries dual to heavy states we consider

---

[18] This is where, for large values of $r_H$, the first image in Figure 11 takes over. For small values of $r_H$, the exponential decay lasts as long as $\frac{r_H}{R_y}v \ll 1$, but in this regime, it does not become parametrically small.

[19] For interesting recent work on CFT three-point function of one light and two similar heavy states see for example [54–57].

[20] Although this was not made explicit in [15], the calculation of the large momentum limit of the Wightman function also applies to microstate geometries that have long BTZ throats and small angular momentum. The microstate result spectacularly agrees with the black hole result, up to $1/N_1 N_5$ corrections (which perhaps makes the title of [15] somewhat ironic).

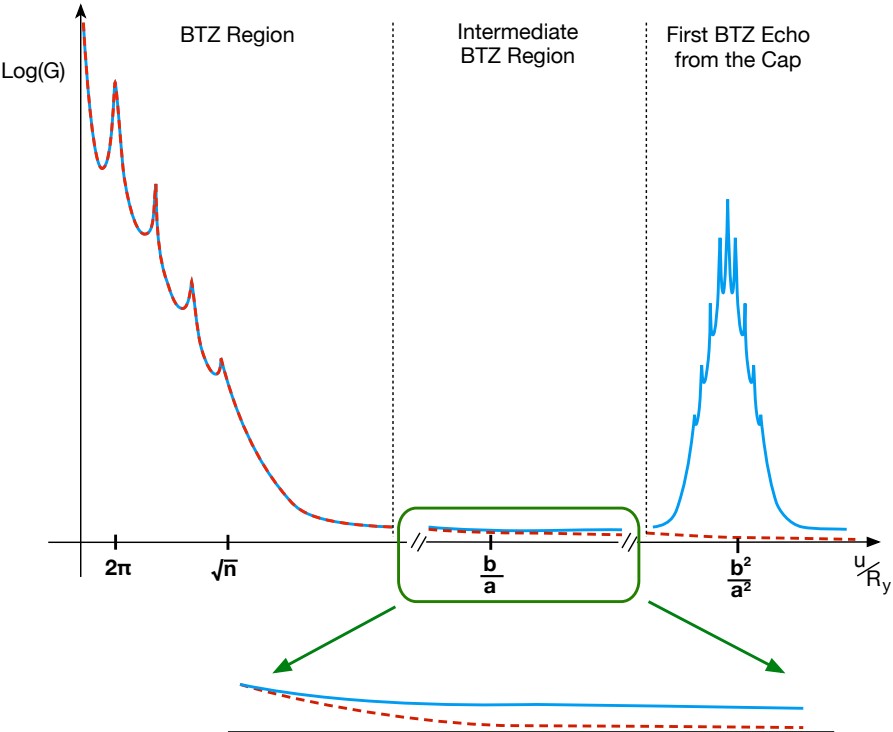

Figure 13: Schematic description of the $(1, 0, n)$-superstratum Green function in position space (continuous line in blue) and its comparison to the extremal BTZ Green function (dashed line in red). At early times of order $R_y$, the behavior is very similar to that in Figure 11, with polynomial fall off briefly interrupted by singularities at every $\Delta u = 2\pi R_y$ from the light-cone wrapping around the $y$-direction. (The singularities are cut off in the picture to clarify their relative weight.) At times of order $b R_y / a \sim \sqrt{N_1 N_5} R_y$, the behavior of the superstratum two-point function starts to deviate from that of extremal BTZ, following (6.22). Finally, at times of order $b^2 R_y / a^2 \sim N_1 N_5 R_y$, the superstratum Green function features its first significant echo which is absent in BTZ. It is a slightly-attenuated and deformed copy of the singularity around $u \approx 0$. Ever less significant echos will follow at times equal to integer multiples of $b^2 R_y / a^2$.

have an arbitrarily long $AdS_2$ throat, and only differ from the BTZ black hole arbitrarily close to the horizon. This can also be seen from the fact that the mass gap of these geometries is the same as the mass gap of the typical momentum-carrying states of the D1-D5 CFT [15, 18–20].

Hence, one expects the HHLL four-point function we compute in the bulk to display thermal behavior for times smaller than the CFT central charge, and thus to match the four-point function computed when the heavy state is taken to be the thermal state (dual in the bulk to the BTZ black hole). However, for longer times one expects to see deviations from eternal thermal decay, in particular to avoid a clash with constraints from unitarity [58].

This expectation was worked out in more detail in the CFT, where it was argued that HHLL four-point function computed using typical heavy states are expected to differ at late times from the four-point functions computed using the thermal state [59–67]. Our calculation confirms this from the bulk perspective, and moreover shows that this late-time non-thermal behavior happens exactly because the geometries dual in the bulk to low-mass-gap CFT states (of the type that give rise to the black hole entropy) differ from the BTZ black hole at the scale of the horizon.

Seeing thermal behavior in the absence of a horizon appears to be quite counterintuitive, especially in light of the intuition that thermalization comes from absorption of stuff by the black hole and the common conceit that only solutions with a horizon can describe typical black hole microstates. Our result shows that the horizonless microstate geometries with long $AdS_2$ throats give rise, at times shorter than the central charge, $N_1 N_5$, to exactly the same thermal behavior one finds in the black hole solution, while avoiding the information-loss problems associated to the presence of a horizon and restoring the information after long times.

This being said, the supergravity solutions that we use to compute HHLL correlators are quite far from the most generic horizonless supergravity solutions one can construct, and hence the late-time behavior of their correlators, while consistent with information recovery, is far from generic. Indeed, we have found that after times of order the central charge, the two-point function "comes back from the grave" to a value that is just a tiny bit smaller than its starting value, then decays thermally again, then comes back from the grave to a smaller value, then decays again, etc.

It would be very interesting to try to extend our calculations to more generic superstrata, including the recently-constructed supercharged [68,69] and internally-excited ones [70] and to see whether one can obtain a behavior at long times that better approximates thermal behavior. Unfortunately, the most generic superstratum solutions that have the same asymptotics and throat as a BTZ black hole are parameterized by arbitrary functions of *three* variables [69], and hence their metric and fields are complicated functions of five variables! While this represents a big achievement for the microstate geometry programme, computing holographic in such cohomogeneity-five solutions is way beyond the current analytical and numerical technology (even finding solutions to the wave equation is hard). The solutions we use are much simpler single-mode $(1, 0, n)$ superstrata, in which the wave equation is separable, and one can compute two-point functions without resorting to heavy numerics. As we have seen in this paper, even to compute two-point functions in our geometries we had to develop a new hybrid-WKB technology. To repeat our calculation for more complicated superstrata one would need to extend this hybrid-WKB technology to functions of two or more variables, which appears quite challenging.

However, being able to compute the two-point functions in more generic superstrata would open up a plaethora of interesting research directions. First, the mass gaps and the level spacing of more complicated superstrata differ significantly from those of the $(1, 0, n)$ solutions we have studied. Thus, the time at which the bounce in the propagator happens would be different. Second, the reason for which the correlators we compute have order one resurgences at times of order $N_1 N_5$ is most likely that the wave equation is separable, and hence a spherically-symmetric wave packet cannot dump energy into modes that contain higher spherical harmonics on $S^3$, and bounces back without any distortions. In a less symmetric superstratum geometry, one expects all the higher spherical harmonics on $S^3$ to be populated after the wave packed bounces back from the cap, and hence the peak one sees after times of order $N_1 N_5$ to be much smaller and much less spiky.

Third, computing two-point functions in several different superstratum geometries would allow us to also calculate two-point functions of the light operator $O_L$ functions in a mixed state described using a density matrix constructed from several heavy states, $H_i$:

$$\rho_{\text{mixed}} = c_i |H_i\rangle\langle H_i|. \tag{7.1}$$

In principle our results allow us to already compute an expectation value in a mixed state described by a density matrix constructed from the heavy states dual to $(1, 0, n)$ superstratum, but since these superstrata have very similar response functions, thermal averaging over them would not do too much. However, if one could average over states whose 2-point functions are very different one would probably obtain a result very similar to what expects from the

BTZ black hole solution. This would be further evidence that the classical BTZ black hole is not dual to a pure state of the dual CFT, but rather to a mixed state described by a thermal density matrix.

Aside from the study of microstate geometries, the hybrid-WKB technology we have developed will have applications in other areas of holography, as it is the only technique that can be used to compute holographic correlation functions in backgrounds where the wave equation can only be solved analytically in the asymptotic region. This will be very important if, for example, one wants to compute such correlation functions in theories that are more general than $\mathcal{N}=4$ Super-Yang-Mills in flat space, but whose holographic dual is still cohomegeneity-one. Important examples of this include the Klebanov-Strassler solution [71] and large black holes in $AdS_5 \times S^5$, which are dual to the thermal phase of $\mathcal{N}=4$ Super-Yang-Mills on a three-sphere. Of course, if one could develop the hybrid-WKB technology to compute correlators in backgrounds of higher cohomogeneity this would open up even more areas of exploration.

# Acknowledgments

We are very grateful to Monica Guică for her insights in early stages of this project and for comments on drafts of this paper. The work of IB, PH and RM was supported in part by the ANR grant Black-dS-String ANR-16-CE31-0004-01. In addition the work of IB was supported by the John Templeton Foundation grant 61169 and the work of RM was also supported by a CEA Enhanced Eurotalents Fellowship and by the ERC Starting Grant 679278 Emergent-BH. The work of NW was supported in part by the ERC Grant 787320 - QBH Structure and by the DOE grant DE-SC0011687.

# Appendices

# A Two-point functions and propagators

In this section, we summarize the definitions as well as some properties and relations of the different types of two-point functions for scalar operators that exist in Lorentzian signature.

## A.1 Definitions

### A.1.1 Euclidean signature

To start with the simplest situation, we will consider a field theory on the Euclidean plane $\mathbb{R}^d$ and assume (as usual) that the theory has a Hilbert space $\mathcal{H}$ of states $|\psi\rangle$ in a representation of the Euclidean Poincaré group. We can label local operators by their mass and spin. For scalar operators $\mathcal{O}$, consider the two-point function in state $\rho$

$$\langle \mathcal{O}_1(x)\mathcal{O}_2(0)\rangle_\rho^E \equiv \text{Tr}\left(\rho\,\mathcal{O}_1(x)\mathcal{O}_2(0)\right), \tag{A.1}$$

where the superscript $E$ clarifies that we are in Euclidean signature. A special class of states $\rho$ are the pure states $|\psi\rangle\langle\psi|$. In particular, we will assume that there is a Poincaré invariant vacuum $|0\rangle\langle 0|$. This is immediately an example of another special class of states, namely the Poincaré-invariant ones that satisfy

$$\rho = e^{-ix^m \hat{P}_m}\,\rho\,e^{ix^m \hat{P}_m}, \qquad\qquad \rho = e^{-iR^{mn}\hat{M}_{mn}}\,\rho\,e^{iR^{mn}\hat{M}_{mn}}, \tag{A.2}$$

where the $\hat{P}_m$ and $\hat{M}_{mn}$ are the generators of translations and rotations, and $m, n = 1 \ldots d$. For these states, the corresponding two-point function (A.1) is only a function of $|x|$.

Apart from the two-point function in (A.1), one could consider the symmetrized and anti-symmetrized version. However, since rotations map any point at a distance $|x|$ from the origin to any other, there seems to be no reason to consider "space-ordered" correlators, at least for rotationally invariant states.

### A.1.2 Unitarity and spectrum conditions

In practice we are often interested in Euclidean field theories which are the Wick rotation of a "unitary" Lorentzian theory and hence one of the directions is singled out, $\tau \equiv x^d$. The corresponding translation generator $\hat{P}_d$ is declared anti-Hermitian with respect to the inner product of the Hilbert space, whereas $\hat{P}_a$ ($a = 1, \ldots d-1$) are Hermitian.[21] This makes the Hamiltonian $\hat{H} \equiv -i\hat{P}_d$ Hermitian. Furthermore, any operator that is Hermitian at $\tau = 0$ will satisfy the "reflection positivity" property

$$\mathcal{O}(\tau, \vec{x})^{\dagger} = \left(e^{\tau\hat{H}-ix^a\hat{P}_a}\mathcal{O}(0,0)e^{-\tau\hat{H}+ix^a\hat{P}_a}\right)^{\dagger} = \mathcal{O}(-\tau, \vec{x}) . \tag{A.3}$$

It is usually assumed that the spectrum of the Hamiltonian is bounded from below by the vacuum: $\hat{H}|0\rangle = 0$. Here, we will make the stronger assumption that $Z_\beta \equiv \text{Tr}(e^{-\beta\hat{H}}) < \infty$ for all positive $\beta$. This assumption essentially means that the spectrum of the Hamiltonian is not too dense; it breaks down for example in string theory at the Hagedorn temperature.

In fact, since we will be interested in Poincaré-invariant Lorentzian theories, we want the above assumptions to hold in all boosted frames. This implies boundedness of $\hat{H} - \Omega^a \hat{P}_a$ for any direction $a$ and for $|\Omega| < 1$, which is called the "spectrum condition" [72]. Our stronger assumption, finiteness of $Z_\beta$, generalizes to

$$\forall |\vec{\Omega}| < 1 : \quad Z_{\beta, \vec{\Omega}} \equiv \text{Tr}(e^{-\beta(\hat{H}-\Omega^a\hat{P}_a)}) < \infty . \tag{A.4}$$

### A.1.3 Thermodynamics and KMS conditions

The above spectrum conditions are obviously meant to allow for the definition of a thermal state. For finite $Z_\beta$, we can define the thermal state at inverse temperature $\beta > 0$,

$$\rho_\beta \equiv \frac{1}{Z_\beta}e^{-\beta\hat{H}} . \tag{A.5}$$

Note the similarity with the measure of the canonical ensemble in statistical physics. The more general spectrum condition (A.4) allows to define the state

$$\rho_{\beta, \vec{\Omega}} \equiv \frac{1}{Z_{\beta, \vec{\Omega}}}e^{-\beta(\hat{H}-\Omega^a\hat{P}_a)} . \tag{A.6}$$

This state resembles the grand canonical ensemble, where the $\Omega^a$ are a vector of chemical potentials dual to the momenta $\hat{P}_a$. Note that these states are not rotationally invariant in the full Euclidean plane. The Poincaré symmetry $ISO(d)$ is broken to $ISO(d-1) \times ISO(1)$. Consequently, it does make sense to define "Euclidean time ordering" (in whichever direction the symmetry is broken) for correlation functions in one of these thermal states. We will revisit orderings in more detail in the next section.

---

[21]Note that this is different from the conventions used in radial quantization of CFTs, which is appropriate to obtain unitary Lorentzian theories on a cylinder (spatial sphere times time), instead of flat space.

Correlation functions in the state $\rho_{\beta,\vec{\Omega}}$ take the form

$$\langle \mathcal{O}_1(\tau,\vec{x})\,\mathcal{O}_2(0,0)\rangle^E_{\beta,\vec{\Omega}} = \frac{1}{Z_{\beta,\vec{\Omega}}}\text{Tr}\Big[e^{-\beta(\hat{H}-\Omega^a\hat{P}_a)}\,\mathcal{O}_1(\tau,\vec{x})\,\mathcal{O}_2(0,0)\Big]\,. \tag{A.7}$$

We can write out the trace as a formal sum over the basis of the Hilbert space,

$$\sum_{\alpha,\gamma}\langle\alpha|e^{-(\beta-\tau)E_\alpha+\beta\,\Omega^a P_{a,\alpha}}\mathcal{O}_1(0,\vec{x})|\gamma\rangle\,\langle\gamma|e^{-\tau E_\gamma}\mathcal{O}_2(0,0)|\alpha\rangle\,. \tag{A.8}$$

To analyze when this formal sum converges, we need the spectrum condition (A.4). This leads to

$$0 < \text{Re}\,\tau < \beta(1-|\vec{\Omega}|)\,. \tag{A.9}$$

Whenever this is satisfied, there is no obstruction to rewriting the two-point function as

$$\sum_{\alpha,\gamma}\langle\gamma|e^{-\beta(E_\gamma-\Omega^a P_{a,\alpha})}e^{(\beta-\tau)E_\gamma-i(-i\beta\,\Omega^a P_{a,\alpha})}\mathcal{O}_2(0,0)|\alpha\rangle$$
$$\cdot\,\langle\alpha|e^{-(\beta-\tau)E_\alpha+i(-i\beta\,\Omega^a P_{a,\alpha})}\mathcal{O}_1(0,\vec{x})|\gamma\rangle\,. \tag{A.10}$$

This is suggestively written to make it clear that this again takes the form of a thermal two-point function, but with the order of operators switched. We have found the KMS condition

$$\langle\mathcal{O}_1(\tau,\vec{x})\,\mathcal{O}_2(0,0)\rangle^E_{\beta,\vec{\Omega}} = \langle\mathcal{O}_2(\beta-\tau,-i\vec{\Omega})\,\mathcal{O}_1(0,\vec{x})\rangle^E_{\beta,\vec{\Omega}}\,. \tag{A.11}$$

## A.2 Lorentzian signature

With the unitarity structure given in §A.1.2, the Hermitian generators furnish a representation of the Lorentzian Poincaré algebra $iso(1,d-1)$. We can now consider correlation functions in real time $t = \pm i\tau$. (More on the sign later.) An essential difference with the Euclidean signature, is that "time-ordering" now makes sense. Contrary to the rotations discussed at the end of §A.1.1, boosts do not map a pair of time ordered points to an anti-time-ordered pair.

A complimentary road to the same conclusion arises when the operator $\mathcal{O}(t,\vec{x})$ satisfies an equation of motion (as an operator equation). The most common example is the Klein–Gordon equation, when the two-point function is also a Green function for the equation of motion. It can only be ambiguous if those equations have a homogeneous solution. This does not happen in Euclidean signature because solutions to the Poisson equation are unique. However, in Lorentzian signature it clearly does happen. Because of this, we will list in the following section several inequivalent two-point functions.

### A.2.1 Definitions

We can obtain Lorentzian two-point functions as the Wick rotation of the Euclidean two-point function (A.1). For a rotationally invariant state $\rho$,

$$G_{1,2,\rho}(\tau^2+x^2) \equiv \langle\mathcal{O}_1(\tau,\vec{x})\,\mathcal{O}_2(0)\rangle^E_\rho\,, \tag{A.12}$$

$G$ is some function that depends on the Euclidean distance. We will assume it to be analytic on the positive real line, but due to short-distance divergences it generically has a pole and branch point in the origin. The Wick rotation now consists of considering $G(e^{2i\theta}|\tau|^2+x^2)$ as a function on the complex plane and analytically continuing $\theta$ from 0 to $\pi/2$.

For $\tau^2 > x^2$, we end up on the negative real line. The continuation of $\theta$ to $\pi/2$ and $-\pi/2$ are generically inequivalent, but can be distinguished using $\epsilon > 0$, which is taken to 0 at the end of the calculation.

We can now declare time to be $t \equiv -i\tau$ and define the Feynman propagator or the time-ordered two-point function

$$G^F(t,\vec{x}) \equiv -i \langle \mathcal{T}\{\mathcal{O}_1(t,\vec{x})\mathcal{O}_2(0,0)\}\rangle_\rho \equiv -iG(-t^2 + x^2 + i\epsilon)\,, \qquad (A.13)$$

where we have used $\tau = e^{i\theta}|\tau|$, sending $\theta \to \pi/2$ from below, giving $\tau = it(1-i\epsilon)$. Note that we need to assume that operators commute within Euclidean correlation functions in order to guarantee the property

$$\langle \mathcal{T}\{\mathcal{O}_1(t,\vec{x})\mathcal{O}_2(0,0)\}\rangle_\rho = \langle \mathcal{T}\{\mathcal{O}_2(0,0)\mathcal{O}_1(t,\vec{x})\}\rangle_\rho\,. \qquad (A.14)$$

We can analogously define the anti-time-ordered two-point function

$$G^{\bar{F}} \equiv i \langle \mathcal{A}\mathcal{T}\{\mathcal{O}_1(t,\vec{x})\mathcal{O}_2(0,0)\}\rangle_\rho \equiv iG(-t^2 - x^2 + i\epsilon)\,,$$

where we have done the Wick rotation in the opposite way.

To define non-time-ordered correlators, we can just observe that they have to coincide with either time or anti-time-ordered correlators depending on the sign of $t$. Using the Heaviside function $\theta(t)$, we have for example the Wightman function

$$G^W(t,\vec{x}) \equiv i[\theta(t)G^F(t,\vec{x}) - \theta(-t)G^{\bar{F}}(t,\vec{x})] = G(-(t-i\epsilon)^2 + x^2)\,. \qquad (A.15)$$

This means that we can get the Wightman function from the Euclidean two-point function by substituting $\tau \to i(t-i\epsilon)$ or requiring $\text{Re}\,\tau > 0$. Had we required $\text{Re}\,\tau < 0$ instead, we would have found the Wightman function with opposite operator ordering

$$G_{12}(-(t+i\epsilon)^2 + x^2) = \langle \mathcal{O}_2(0,0)\mathcal{O}_1(t,\vec{x})\rangle_\rho = G_{21}^W(-t,-\vec{x})\,. \qquad (A.16)$$

Because of the branch cut of $G$ for negative values of $\tau^2 + x^2$ we get a non-vanishing commutator of two operators whenever they are timelike separated,

$$\langle [\mathcal{O}_1(t,\vec{x}),\mathcal{O}_2(0,0)]\rangle_\rho = G_{12}[-(t-i\epsilon)^2 + x^2] - G_{12}[-(t+i\epsilon)^2 + x^2]\,, \qquad (A.17)$$

where we have used that operators commute within Euclidean correlators. This quantity vanishes automatically whenever the operators are spacelike separated. This commutator can be split into the retarded and advanced two-point functions

$$G^R(t,\vec{x}) \equiv i\theta(t) \langle [\mathcal{O}_1(t,\vec{x}),\mathcal{O}_2(0,0)]\rangle_\rho\,,$$
$$G^A(t,\vec{x}) \equiv -i\theta(-t) \langle [\mathcal{O}_1(t,\vec{x}),\mathcal{O}_2(0,0)]\rangle_\rho\,. \qquad (A.18)$$

From the definitions above, we can see that the (advanced) retarded propagator vanishes identically outside of the (past) future light cone. Furthermore, the poles of the Wightman function are located at $i\epsilon$. Whenever it has no other poles, it is analytic in the lower half of the complex $t$ plane. This happens for vacuum correlation functions in relativistic theories but does not happen for example in the thermal state, where the correlators are instead constrained by the KMS condition.

### A.2.2 Thermodynamics and the KMS condition

The derivation of the KMS condition (A.11) in Lorentzian time is a bit subtle: the requirement (A.9) of convergence of the trace means that we can only apply the KMS equation for $0 < \text{Re}\,\tau < \beta - \Omega$. As we have seen in §A.2.1, if one keeps a small positive real part in $\tau$ ($\tau = i(t - i\epsilon)$) one obtains the Wightman function $G_{12}^W(t, x)$ on the left-hand side of the KMS equation. On the right-hand side one obtains $G_{21}(\beta - \tau, -\vec{x} - i\vec{\Omega})$. The second inequality in the KMS condition tells us exactly that this also turns into a Wightman function,

$$G_{12}^W(t, \vec{x}) = G_{21}^W(-t - i\beta, -\vec{x} - i\vec{\Omega}) . \tag{A.19}$$

We can express this statement in terms of the momentum space two-point function

$$\tilde{G}(\omega, \vec{k}) \equiv \int \mathrm{d}t\,\mathrm{d}\vec{x}\, e^{i\omega t - i\vec{k}\cdot\vec{x}} G(t, \vec{x}) , \qquad G(t, \vec{x}) = \int \frac{\mathrm{d}\omega\,\mathrm{d}\vec{k}}{(2\pi)^d} e^{-i\omega t + i\vec{k}\cdot\vec{x}} \tilde{G}(\omega, \vec{k}) . \tag{A.20}$$

The KMS condition in Fourier space is then

$$\begin{aligned}
\tilde{G}_{12,\beta,\vec{\Omega}}^W(\omega, \vec{k}) &= \int \mathrm{d}t\,\mathrm{d}\vec{x}\, e^{i\omega t - i\vec{k}\cdot\vec{x}} G_{21,\beta,\vec{\Omega}}^W(-t - i\beta, -\vec{x} - i\vec{\Omega}) \\
&= e^{\beta\omega - \vec{\Omega}\cdot\vec{k}} \int \mathrm{d}t'\mathrm{d}\vec{x}' e^{i\omega t' - i\vec{k}\cdot\vec{x}'} G_{21,\beta,\vec{\Omega}}^W(-t', -\vec{x}') ,
\end{aligned} \tag{A.21}$$

where $t'$ runs over $\mathbb{R} + i\beta$ and $x'^a$ takes values on $\mathbb{R} + i\Omega^a$, for $a = 1 \ldots d - 1$. Assuming that the value of the integral does not change as we move from the original contour to this new one, we get the momentum space KMS condition

$$\tilde{G}_{12,\beta,\vec{\Omega}}^W(\omega, \vec{k}) = e^{\beta\omega - \vec{\Omega}\cdot\vec{k}} \tilde{G}_{21,\beta,\vec{\Omega}}^W(-\omega, -\vec{k}) . \tag{A.22}$$

## B Thermal two-dimensional CFT

Consider a conformal field theory on a Euclidean cylinder with line element $\mathrm{d}s^2 = \mathrm{d}x^2 + \mathrm{d}y^2$, where $x \in \mathbb{R}$ and $0 \le y < 2\pi R_y$. Equivalently, we can use complex coordinates

$$\begin{cases} x + iy = \frac{2\pi R_y}{\beta_L} v \\ x - iy = \frac{2\pi R_y}{\beta_R} u \end{cases} , \qquad (v, u) \cong (v + i\beta_L, u - i\beta_R) , \tag{B.1}$$

where the latter equation encodes the periodicity around the cylinder. The two-point function of a local primary operator $\mathcal{O}$ of weight $h + \bar{h}$ and spin $h - \bar{h}$ is

$$G(v, u) \equiv \langle \mathcal{O}(v, u)\,\mathcal{O}(0, 0)\rangle = I_L(v) I_R(u) , \qquad I_L(v) \equiv \left[ \frac{\beta_L}{\pi} \sinh\left(\frac{\pi v}{\beta_L}\right) \right]^{-2h} , \tag{B.2}$$

where $I_R(u)$ is the function $I_L$ with substitutions $(\beta_L, h) \to (\beta_R, \bar{h})$. For scalar primaries, with $\bar{h} = h$, this two-point function respects the periodicity (B.1). Comparing to the KMS periodicity (A.11), which holds generally for thermal scalar correlators in relativistic theories, leads to the identifications

$$\begin{cases} v = y + i\tau \\ u = y - i\tau \end{cases} , \qquad \begin{cases} \beta = \frac{\beta_L + \beta_R}{2} \\ \Omega = \frac{\beta_R - \beta_L}{\beta_L + \beta_R} \end{cases} , \tag{B.3}$$

where $\tau$ is the "Euclidean time".

As reviewed in §A.2, the Euclidean two-point function can be Wick rotated in several ways to obtain Lorentzian CFT two-point functions. They are again given by (B.2), but now with real coordinates

$$v = y - t , \qquad\qquad u = y + t , \tag{B.4}$$

and with $t \to t - i\epsilon$ for the Wightman function and $t \to t(1 - i\epsilon)$ for the Feynman propagator. We can express these different two-point functions in momentum space to be able to compare with the bulk calculation later on. The Fourier transform of the Wightman function[22]

$$\tilde{G}^W(\omega, p) \equiv \int du\, dv\, e^{i\omega u + ipv} I_L(v + i\epsilon) I_R(u - i\epsilon) = \tilde{I}_L(\omega) \tilde{I}_R(p) , \tag{B.5}$$

separates into two similar integrals. Focusing on the left-moving part first, we can split up the integral into ranges $v > 0$ and $v < 0$ [72]. For positive $v$, the $\epsilon \to 0$ limit of $I_L(v)$ is trivial, giving

$$\tilde{I}_L^{(+)}(p) \equiv \int_0^\infty dv\, e^{ipv} I_L(v) = \sin(h\pi + \tfrac{i\beta_L p}{2}) g_L(p) ,$$
$$g_L(p) \equiv \frac{1}{\pi} \left( \frac{2\pi}{\beta_L} \right)^{2h-1} \Gamma(1 - 2h) \left| \Gamma(h + \tfrac{i\beta_L p}{2\pi}) \right|^2 , \tag{B.6}$$

where we have defined a function $g_L(p)$ that does not depend on the sign of $p$. For negative values of $v$, the (generically fractional) power in (B.2) has a branch cut and the $i\epsilon$-prescription becomes important: it gives a phase $I_L(v \pm i\epsilon) = e^{\mp 2i\pi h} I_L(|v|)$. This implies that for the Wightman function, $\tilde{I}_L^{(-)}(p) = e^{-2i\pi h} \tilde{I}_L^{(+)}(-p)$. The left-moving part of the Wightman function is therefore

$$\tilde{I}_L(p) = \sin(2\pi h) e^{-i\pi h - \frac{\beta_L p}{2}} g_L(p) . \tag{B.7}$$

The right-moving part is analogous, but we will be interested in the extremal limit $\beta_R \to \infty$ for which it becomes $I_R(u) = u^{-2\bar{h}}$. The Fourier transform of the Wightman function has the following contribution from $u > 0$:

$$\tilde{I}_R^{(+)}(\omega) \equiv \int_0^\infty du\, e^{i\omega u} I_R(u) = e^{i\pi(\frac{1}{2} - \bar{h})\sigma} g_R(\omega) ,$$
$$g_R(\omega) \equiv \Gamma(1 - 2\bar{h}) |\omega|^{2\bar{h} - 1} , \tag{B.8}$$

where $\sigma \equiv \text{sign}(\omega)$. The contribution from negative $u$ depends again on the $i\epsilon$-prescription: $\tilde{I}_R^{(-)}(\omega) = e^{2i\pi\bar{h}} \tilde{I}_R^{(+)}(-\omega)$, giving the following left-moving part of the Wightman function

$$\tilde{I}_R(\omega) = 2 \sin(2\pi\bar{h}) e^{i\pi\bar{h}} \theta(\omega) g_R(\omega) , \tag{B.9}$$

where $\theta(\omega)$ is the Heaviside step function. The fact that the right-moving part of the Wightman function vanishes identically for $\omega < 0$ is consistent with $I_R(u - i\epsilon)$ being analytic in the negative $u$-plane.

---

[22]These modes are related to the usual Fourier modes as $e^{i\omega u + ipv} = e^{i\varpi t + iky}$, so $\varpi = \omega - p$ and $k = \omega + p$ are the usual energy and momentum.

The calculation for the momentum-space Feynman propagator is similar but does not separate as straightforwardly because of the $i\epsilon$ prescription. It nevertheless separates into four pieces (an integral for each sign of $u$ and $v$) resulting for bosonic operators (with $h - \bar{h} \in \mathbb{Z}$) in

$$
\begin{aligned}
i\,\tilde{G}^F(\omega, p) &= \tilde{I}_L^{(+)}(p)\tilde{I}_R^{(+)}(\omega) + \tilde{I}_L^{(+)}(-p)\tilde{I}_R^{(+)}(-\omega) \\
&\quad + e^{-2i\pi\bar{h}}\tilde{I}_L^{(+)}(p)\tilde{I}_R^{(+)}(-\omega) + e^{-2i\pi h}\tilde{I}_L^{(+)}(-p)\tilde{I}_R^{(+)}(\omega) \\
&= 2\,e^{-i\pi h}\sin[\pi(h+\bar{h})]\sin\left[\pi h + \sigma\frac{i\beta_L p}{2}\right]g_L(p)\,g_R(\omega)\,.
\end{aligned}
\tag{B.10}
$$

The retarded two-point function only gets a contribution from the future light-cone $u > 0$, $v < 0$,

$$
\tilde{G}^R(\omega, p) = i\left(e^{-2i\pi h} - e^{2i\pi\bar{h}}\right)\tilde{I}_L^{(+)}(-p)\tilde{I}_R^{(+)}(\omega)
\tag{B.11}
$$

$$
= 2\sin(2\pi h)\sin(\pi h + \frac{i\beta_L p}{2})e^{i\pi(\frac{1}{2}-\bar{h})\sigma}g_L(p)g_R(\omega)\,.
$$

This result is consistent with the identity $\tilde{G}^R(\omega) = -\tilde{G}^F(\omega) - i\tilde{G}^W(-\omega)$.

## C  Potentials with arbitrarily many turning points

We consider a Schrödinger equation of the type (2.4) with a potential with arbitrarily many classical turning points where the potential vanishes. In this appendix we will give the expression of the quantity, $\mathcal{A}$, which encodes the information about the potential $V(x)$ for $x < x_+$ as well as the physical boundary condition imposed as $x \to -\infty$ in the response function computed from the WKB hybrid technique (2.17):

- If $V(x)$ has an even number, $2k$, of turning points, $x_1, x_2 \ldots x_{2k}$, one necessarily has $V(x) \geq 0$ in the "interior region," $x < x_1$. The physical boundary condition as $x \to -\infty$ is that $\Psi$ is smooth. Thus, we have

$$
\mathcal{A} = 2\frac{M_{22}}{M_{12}}\,,
\tag{C.1}
$$

where $M_{ij}$ are the elements of the following matrix

$$
M \equiv \begin{pmatrix} -\sin\Theta_{2k-1} & 2\cos\Theta_{2k-1} \\ \frac{1}{2}\cos\Theta_{2k-1} & \sin\Theta_{2k-1} \end{pmatrix} \cdot \prod_{j=1}^{k-1}\begin{pmatrix} \frac{1}{2}e^{-\Theta_{2j}}\cos\Theta_{2j-1} & e^{-\Theta_{2j}}\sin\Theta_{2j-1} \\ -e^{\Theta_{2j}}\sin\Theta_{2j-1} & 2\,e^{\Theta_{2j}}\cos\Theta_{2j-1} \end{pmatrix}\,,
\tag{C.2}
$$

with

$$
\Theta_i \equiv \int_{x_i}^{x_i+1} |V(z)|^{\frac{1}{2}}\,dz\,.
\tag{C.3}
$$

The $\Theta_{2j}$ with even indexes correspond to the tunnelling factors where the potential is positive and the $\Theta_{2j-1}$ give the phase factors from the field oscillations. One should order the matrices in the product according to: $\prod_{j=1}^{k-1}M^{(j)} = M^{(k-1)}\cdot M^{(k-2)}\cdot\ldots\cdot M^{(1)}$.

- If $V(x)$ has an odd number, $2k-1$, of turning points, $x_1, x_2 \ldots x_{2k-1}$, one necessarily has $V(x) < 0$ for $x < x_1$. The interesting physical boundary conditions are those of a black hole in which the modes are required to be purely infalling as $x \to -\infty$. The prescription used for the modes as a function of the time coordinate, $u$, is $\Psi(x, u) = \Psi(x)e^{-i\omega u}$. Thus, we obtain

$$
\mathcal{A} = 2\frac{\bar{M}_{21}}{\bar{M}_{11}}\,,
\tag{C.4}
$$

where $\bar{M}_{ij}$ are the elements of the following matrix

$$\bar{M} \equiv -\mathrm{sign}(\omega) \prod_{j=1}^{k-1} \begin{pmatrix} 2 e^{\Theta_{2j-1}} \cos\Theta_{2j} & -e^{-\Theta_{2j-1}} \sin\Theta_{2j} \\ e^{\Theta_{2j-1}} \sin\Theta_{2j} & \frac{1}{2} e^{-\Theta_{2j-1}} \cos\Theta_{2j} \end{pmatrix} \cdot \begin{pmatrix} e^{-i\frac{\pi}{4}} & e^{i\frac{\pi}{4}} \\ \frac{1}{2} e^{i\frac{\pi}{4}} & \frac{1}{2} e^{-i\frac{\pi}{4}} \end{pmatrix}, \quad \text{(C.5)}$$

with the same definitions as for Equation (C.3) and the same matrix product conventions.

# D Comparison of the exact and approximate response functions

## D.1 The exact and approximate BTZ response functions

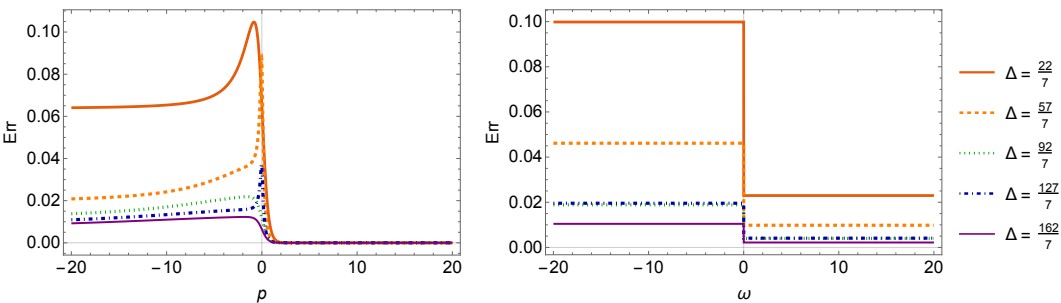

Figure 14: The error function of the WKB formula (2.17), Err (D.1), for an extremal-BTZ geometry as a function of $\omega$, $p$ and $\Delta$. The graph on the left gives Err as a function of $p$ for different values of $\Delta$ and for $\omega = 1/2$ and $r_H = 1$. The graph on the right gives Err as a function of $\omega$ for the same values of $\Delta$ and for $p = 1/2$ and $r_H = 1$.

The exact response function of a scalar field in an extremal-BTZ black hole for $\Delta$ non-integer has been computed in Section 3.2. We want to check the accuracy of the WKB formula (2.17) to retrieve the response function of an asymptotically BTZ geometry. For that purpose, we compare the WKB response function computed in a full-BTZ geometry (see Section 3.2.2) with its exact result, (3.25), by plotting the error function

$$\mathrm{Err} \equiv \left| \frac{R_{WKB}^{\mathrm{BTZ}} - R^{\mathrm{BTZ}}}{R^{\mathrm{BTZ}}} \right|, \quad \text{(D.1)}$$

as a function of $\omega$ and $p$ for different values of $\Delta$ (see Fig.14). From the graphs Fig.14, we observe that the accuracy of the WKB response function depends strongly on the sign of $\omega p$. For positive values of $\omega p$, the WKB formula is extremely close to the exact result for any values of $\Delta$ whereas for negative values, larger values of $\Delta$ give a better accuracy. Moreover, it is really surprising how the accuracy of the formula does not depend on the order of magnitude of $\omega$ and $p$ [23]. The only condition of validity of the formula we derived is that the order of magnitude of $\Delta$ is slightly higher than 1 (for $\Delta \sim 5$ the error is already below 5% for any values of $\omega$ and $p$). This is usually required by the WKB condition on the potential (2.5).

---

[23]Usually the WKB approximation works for large momenta and frequencies.

## D.2 The exact and approximate AdS$_3$ response functions

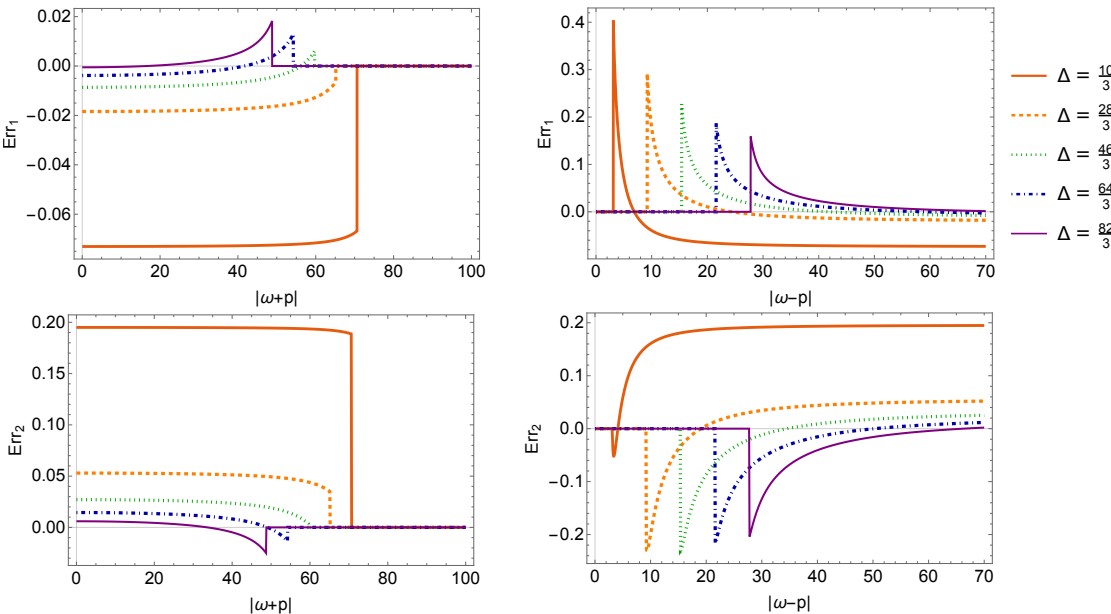

Figure 15: The error functions of the WKB formula (2.17), Err$_{1/2}$ (D.4), for a Global AdS$_3$ geometry as a function of $|\omega - p|$, $|\omega + p|$ and $\Delta$. The graphs on the left give Err$_1$ and Err$_2$ as functions of $|\omega - p|$ for different values of $\Delta$ and for $|\omega + p| = 1/2$. The graphs on the right give Err$_1$ and Err$_2$ as functions of $|\omega + p|$ for the same values of $\Delta$ and for $|\omega - p| = 80$. Each curve has a sharp ending point which corresponds to the point where (3.53) is not satisfied anymore.

Here we check the accuracy of the WKB formula (2.17) for global AdS$_3$. The poles of the WKB response function which give the frequencies and momenta of the normalizable modes are given by $\Theta$ (3.55): $\Theta \in \frac{\pi}{2}\mathbb{Z}$. It is straightforward to see that they match the exact spectrum given by (3.59) if

$$\sqrt{(\omega - p)^2 - 1} \sim |\omega - p| \quad \Rightarrow \quad |\omega - p| \gg 1, \tag{D.2}$$

which is directly satisfied in our regime of parameters (3.53) because of the common validity condition of the WKB approximation: $\Delta \gg 1$ .

Moreover, we can also compute the difference between the WKB result and the exact result as a function of $\omega$ and $p$ as we did for BTZ. However, one cannot use the same error function (D.1) since now the response functions have poles and zeroes which are not exactly at the same location. It is preferable to compare smooth functions with no zero. Thus, we will compare the two exact functions, $g_1(\omega, p)$ and $g_2(\omega, p)$, in (3.58), to their WKB equivalents

$$g_1^{WKB}(\omega, p) \equiv \frac{\sqrt{3}}{2} e^{-2I_+} - \frac{\Psi_{ex}^{grow}(x_+)}{\Psi_{ex}^{dec}(x_+)}, \qquad g_2^{WKB}(\omega, p) \equiv \frac{1}{2} e^{-2I_+}. \tag{D.3}$$

We use a similar error function to (D.1)

$$\text{Err}_i \equiv \frac{g_i^{WKB} - g_i}{g_i}, \qquad i = 1, 2. \tag{D.4}$$

In Fig.15, we have plotted Err$_1$ and Err$_2$ for different values of $\Delta$ as a function of $\omega - p$ and $\omega + p$. As for the BTZ black hole, the WKB result becomes highly accurate as soon as $\Delta$ is quite large and when $|\omega - p|$ is also larger than $\Delta - 1 + |\omega + p|$. Indeed for $\Delta \sim 5$ and $|\omega - p| - \Delta + 1 - |\omega + p| \sim 5$ the error of the WKB formula is already below 5%.

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
