# Peer review of "Thermal Decay without Information Loss in Horizonless Microstate Geometries"

_SciPost Physics, doi:SciPost Phys. 7, 063 (2019)_

## Round 1 · Referee Report · Anonymous (Referee 1) · 2019-9-19

Report

The paper "Thermal decay without information loss in horizonless micro state geometries" is very interesting and technically sound. It addresses a crucial issue i.e. the unitary evolution of physical systems a.k.a. micro-states (regular horizonless geometries) that have the same charges and angular momentum as large D1-D5-P black holes with AdS3xS3 asymptotics. It contains many far-reaching results, a novel technique, dubbed hybrid WKB' analysis, and several illustrative examples. I strongly recommend its publication in "SciPost" after marginal corrections, mostly to fix typos.

After a clear and motivated introduction to micro-states, holography for AdS BH's, and above all to the so-called hybrid WKB approach, in Section 2 the authors discuss how to apply this new technique to holographic correlators and in particular to the Holographic Response Function using exactly solvable problems. Details of WKB analysis and the general procedure are described in Section 3 and successfully applied to BTZ BH's, AdS3 and in Section 4 to a class of micro-state geometries known as superstrata, more precisely of the (1,0,n) kind.
After recalling the necessary ingredients such as the 6-d metric, the massless neutral scalar wave equation, in Section 5 the authors proceed to derive the Response Function for (1,0,n) superstrata by means of their hybrid WKB method, that requires several steps to address the various regimes (cap, extremal BTZ, intermediate BTZ, and centrifugal barrier). In Section 6 the behaviour of position-space Green functions is discussed for the extremal BTZ BH, for AdS3 and for (1,0,n) superstrata that lends support to the unitary thermal decay without information loss of horizonless micro state geometries. The presence of echoes from the highly red-shifted cap region are emphasised that follow the short-time BH-like exponential decay. Section 7 contains final comments and directions for future investigation. Several appendices contain technical details on 2-point functions, thermal 2-d CFT, potential with several turning points, sand comparison of exact and approximate response functions.

The presentation is well-organised clear and detailed, the reader is guided through the intricacies of the computations with mastery by the authors. The analysis is well motivated and far-reaching. The examples chosen to illustrate the procedure are neat.

There is not much room nor need for improvement.

There are some misprints here and there (e.g. Schrdinger four lines below Eq. (2.3) should read `Schr\"odinger'). The subscript $_E$ used for `exact' in Eq. (2.11) and subsequent ones may be confused with the common use of $E$ for Energy (eigenvalue of $H$) or Euclidean as in Appendix A Eq. (A.1). The notation $x<<\infty$ does not make much `physical' sense as it is always true for any real $x$. The authors should identify and indicate the relevant length scale $L$ for which the condition $x<<L$ makes sense. The explicit definition of Airy function $Ai$ and $Bi$ would be very useful. After Eq. (4.9) the statement that $J_R=1/2$ is the minimum value should be explained. Further comments on how to address the case of micro-state of asymptotically flat BH's would be very welcome.
Some reference to the relevant results of the Virgo/LIGO and of the Event Horizon Telescope should be given at the end of the first paragraph in the introduction.

---

## Round 1 · Referee Report · Anonymous (Referee 2) · 2019-9-24

Strengths

(1) This is a very well written paper on an exciting topic.

(2) The paper is technically very solid, and involves several very detailed mathematical computations.

(3) The paper obtains a very physical result on a problem of prime importance -- the black hole information paradox

Report

The standard picture of a black hole with a smooth horizon leads to the information paradox. In string theory there is growing evidence that microstates have no horizon; rather the spacetime ends in a cap just before the horizon is reached. There is no information loss in this situation, but one may ask if the rough properties expected from a traditional hole should hold at least in some approximate way.

This paper considers a set of microstates (constructed earlier by some of the present authors and their colleagues). The microstates describe extremal holes with deep throats; the traditional extremal hole has an infinite throat. With these microstates they compute the 2-point Green's function in the microstate, when the two points are held fixed at the boundary of the AdS region, i.e., at the upper boundary of the throat part of the geometry. It is then found that for short time intervals the Green's function agrees  approximately with the Green's function for the extremal hole, while for long time scales it shows a resurgence as opposed to the decay expected for the traditional hole. This accords well with the fact the information should not be lost down the horizon if the microstate ends in a cap.

To compute the Green's function the author's develop a hybrid WKB technique where the solve the field equation in the outer AdS region and the inner region separately, and match across the middle. This enables an accurate the computation of the subleading part which falls at infinity, and which is needed to compute the response functions for the theory.

The computations are very intricate, and lead to a beautiful answer. The hybrid WKB technique could have applications elsewhere as well, as the authors note.

A suggestion for future study: In general the infall of a quantum down the throat of a generic microstate can lead to a change in the microstate itself; this can happen because generic microstates are very closely spaced in energy due to the large entropy of the hole, Seeing such a change is technically a very complicated problem. I hope the authors consider this more general problem in some later work.

---

## Round 2 · Author Response

We would like to thank the referees for their thorough reading of our paper, for their very positive assessment and for their valuable suggestions. We have addressed their comments in this new version and we hope to be able to revisit their suggestions for further study in the future.

---

## Round 2 · List of Changes

We have:
- corrected misprints such as Schrödinger (p.9 and onward),
- replaced the notation $\Psi_E$ by $\Psi_{ex}$ in order to avoid notational conflict with other conventions (eq(2.12) and onward),
- addressed the confusing statement $x \ll \infty$ (p.11),
- added the definitions of the Airy functions (p.11),
- explained the statement that $J_R = 1/2$ is the minimal value (p.27),
- commented briefly on the difficulty extending our method to flat space superstrata (p.5),
- added references to the Virgo/LIGO and Event Horizon Telescope publications.

You are currently on this page

Resubmission 1905.05194v2 on 25 October 2019

---

## Editorial Decision

published